# WONDERBREAD: A Benchmark for Evaluating Multimodal Foundation Models on Business Process Management Tasks

**Michael Wornow**    **Avanika Narayan**
**Ben Viggiano**    **Ishan S. Khare**    **Tathagat Verma**
**Tibor Thompson**    **Miguel Angel Fuentes Hernandez**    **Sudharsan Sundar**
**Chloe Trujillo**    **Krrish Chawla**    **Rongfei Lu**    **Justin Shen**
**Divya Nagaraj**    **Joshua Martinez**    **Vardhan Agrawal**    **Althea Hudson**
**Nigam H. Shah**    **Christopher Ré**
Stanford University

## Abstract

Existing ML benchmarks lack the depth and diversity of annotations needed for evaluating models on business process management (BPM) tasks. BPM is the practice of documenting, measuring, improving, and automating enterprise workflows. However, research has focused almost exclusively on one task – full end-to-end automation using agents based on multimodal foundation models (FMs) like GPT-4. This focus on automation ignores the reality of how most BPM tools are applied today – simply documenting the relevant workflow takes 60% of the time of the typical process optimization project. To address this gap we present 🔴🔵 **WONDERBREAD**, the first benchmark for evaluating multimodal FMs on BPM tasks beyond automation. Our contributions are: (1) a dataset containing 2928 documented workflow demonstrations; (2) 6 novel BPM tasks sourced from real-world applications ranging from workflow documentation to knowledge transfer to process improvement; and (3) an automated evaluation harness. Our benchmark shows that while state-of-the-art FMs can automatically generate documentation (e.g. recalling 88% of the steps taken in a video demonstration of a workflow), they struggle to re-apply that knowledge towards finer-grained validation of workflow completion (F1 < 0.3). We hope 🔴🔵 **WONDERBREAD** encourages the development of more "human-centered" AI tooling for enterprise applications and furthers the exploration of multimodal FMs for the broader universe of BPM tasks. We publish our dataset and experiments here: ⊙ https://github.com/HazyResearch/wonderbread.

## 1  Introduction

Multimodal foundation models (FMs) such as GPT-4 [43] have the potential to revolutionize **business process management (BPM)**, which is the discipline of measuring and improving enterprise workflows – e.g. a physician submitting a medication order. Typical BPM projects progress in four stages across the following *BPM tasks*: (1) *Documentation* – mapping the steps of an existing workflow; (2) *Knowledge Transfer* – ensuring a shared understanding of the documented workflow; (3) *Improvement* – identifying workflow inefficiencies and proposing fixes; and (4) *Automation* – writing software to execute the workflow without human involvement [53, 57]. Please see Appendix Section A.7 for a concrete example. FMs could be well-suited for these tasks due to their robust reasoning [69, 61, 2] and visual [9, 58, 71] understanding skills.

**However, existing ML benchmarks [73, 65, 18, 64] focus almost exclusively on one BPM task: end-to-end workflow automation** using agents based on multimodal FMs (see Table 1). This is

38th Conference on Neural Information Processing Systems (NeurIPS 2024) Track on Datasets and Benchmarks.

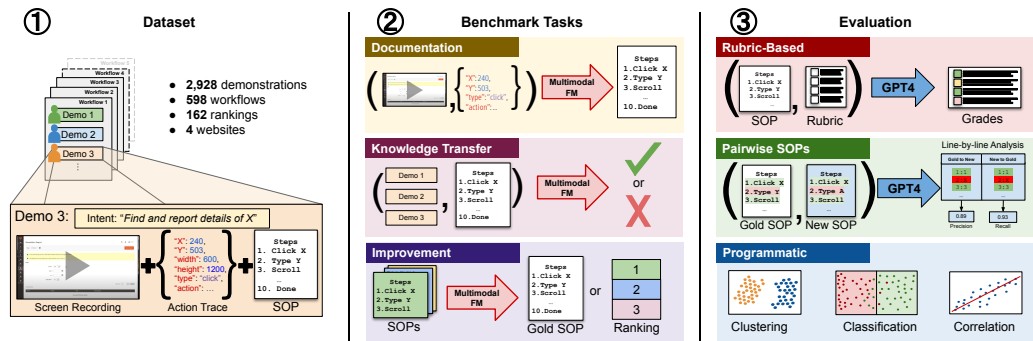

Figure 1: The three components of **WONDERBREAD**. (1) We curate 2928 human demonstrations across 598 web navigation tasks. Each demonstration includes an intent, a full screen recording, an action trace, and a written guide (SOP) describing the steps taken in the demonstration. (2) We create 6 BPM tasks that measure a model's ability to generate accurate documentation, assist in knowledge transfer, and improve workflows. (3) We provide automated evaluation pipelines for all tasks. See Appendix Figure 8 for a detailed example of the data included with each demonstration.

despite the fact that **simply defining the relevant workflow takes 60% of the time** of the typical BPM project [24], and the BPM market is 4x larger than that of automation tools [51, 52, 29, 30].

By **ignoring the most time-consuming aspects of BPM projects,** we overlook key opportunities to provide near-term value to enterprises. Several case studies have applied multimodal FMs to these broader BPM tasks and demonstrated better performance, easier set-up, and simpler maintenance than traditional BPM tools such as process mining [21, 53, 19, 57, 10, 25, 42]. While promising, however, these papers were largely anecdotal with small datasets ($< 50$ examples). **This motivates the creation of a large-scale benchmark and dataset specifically for BPM tasks.**

Unfortunately, no such dataset exists, and **current benchmarks designed around workflow automation cannot be readily repurposed** due to several limitations. First, their datasets either lack human demonstrations of workflows [73, 18] or do not contain sufficient annotation detail for BPM tasks [65, 16, 38, 32] – e.g. evaluating a model's ability to document a workflow requires reference documentation. Second, their evaluations typically only measure end-to-end workflow completion rates [73, 68, 18, 65] and thus do not consider the intermediate reasoning required for BPM tasks such as identifying inefficiencies within a successfully completed workflow. Third, they do not model real-world BPM use cases and instead focus on navigating websites or mobile apps – i.e. they are focused on workflow *execution* rather than *understanding* [16, 48, 68, 17, 32, 73, 35, 18, 70, 65, 38, 32].

Motivated by the overlooked potential for using multimodal FMs on a broader suite of BPM tasks, we thus introduce 🔴🟡🔵 **WONDERBREAD**, a **WO**rkflow u**NDER**standing **B**enchma**R**k, **EvA**luation harness, and **D**ataset. Our contributions are as follows:

1. **Dataset:** We publish **2928 human demonstrations across 598 previously unannotated workflows** sourced from the WebArena benchmark [73]. Each workflow has an average of 4.9 independently collected demonstrations, and each demonstration contains a full screen recording, event log of all clicks/keystrokes/scrolls, and a manually written standard operating procedure ("SOP") – i.e. a step-by-step written guide which reflects the annotator's reasoning at each step of the workflow. For a subset of 162 workflows, we also have annotators rank all 5 demonstrations in order of perceived quality. On average, each workflow takes 7.8 steps and 37.2 seconds. We provide a detailed example of the data in our benchmark in Appendix Figure 8.

2. **Tasks:** Based on use cases drawn from the BPM literature around (1) Documentation, (2) Knowledge Transfer, and (3) Improvement, we define **6 novel BPM tasks** which require reasoning over multimodal data.

   (a) **Documentation:** Generate standard operating procedures (i.e. synthesize the steps of a workflow in writing) to fulfill quality control and audit requirements [5, 63].

(b) **Knowledge Transfer:** Answer user queries about how workflows operate to simplify onboarding and reduce the 5.3 hours per week that knowledge workers spend waiting for information from colleagues.[46].

(c) **Improvement:** Analyze workflows to identify inefficiencies and correct execution errors [20, 55].

3. **Evaluation:** We offer evaluation pipelines using automated metrics (e.g., F1, accuracy) and LLM-based evaluators with high correlation to human raters ($\rho > 0.8$). By focusing on intermediate workflow steps, these evaluations provide a more comprehensive and transparent assessment of models than end-to-end workflow completion rates.

**Results.** We provide baseline results for three state-of-the-art multimodal FMs — GPT-4 [43], Claude 3 [4], and Gemini Pro [54]. Based on screen recordings, we find that models can generate accurate written documentation (F1 of 0.82) and determine whether a demonstration successfully achieved its desired goal (F1 of 0.90). While promising, increasing these numbers to enterprise-level accuracy (i.e. 0.99+) remains an open research challenge. We also identify more significant performance gaps. Models struggle with low-level error correction — for example, when prompted to classify whether a demonstration exactly followed a specific sequence of steps, the peak F1 achieved is 0.27. Models also score poorly when ranking multiple demonstrations of the same workflow on perceived quality and efficiency. We identify long context reasoning, lower-level process understanding, and human workflow preference alignment as key areas for future research.

Our dataset and code available at our Github repo:  `https://github.com/HazyResearch/wonderbread`.

# 2  Background

We summarize traditional process mining approaches for BPM tasks, discuss recent work on applying multimodal FMs, and compare **WONDERBREAD** to existing multimodal FM benchmarks.

## 2.1  Process Mining

Process mining is the *de facto* tool currently used for most BPM tasks, acting as an organizational "X-Ray" [50] that enables large enterprises to identify, measure, and improve their workflows [56, 50, 6]. Techniques include statistical analysis of event logs, unsupervised machine learning, manual review of screen recordings, and user interviews [36, 50]. While interviews can provide an accurate picture of a workflow, they are costly and time-consuming; automated process mining tools are faster but significantly less accurate [1, 36]. Bridging the "semantic gap" between machine and human workflow understanding is an ongoing challenge [41, 36, 1] that **WONDERBREAD** aims to address.

## 2.2  Multimodal FMs

Foundation models (FMs) are large-scale ML models trained on vast datasets of unlabeled data which can be applied to a broad range of tasks with minimal adaptation [12]. Multimodal FMs such as GPT-4 combine natural language understanding with a vision model to process images and text jointly [71]. These models have shown promise in navigating graphical user interfaces and executing simple workflows [16, 67, 26, 27, 70, 64]. While the use of multimodal FMs for BPM tasks has been advocated [53], it has not yet been implemented. A failure mode of text-only FMs is the lack of an ability to "read between the lines" of human-generated textual summaries of workflows – e.g. when creating a process model from text, GPT-4 misses half the steps that a human would include [34, 25]. This motivates having multimodal FMs directly observe workflows, as done in our benchmark.

## 2.3  Benchmarks

A number of multimodal datasets have been published for end-to-end automation of websites [73, 18], mobile apps [48], and desktop applications [64, 65]. However, these datasets do not include step-by-step written guides (SOPs), nor do they evaluate on BPM tasks such as documentation, knowledge transfer, or process improvement [16, 48, 68, 17, 32, 73, 35, 18, 70, 65, 38, 32]. Several works have applied large language models to BPM tasks [21, 53, 19, 57, 10, 25, 42], but they conduct limited case studies (i.e. dozens of examples), rely on manual human evaluation, and do not consider multimodal inputs like screen recordings. Please see Table 1 for a detailed comparison with prior benchmarks.

Table 1: Comparison of **WONDERBREAD** to existing benchmarks for workflows. For **Workflows**, "Env" stands for environment – $W$ is website, $M$ is mobile, and $D$ is desktop. For **Evaluation**, "Auto" means the benchmark contains evaluations for end-to-end workflow automation, "Doc" for documenting workflows, "KT" for knowledge transfer, and "Imp" for process improvement.

| Benchmark | Workflows | | | Human Demonstrations | | | | | Evaluation | | | |
|---|---|---|---|---|---|---|---|---|---|---|---|---|
| | # Tasks | # Envs | Env Type | Action | Video | SOP | Ranking | Demos/Task | Auto | Doc | KT | Imp |
| AITW [48] | 30,378 | 357 | M | ✔ | ✔ | – | – | 23.5 | ✔ | – | – | – |
| Mind2Web [17] | 2,350 | 137 | W | ✔ | ✔ | – | – | 1 | ✔ | – | – | – |
| MoTIF [14] | 6,100 | 125 | M | ✔ | ✔ | – | – | 0.77 | – | – | – | – |
| WebArena [73] | 812 | 4 | W | ✔ | ✔ | – | – | 0.22 | ✔ | – | – | – |
| OmniAct [32] | 9,802 | 65 | D + W | ✔ | – | – | – | 1 | ✔ | – | – | – |
| WebShop [68] | 12,087 | 1 | W | ✔ | – | – | – | 0.13 | ✔ | – | – | – |
| VWA [35] | 910 | 3 | W | – | – | – | – | 0 | ✔ | – | – | – |
| WorkArena [18] | 23,150 | 5 | W | – | – | – | – | 0 | ✔ | – | – | – |
| WebLINX [38] | 2,337 | 155 | W | ✔ | ✔ | – | – | 1 | ✔ | – | – | – |
| OSWorld [65] | 369 | 13 | D + W | ✔ | ✔ | – | – | 1 | ✔ | – | – | – |
| Wonderbread | 598 | 4 | W | ✔ | ✔ | ✔ | ✔ | 4.9 | ✔ | ✔ | ✔ | ✔ |

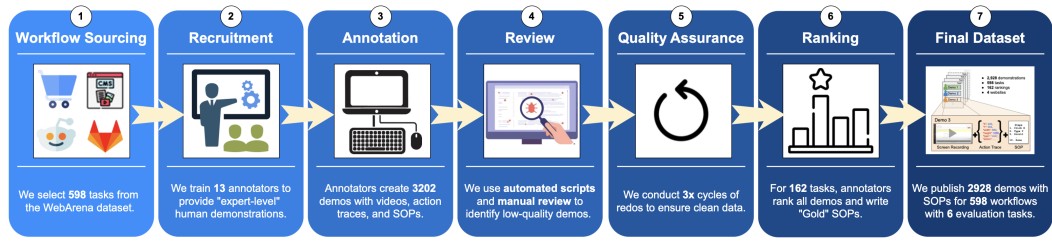

Figure 2: The dataset collection process began by selecting 598 web navigation workflows from the WebArena dataset [73]. Thirteen annotators then recorded themselves demonstrating roughly 300 workflows each. After multiple rounds of QA, annotators ranked demonstrations for 162 workflows based on perceived quality. The final dataset contains 2928 demonstrations and 6 evaluation tasks.

# 3 Dataset

**WONDERBREAD** includes 2928 human demonstrations across 598 distinct workflows. Each demonstration contains:

1. **Intent** – a short natural language description of the workflow's goal
2. **Recording** – a full screen recording of the annotator performing the workflow
3. **Action Trace** – a log of all actions taken (clicks, keystrokes, scrolls) and webpage states before/after each action
4. **Key Frames** – images taken from the Recording at each action's timestamp
5. **SOP** – a written guide detailing all of the steps taken by the annotator

The full dataset collection process is illustrated in Figure 2. Each workflow has demonstrations from at least 4 annotators to reflect the diversity of work habits present in an enterprise. For a detailed example of each data type, please see Appendix Figure 8 and Appendix Section A.2 for several example SOPs. Complete definitions for each demonstration component are provided in Table 2.

We start with WebArena, a benchmark containing 812 workflows that require an agent to navigate open-source clones of an e-commerce, content management, forum, and developer tool website [73]. We filter this to 598 workflows by excluding workflows deemed impossible or inadequately specified. Additional details are provided in Appendix A.3.

We recruited 13 annotators to record themselves completing each workflow using a custom Python script. Existing workflow benchmarks often have low-quality demonstrations or inaccurate annotations [62], thus a key contribution of **WONDERBREAD** is the high quality of demonstrations achieved through several months of quality assurance. More details are provided in Appendix A.3.

Table 2: Key terms and definitions

| Term | Definition | File Format |
|---|---|---|
| Task | One of the 6 evaluation tasks in our benchmark, as detailed in Section 4. | – |
| Workflow | A sequence of actions taken to complete a specific business goal. Also referred to as a process. A single workflow can have multiple demonstrations. | – |
| Demonstration | A single execution of a workflow. Each demonstration contains an Intent, Recording, Action Trace, Key Frames, and SOP. | Folder |
| Intent | A brief natural language specification of a workflow's goal, e.g. *"Cancel my last order"*. | .TXT |
| Recording | A video containing a full recording of the user's screen. | .MP4 |
| Action Trace | A log of all click, keystroke, and scroll actions (including associated elements and coordinates). | .JSON |
| Key Frames | Images taken from a Recording that are synced to events in the Action Trace. | .PNG(S) |
| SOP | A "Standard Operating Procedure" detailing (in writing) all of the steps taken in a demonstration. | .TXT |

In addition to demonstrations, we also curated 120 free response question-answer pairs to simulate inquiries that a BPM consultant might ask of a workflow. Examples are listed in Appendix A.5.

# 4 Benchmark

🔴 **WONDERBREAD** contains 6 tasks which cover three BPM applications not evaluated in prior benchmarks: automatically generating documentation from workflow demonstrations (**Documentation**), facilitating knowledge transfer (**Knowledge Transfer**), and identifying ways to improve inefficient workflows (**Improvement**). We provide a summary of each task below. Further details on the inputs, outputs, and evaluations are in Appendix B. Full prompts associated with each task are included in Appendix F.

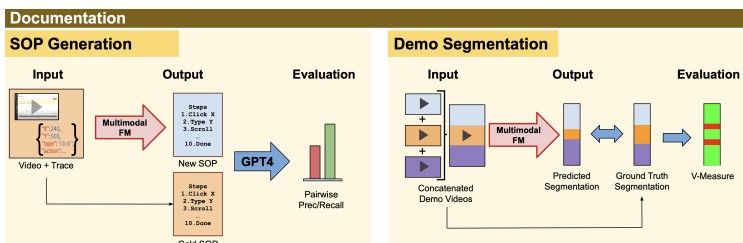

Figure 3: Expected inputs, outputs, and evaluation settings for **Documentation** tasks.

## 4.1 Documentation

Creating clear documentation of complex workflows is essential for operational continuity, compliance, and accountability [63, 5]. This can be achieved through Standard Operating Procedures ("SOP"), Process Definition Documents ("PDD"), or process maps. Our two documentation tasks – SOP Generation and Demo Segmentation – evaluate a model's ability to generate SOPs and accurately distill video recordings into discrete workflows.

**(A) SOP Generation**. Evaluation involves using GPT-4 to compare the generated SOP to an annotator-generated reference SOP, calculating precision (how many steps in the generated SOP are in the reference) and recall (how many steps in the reference are in the generated SOP). Each SOP step is evaluated atomically by GPT-4 for semantic equivalence. Details are in Appendix Section C.2.

**(B) Demo Segmentation**. We concatenate multiple workflow demonstrations into a single video and provide it to the model, which identifies the start and end of each workflow. This tests the model's ability to distinguish between sequential workflows. For evaluation, we calculate the adjusted rand index based on the model's assignment of each video frame to a workflow.

## 4.2 Knowledge Transfer

The sharing of skills, know-how, and best practices within large organizations can be challenging [46]. By learning from workflow demonstrations, FMs could serve as a query-able repository of

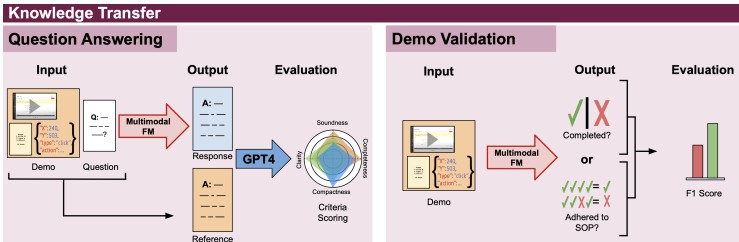

Figure 4: Expected inputs, outputs, and evaluation settings for **Knowledge Transfer** tasks.

organizational knowledge for existing employees, and accelerate on-boarding of new hires by more quickly disseminating key information to trainees [23]. Our two Knowledge Transfer tasks – Question Answering and Demo Validation – assess whether a model can perform higher-level reasoning about the properties and correctness of a workflow.

**(A) Question Answering**. For questions about workflow demonstrations, the model generates a natural language answer, assessing its understanding of workflow semantics. We use GPT-4 to compare the generated answer to a reference answer for evaluation.

**(B) Demo Validation**. Given a demonstration, we predict whether (a) the workflow was successfully completed, or (b) the workflow followed the SOP exactly, with individual steps matching precisely. Since each demonstration in 🔵 **WONDERBREAD** is "correct" by definition, we create synthetic negative examples by truncating recordings and shuffling frames. These binary classification tasks assess a model's ability to self-monitor and error-correct.

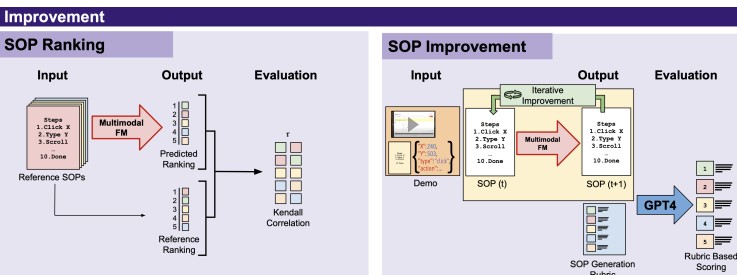

Figure 5: Expected inputs, outputs, and evaluation settings for **Improvement** tasks.

### 4.3 Improvement

The ability to continuously refine and enhance the workflows of an organization is crucial for reducing costs and staying ahead of competitors [20]. By focusing on the improvement of demonstrations and SOPs, we highlight the role of iterative learning and optimization in driving the evolution of workflows [55]. Our two Improvement tasks – SOP Ranking and SOP Improvement – evaluate whether a model can identify workflow inefficiencies and improve inaccurate documentation.

**(A) SOP Ranking**. The same end goal can often be achieved via many different sequences of actions. However, some sequences may be preferable to others as they are more efficient, robust, or avoid intermediate steps that could have undesirable side effects. Given a set of SOPs written by different annotators for the same workflow, this task requires the model to rank them in order of quality. This assesses a model's alignment with human perception of workflow quality. For evaluation, we measure the Kendall $\tau$ correlation between the generated ranking and a human annotator's ranking.

**(B) SOP Improvement**. Given a demonstration and a low-quality SOP, the model must generate an improved SOP that better aligns with the demonstration. The model will iterate to refine the SOP to a specified depth, assessing its ability to assist humans in documenting workflows. GPT-4 will evaluate the generated SOPs against a reference "gold" SOP.

### 4.4 Evaluation

We use programmatic metrics and LLM-based raters for our evaluations. Tasks involving clustering, classification, or ranking use metrics like adjusted rand index, F1, and correlation, respectively.

Natural language tasks are evaluated using GPT-4-as-a-judge to assess input quality [15, 72]. Please see Appendix Table 6 for the specific metrics per task. Our LLM-based evaluations show high correlation with human raters ($\rho > 0.8$) (see Appendix Tables 8 and 9).

## 5 Results

Our initial results show that current multimodal FMs, including GPT-4, Gemini, and Claude, excel at generating documentation which captures the higher-level characteristics of workflows but struggle with finer-grained analyses such as question answering and workflow quality assessment. Our zero-shot evaluations focus on the out-of-the-box capabilities of these models across 162 workflows with rankings. Some models were excluded from specific tasks due to API budget and quota limitations.

### 5.1 Documentation

**(A) SOP Generation.** *Description:* A model must generate a SOP that summarizes all of the actions taken in a video recording of a workflow. We ablate over different demonstration formats: only intent; intent with key frame screenshots; and intent with key frames plus a textual action log of clicks and keystrokes. *Results:* As shown in Table 3, GPT-4 performs best (F1-score of 0.82) with intent, keyframes, and action trace. Most model-demonstration pairs have higher recall than precision (avg. 0.06 points), indicating a tendency to hallucinate workflow steps. Upon qualitative review, we found that many hallucinated actions seemed reasonable but were not actually taken in the demonstration, e.g. adding *"Navigate to the shopping admin page"* even though the demonstration started on that page. Exact scores for each workflow and model are in Appendix Figure 10.

Table 3: **SOP Generation:** Accuracy of generated SOPs versus ground truth SOPs.

| Model | Intent | Keyframes | Trace | Precision | Recall | F1 | Avg. # of Steps |
|---|---|---|---|---|---|---|---|
| GPT-4 | ✓ | ✓ | ✓ | **0.80** | **0.88** | **0.82** | 10.26 |
| GPT-4 | ✓ | ✓ | | 0.69 | 0.79 | 0.71 | 10.32 |
| GPT-4 | ✓ | | | 0.48 | 0.59 | 0.49 | 13.10 |
| Claude 3 Sonnet | ✓ | ✓ | ✓ | 0.72 | 0.85 | 0.76 | 10.94 |
| Claude 3 Sonnet | ✓ | ✓ | | 0.67 | 0.78 | 0.70 | 11.35 |
| Claude 3 Sonnet | ✓ | | | 0.53 | 0.54 | 0.50 | 11.34 |
| Gemini Pro 1 | ✓ | ✓ | ✓ | 0.58 | 0.63 | 0.58 | 11.09 |
| Gemini Pro 1 | ✓ | ✓ | | 0.48 | 0.51 | 0.46 | 11.28 |
| Gemini Pro 1 | ✓ | | | 0.40 | 0.36 | 0.34 | 7.31 |
| Ground Truth | ✓ | ✓ | ✓ | 1 | 1 | 1 | 8.40 |

**(B) Demo Segmentation.** *Description:* This task mimics what a video recording of a person's screen would capture during the typical workday, i.e. multiple workflows without clear boundaries. Concretely, the model receives $k$ concatenated demonstrations sampled from different workflows from our dataset, and must determine which frames belong to the same workflow. We set $k = 3$ and choose workflows that utilize the same website. *Results:* As shown in Table 4, segmenting a recording remains challenging. Providing additional information via an SOP and intent slightly increases performance for GPT-4 yet decreases performance for Gemini Pro 1. On inspection, we find that the frequency at which Gemini Pro 1 outputs blank state mappings (i.e. not assigning a keyframe to any workflow, which under our evaluation framework gets penalized as an incorrect mapping) increases with longer prompts, indicating a worse ability to follow the full context of the prompt.

Table 4: **Demo Segmentation:** Accuracy of clustering with $k = 3$ concatenated workflows.

| Model | Intent | SOP | Keyframes | Adj. RI | V-Measure |
|---|---|---|---|---|---|
| GPT-4 | ✓ | ✓ | ✓ | **0.85** | **0.88** |
| GPT-4 | | ✓ | ✓ | **0.85** | 0.87 |
| GPT-4 | | | ✓ | 0.80 | 0.86 |
| Gemini Pro 1 | ✓ | ✓ | ✓ | 0.55 | 0.66 |
| Gemini Pro 1 | | ✓ | ✓ | 0.53 | 0.65 |
| Gemini Pro 1 | | | ✓ | 0.58 | 0.69 |

## 5.2 Knowledge Transfer

**(A) Question Answering.** *Description:* This task involves answering 120 free response questions about workflows, such as *"How would a user know the workflow is complete?"* and *"What is the purpose of this workflow?"*. These questions were drawn from the process mining literature [10, 21] and are provided in Appendix A.5. We use GPT-4-as-a-judge to evaluate model-generated answers by comparing to a reference answer from a human annotator. Following prior work [21], we have GPT-4 output four scores on a scale of 1 (bad) to 3 (good): completeness, soundness, clarity, and compactness. The Pearson correlation between GPT-4 and human raters was between 0.80 and 0.89 across all axes (see Appendix Table 8). *Results:* Results are shown in Figure 6. All models perform well in "compactness" and "clarity" but score lower on "soundness" and "completeness." The former two are measures of the syntactic quality of writing, while the latter two are measures of the accuracy of the answer. As "soundness" measures whether an answer avoids containing inaccurate details, these lower scores can be explained by the tendency of LLMs to hallucinate and infer information based on patterns learned from training data (i.e. includes content from websites like GitLab, Amazon, etc.) that are not present in the specific demonstrations in **WONDERBREAD** [28]. Lower scores on "completeness" may be due to the difficulty of fully attending to multimodal prompts with multiple states and actions [37, 39], thus leading to occasional omissions of relevant details.

**(B) Demo Validation.** *Description:* We consider two forms of validation: (a) workflow completion, where a demonstration is "correct" if the workflow's goal is achieved; and (b) workflow trajectory, where it is "correct" only if the goal is achieved and the steps taken exactly follow a specific SOP. "Correct" examples are sampled from our dataset, while "incorrect" examples are created by truncating, shuffling, or skipping states. *Results:* As shown in Table 5, GPT-4 performs best. It can accurately determine whether a workflow completed its overall goal (F1 of 0.90) but struggles to validate that a demonstration followed the specific steps of an SOP (F1 of 0.27).

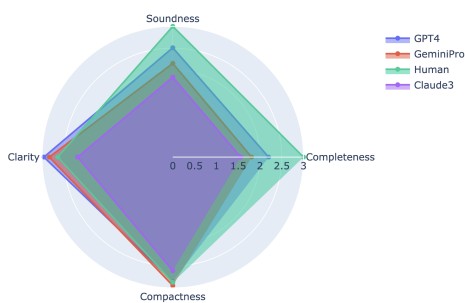

Figure 6: **Knowledge Transfer:** Scores across four axes – soundness, completeness, clarity, and compactness – across 120 free response questions for evaluating workflow understanding.

Table 5: **Demo Validation:** Accuracy on binary classification of whether a workflow was completed (*Completion*) or followed the exact steps outlined in the SOP (*Trajectory*).

| Model | Intent | Keyframes | SOP | Precision | Recall | F1 |
|---|---|---|---|---|---|---|
| **Completion** | | | | | | |
| GPT-4 | ✓ | ✓ | ✓ | **0.89** | **0.90** | **0.90** |
| GPT-4 | ✓ | ✓ | | 0.84 | 0.77 | 0.81 |
| Gemini Pro 1 | ✓ | ✓ | ✓ | 0.94 | 0.25 | 0.40 |
| Gemini Pro 1 | ✓ | ✓ | | 0.94 | 0.26 | 0.41 |
| Claude3 Sonnet | ✓ | ✓ | ✓ | 0.58 | 0.31 | 0.40 |
| Claude3 Sonnet | ✓ | ✓ | | 0.85 | 0.50 | 0.63 |
| **Trajectory** | | | | | | |
| GPT-4 | ✓ | ✓ | ✓ | 0.52 | **0.18** | **0.27** |
| Gemini Pro 1 | ✓ | ✓ | ✓ | **0.94** | 0.14 | 0.25 |

## 5.3 Improvement

**(A) SOP Ranking.** *Description:* In this task, we provide a model with SOPs from various annotators and have it rank them by quality. We then compare this ranking to a ground truth ranking by an annotator and measure the correlation between the model's and human's judgments. *Results:* As shown in Table 7a, current models struggle to rank SOPs based on perceived quality to human raters. The best model achieves a mean Kendall correlation of 0.05 with a standard deviation of 0.47, indicating essentially random rankings. Improving alignment between model and human judgment of workflow quality remains an area for further research.

**(B) SOP Improvement.** *Description.* In this task we provide a model with a task recording and an SOP. The model is then tasked with subsequently improving the SOP given and SOP rubric. *Results.* As shown in Table 7b, current models are capable of improving the quality of their own SOPs (up to 1.4 points), conditioned upon a SOP rubric.

| Model | Spearman $\rho$ | Kendall $\tau$ |
|---|---|---|
| GPT-4 | **0.07 ± 0.58** | **0.06 ± 0.49** |
| Claude3 Sonnet | 0.06 ± 0.59 | 0.03 ± 0.50 |
| Gemini Pro 1 | 0.03 ± 0.58 | 0.03 ± 0.49 |

| Model | Original SOP | Improved SOP |
|---|---|---|
| GPT-4 | 3.43 | **4.82** |
| Claude3 Sonnet | 3.43 | 4.26 |
| Gemini Pro 1 | 3.43 | 3.65 |

(a) **SOP Ranking:** Corr. between model and human rankings of demonstrations for the same workflow.

(b) **SOP Improvement:** Scores from 1 (bad) to 5 (good) for SOPs before/after model improvement.

Figure 7: Results for the two **Improvement** benchmark tasks.

## 6   Discussion

We discuss next steps, limitations, and the broader impacts of **WONDERBREAD** below.

**Improving Human-Model Alignment for BPM Tasks.** We find that out-of-the-box human and multimodal models alignment is low for SOP evaluation (see Section 5.3). Similar to how "human-model" alignment can be achieved for tasks like question-answering and instruction-following [59, 33], alignment also appears necessary for workflow understanding tasks. This might require fine-tuning models via supervised learning [60] or reinforcement learning on preference data [44].

**Expanding Multimodal Context Windows.** Even a 1-minute workflow can generate dozens of actions and key frames. Our results show that model accuracy on BPM tasks improves as more information is provided in the prompt. This might not be possible with longer workflows, leading to an incomplete representation for a workflow and lower downstream task performance. Longer context windows can help solve this problem and are a focal point of study in the community [31, 66].

**Low-Level Workflow Understanding**. Our results show that while multimodal FMs excel in high-level workflow analyses, they struggle with precise validation of individual steps (see Section 5.2). Enhancing this lower-level understanding may require supervised fine-tuning on GUIs as in [27, 7].

**Self-Improvement.** Our findings suggest that multimodal FMs can improve their outputs (i.e., SOPs) through multiple iterations of self-reflection (see Section 5.3). This highlights the potential of these models to refine their outputs without human intervention [22, 3]. In the context of BPM tasks, this capability can help systems adapt to workflows as they change over time.

**Limitations.** There are several limitations to our work. First, dataset construction was constrained by our lack of access to real-world enterprise data due to privacy concerns. Second, the workflows in our dataset are taken from a limited set of 4 websites [73], and it is unclear how our results generalize to different environments with complex or longer workflows. Contemporaneous to our work, several datasets have been released which could be re-annotated following the process described in our paper [65, 38, 32], which we leave to future work. Third, our baseline results lack open-source models. Matching the performance of state-of-the-art proprietary models on these benchmarks with open source models remains an open research challenge.

**Scaling.** To our knowledge, **WONDERBREAD** is currently the largest dataset for BPM tasks. However, it is still limited in its ability to capture the broad variety of real-world enterprise workflows. Scaling the approach outlined in this paper represents an exciting future research direction. We propose several ways to increase the size and diversity of data: (1) Synthetically generate demonstrations using AI agents trained on existing workflow examples and reject invalid demonstrations, as detailed in [8, 45]. (2) Crowdsource human demonstrations through platforms like Amazon Mechanical Turk. (3) Collaborate with a large enterprise willing to deploy our recording script to collect real-world workflows. (4) Scrape how-to videos and screen recordings of workflows from sites like Youtube.

**Societal Impact.** Our field's collective focus on end-to-end automation contradicts recent advocacy for more *human-centered AI*, which aims to *augment* rather than *replace* human labor [47, 49, 13, 11]. While we intend for **WONDERBREAD** to serve as a counterpoint to this focus, we acknowledge that any AI tools aimed at improving productivity run the risk of replacing human labor.

# 7 Conclusion

We present 🔴🔵 **WONDERBREAD**, the first benchmark for evaluating multimodal models on common process mining tasks. It includes 2928 human demonstrations across videos, images, and text, along with step-by-step written guides (SOPs) and full action traces. We focus on applying these models to three BPM tasks that have been overlooked by existing ML benchmarks for workflow automation – documentation, knowledge transfer, and process improvement. 🔴🔵 **WONDERBREAD** features an automated evaluation harness with programmatic metrics and LLM-based assessments, providing baseline results for state-of-the-art multimodal models. Our work aims to inspire further efforts to support workers by *augmenting* rather than *replacing* human labor.

## Acknowledgments and Disclosure of Funding

MW is supported by the NSF Fellowship, a Stanford HAI Graduate Fellowship, and Stanford Healthcare. AN is supported by the Knight-Hennessy Fellowship and the NSF fellowship. We thank Neel Guha, Dan Fu, Mayee Chen, Eric Nguyen, Jordan Juravsky, Jerry Liu, and Sabri Eyuboglu for providing helpful feedback on this manuscript. We gratefully acknowledge the support of NIH under No. U54EB020405 (Mobilize), NSF under Nos. CCF2247015 (Hardware-Aware), CCF1763315 (Beyond Sparsity), CCF1563078 (Volume to Velocity), and 1937301 (RTML); US DEVCOM ARL under Nos. W911NF-23-2-0184 (Long-context) and W911NF-21-2-0251 (Interactive Human-AI Teaming); ONR under Nos. N000142312633 (Deep Signal Processing), N000141712266 (Unifying Weak Supervision), N000142012480 (Non-Euclidean Geometry), and N000142012275 (NEPTUNE); Stanford HAI under No. 247183; NXP, Xilinx, LETI-CEA, Intel, IBM, Microsoft, NEC, Toshiba, TSMC, ARM, Hitachi, BASF, Accenture, Ericsson, Qualcomm, Analog Devices, Google Cloud, Salesforce, Total, the HAI-GCP Cloud Credits for Research program, the Stanford Data Science Initiative (SDSI), and members of the Stanford DAWN project: Facebook, Google, and VMWare. The U.S. Government is authorized to reproduce and distribute reprints for Governmental purposes notwithstanding any copyright notation thereon. Any opinions, findings, and conclusions or recommendations expressed in this material are those of the authors and do not necessarily reflect the views, policies, or endorsements, either expressed or implied, of NIH, ONR, or the U.S. Government.

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

# A Dataset

## A.1 License & Availability

We license our code and dataset under the Apache 2.0 license. The authors bear all responsibility in case of violation of rights. Our code and data are available here: ⦿ https://github.com/HazyResearch/wonderbread.

Our dataset is based on the WebArena benchmark [73], which also has an Apache 2.0 license and is available here: https://github.com/web-arena-x/webarena

## A.2 Example Standard Operating Procedures (SOPs)

In this section, we include 3 example SOPs from our dataset.

Below is an SOP describing how to accomplish **Task #1** (*"What is the top-1 best-selling brand in Quarter 1 2022?"*) taken from demonstration **"1 @ 2023-12-25-17-08-00"**.

```
1. On the left bar, click on the "Reports" button.
2. Under the "Products" header, click on "Bestsellers".
3. Click in the box next to the "From" field.
4. Enter the first date of Quarter 1 2022, which is 01/01/2022.
5. Click in the box next to the "To" field.
6. Enter the last date of Quarter 1 2022, which is 03/31/2022.
7. Click on the "Show Report" button on the top right side of the page.
8. From the table shown, report the top-1 best-selling brand in Quarter 1 2022.
```

The SOP below was also written for **Task #1**, but by a different annotator for demonstration **"1 @ 2023-12-25-15-44-04"**.

```
1. Click on the "Reports" button on the far lefthand sidebar. It has an icon
   which looks like a chart. It should be located directly above the "Stores"
   button and below the "Content" button.
2. In the popup menu that appears, click on the "Bestsellers" link to go to
   the "Bestsellers Report" page. The link should be located under the
   "Products" section.
3. Click on the "From" textbox to focus it. It should be located directly
   underneath the "Period" field.
4. Type in the first day of our desired time period, which in this case
   is "01/01/2022"
5. Click on the "To" textbox to focus it. It should be located directly
   underneath the "From" field.
6. Type in the last day of our desired time period, which in this case
   is "03/31/2022"
7. Click on the orange "Show Report" button, which can be found on the
   top right of the page, in order to generate our best-selling product report.
8. The best-selling products will appear in a table at the bottom of the
   page. Scroll down through each row of the report until you reach the bottom
   of the page. While scrolling, keep track of which brand has the sold the
   greatest quantity. The quantity for each product is found in the "Order
   Quantity" column of the bestsellers table, on the right hand side of the page.
   The name of the product, containing information on the brand of the item, is
   found in the Product column of the Bestsellers table.
9. The product brand that has the greatest total order quantity is the best
   selling product brand in quarter 1 2022.
```

Below is an SOP for a different task – **Task #494** (*"Notify Alex Thomas in their most recent pending order with message 'Yo, your order will be shipped soon!'."*) – which was written for demonstration **"494 @ 2023-12-30-23-48-17"**.

```
1. Click on the "SALES" option in the left side bar under the "DASHBOARD" option.
2. Click on the "Orders" option in the "Sales" menu that appeared.
3. Type "Alex Thomas" in the "Search by keyword" search bar.
4. Click on the magnifying glass in the "Search by keyword" search bar.
```

```
5. Click on the blue "View" link under the column "Action" corresponding
   to the "000000304" order.
6. Scroll down until you see the brown "Submit Comment" button in the "Order
   Total" section.
7. Type "Yo, your order will be shipped soon!" in the "Comment" text box under
   the "Status" dropdown menu.
8. Click on the "Notify Customer by Email" checkbox under the "Comment" text box.
9. Click on the brown "Submit Comment" button under at the bottom left of the
   screen.
```

## A.3   Dataset Curation

**1. Workflow Selection**. We begin with the WebArena [73] benchmark, which is a collection of 812 workflows instantiated from 187 workflow intents. For example, the template "Search for (term)" could have instantiations "Search for *jacket*" and "Search for *coat*". These 812 tasks require an agent to navigate fully functional open source clones of popular websites. In this dataset we use the e-commerce, content management system (Adobe Magneto), forum (PostMill), and developer tool (GitLab) sites provided by WebArena. We find that many workflows in WebArena are designed to be impossible, are *de facto* impossible, are underspecified, or have incorrect evaluations, and we purposely exclude these workflows from our dataset.

First, several workflows in WebArena are designed to be impossible. These are the workflows that have a correct answer marked as "N/A". For example, one workflow has the intent "What are the main criticisms of this product?" and marks the correct answer as "N/A" since there are no criticisms. We remove all of these workflows. An example of a *de facto* impossible workflow is "Assign the issue regarding flash alerts to myself and primer." Though there is a non-N/A answer for this workflow, upon manual inspection we found that the Gitlab interface does not actually allow issues to be assigned to more than one user, and thus we removed it from our dataset. Underspecified workflows are those whose answer we found arbitrary upon manual inspection. For example, an intent such as "Show me the email address of the customer who is the most unhappy with Circe fleece" is underspecified as the phrase "most unhappy" is unquantifiable when there are multiple one-star reviews. We remove all of these underspecified workflows. Finally, we exclude workflows that have valid intents but whose expected answers were deemed incorrect upon manual inspection. Example workflows from the Webarena dataset with these mistakes include "the number of commits of the contributor who has the most commits to branch main" in Gitlab being stated incorrectly or that "the amount spent on home decoration shopping during 1/29/2023" being calculated incorrectly are also excluded. Finally, we ignore the 23 workflows in WebArena that include multiple websites. We do this for simplicity as our recording script could only handle one website at a time. This left us with a final total of 598 workflows.

**2. Annotator Recruitment and Training**. We enlisted 13 human annotators from a pool of approximately 60 applicants (all students at Stanford University) to participate in our data collection process. All selected annotators, who are undergraduate or graduate students at Stanford University with proficient computer literacy skills, were fully informed and consented to the publication of their complete demonstrations. They were also given the opportunity to review the entire codebase, experiments, and manuscript prior to submission. Annotators were aware that their full screen recordings would be made public and were advised to remove any personally identifiable information before recording. Prior to applying, they were informed that there would be no monetary compensation, as their participation would be on a voluntary basis for a research project.

An important distinction from the demonstrations contained in our dataset versus prior work is that our annotators were explicitly instructed not to perform "zero-shot" recordings, meaning annotators were told to rehearse each task before recording to ensure that the collected demonstrations were free of mistakes. More specifically, annotators were told to follow these principles:

- We are simulating **expert** users of the interface.
  - Do the optimal (i.e. most direct) way to complete each task.
  - Ensure that your demonstration contains no wasted clicks / typing.
  - Ensure that your demonstratoin has no mistakes – If you make a mistake while performing the demonstration, stop recording and re-record from scratch.
- We want a **clean** dataset

- When you record, ensure that the selected interface within Google Chrome is visible.
- Ensure you do not show any other applications.
- Ensure you do not show personal information.

Therefore, the final dataset has a 100% task completion rate. In contrast, in the original WebArena benchmark [73], untrained human annotators could only complete 78% of tasks.

**3. Data Collection**. Each annotator utilized a custom Python script to record demonstrations of approximately 300 unique tasks. This script operated in the background while the annotator completed the demonstration, capturing and outputting four primary types of data: (1) a JSON trace detailing all user actions (clicks, keystrokes, and scrolls), including the precise HTML state of the website at the time of each action and attributes of the elements interacted with; (2) a video of the full screen recording of the annotator's computer; (3) a collection of screenshots corresponding to each recorded action; and (4) an initially blank Standard Operating Procedure (SOP) file.

Once the recording was complete, each annotator filled out the SOP file, creating a detailed, step-by-step list of the actions they performed. Annotators were directed to explain these steps with the simplicity and clarity necessary for a five-year-old to follow. The annotators were instructed to provide the level of detail that a 5-year-old would need to complete the task. Finally, annotators assessed the difficulty of each task, classifying them as Easy, Medium, or Hard. On average, each annotator dedicated approximately 30 hours to this process, amounting to a collective total of nearly 300 man-hours of labeling over several months.

**4. Demonstration Ranking**. After completing the dataset collection process, we chose a subset of 162 tasks (all derived from different task templates) to form our collection of "Gold Workflows". Each annotator was then tasked with watching the demonstrations of approximately 15 "Gold Workflows", relatively ranking the demonstrations of the same task from 1 (best) to 5 (worst). The annotators then developed a more thorough SOP we call a "Gold SOP" based on the demonstration that received the top ranking. This process resulted in 162 tasks in our dataset containing demonstrations of ranked relatively quality, along with high quality "Gold SOPs" we use as the highest quality SOP representation of the "Gold Task"'s demonstrations. More details about this ranking procedure are included in Appendix A.6.

**5. Quality Assurance**. A key contribution of **WONDERBREAD** is high quality human task demonstrations. A review of existing benchmarks for web navigation tasks found consistently low quality demonstrations that have inaccurate annotations (e.g. misplaced bounding boxes for HTML elements) [62]. This made quality assurance a key concern while curating **WONDERBREAD**. We performed three rounds of quality assurance checks over the course of two months using a combination of automated scripts, manual review, and cross-referencing demonstrations across annotators. We had annotators redo any tasks that were of insufficient quality, and discarded any tasks that had less than 4 successful demonstrations. Additional details are available in the Appendix A.4.

**6. Workflow Understanding Questions**. To enable deeper evaluations of a model's workflow understanding, we also created a set of 11 free responses question templates, which are listed in Appendix A.5. These questions attempted to simulate actual inquiries that a BPM consultant might ask. Examples include *"Explain what the most common failure modes might be for a user performing this task"* and *"Why does the user click the "Commits" button in step #5?"*. We created 10 instances of all question templates, and an additional 10 instances for question template #2. This gives a total of 120 questions. We then had had a set of annotators write brief free-form answers based on the corresponding task.

### A.4 Quality Assurance

We ran a series of automated scripts to flag systematic errors, and had our annotators redo any tasks that were flagged. For example, we verify that all actions occur within Google Chrome and that major disagreements between annotators on each task are resolved. For example, we cross-reference task demonstrations across annotators and redo tasks where someone marked it as infeasible while someone else marked it as feasible. We also conduct manual review of all demonstrations corresponding to the 179 Gold workflows, as well as a random sampling of 300 other demonstrations across all tasks.

### A.5 Question Answering Dataset Questions

Listed below are the free response questions templates that we created for our Question Answering task, largely inspired by prior work in the process mining literature [21, 10].

1. Explain what the most common failure modes might be for a user performing this task.
2. How would a user completing the task know that the workflow is completed?
3. What is the purpose of doing this workflow?
4. What if instead of X we wanted to do Y. How would you change this workflow to accomplish that?
5. Why does the user click the button X in step #Z?
6. Why does the user click the button X in screenshot #Y?
7. Why does the user type the string X in step #Z?
8. Why does the user type the string X in screenshot #Y?
9. How would a user completing the task know that the workflow is completed?
10. Here are two workflows. Please identify the key differences between them.
11. Here are two demonstrations, one of which is more efficient than the other. Please describe ways to improve the less optimal workflow.

Question templates #1-9 only involve reasoning over a single demonstration, but #10-11 require reasoning over multiple demonstrations.

After creating these question templates based on our review of question types asked in prior work on process mining [10] [21], we then transformed them into concrete questions instantiated with specific demonstration(s) from our dataset. In other words, turning "Why does the user click the button X in step #Z?" into "Why does the user click on ""Not Approved"" in step #4?" for demonstration 79 @ 2023-12-27-22-50-34. This was accomplished in three steps. First, for each question template we first came up with a list of characteristics that a linked demonstration would need. For example, a workflow with no button clicks would not be a viable candidate for the question template "Why does the user click the button X in step #Z?." Next, we randomly sampled demonstrations without replacement from our set of Gold demonstrations until we came up with 10 instantiations of each question (20 for question template #11). Finally, we conducted two rounds of manual review to write "ground truth" answers and ensure each question was instantiated correctly.

### A.6 Factors for Quality of Gold SOPs

Listed below is the information given to annotators to aid them with writing high-quality Gold SOPs.

1. **Coverage of edge cases** – help the user complete the task by making note of ways in which the interface might change, and how to adapt:
   - e.g. If a task involves looking through a table of shipping orders to find a specific order, and your specific order just happens to be the first one, you should still make a note that the user might have to scroll / paginate through the results until they find the correct shipping order.
   - e.g. If you need to click a button at the bottom of a page, you should not assume that the user's browser window has the same size as yours, so you should let them know that they might need to scroll down if they can't see the button.
   - **Example**: Instead of "Click on the toggle labeled 'Enable Product'", you "should write "Look for the toggle labeled "Enable Product" which should be directly below the "Quantity" field. If the toggle is currently green, that means the product is currently enabled, which means you should click the toggle in order to disable the product. The toggle should change to a grey color to indicate the product is disabled. However, if the toggle is already greyed out, then do nothing since the product had already been disabled."
2. **Detailed localization of UI elements** – let the user know exactly where to find the element

- ○ e.g. "Click the 'Go to Result' button" is not sufficient. You must be extremely detailed in your specification of each element, i.e. its relative position on the screen, its proximity to other landmark elements, its color, what type of element it is, etc.
- ○ **Example**: Instead of "Click the 'Edit' link", you should write "Click on the blue "Edit" link at the far righthand side of the row corresponding to the "Configurable Product" we previously found."

3. **Generalizability** – the instructions should be written so that they could apply to any instantiation of the **Intent Template** corresponding to the task
   - ○ e.g. The instructions should be written generally, providing task-specific information as asides.
   - ○ **Example**: Instead of "Type "Out of Office" in the "What's your status?" input box.", you should write "Type the desired Gitlab status in the "What's your status?" input box. In this case, we should type "Out of Office""

4. **Explanations of each action** – briefly explain why we take each step (in the context of the next action, or the larger task)
   - ○ e.g. What is the point of each individual action?
   - ○ Example: Instead of "Click the "From" text field", you should write "Click the "From" text field to focus it."
   - ○ Example: Instead of "Click on the toggle labeled 'Enable Product'", you should write "Click on the toggle labeled 'Enable Product' to disable the product."

## A.7 Example Hypothetical BPM Project

For clarity, we provide the following as an example of what a hypothetical BPM project might entail. Let's say a hospital wants to accelerate the workflow by which admitted patients have their insurance verified. Today, the process is done completely manually by a team of billing specialists. The workflow involves copying the patient's demographic information into several databases and visiting an insurer's web portal to verify that the patient's insurance coverage is accurate and up-to-date. A hypothetical BPM project for accelerating this workflow might progress as follows: (1) Documentation: First, a business development (BD) analyst interviews all of the billing specialists on the team, conducts shadowing sessions over Zoom, and watches screen recordings collected by the team in order to create written documentation of the insurance verification workflow. (2) Knowledge Transfer: After creating a draft of the workflow, the BD analyst hosts a series of in-person brainstorming sessions with the team; the BD analyst identifies several gaps in her current understanding of the workflow, and they collaboratively arrive at a shared consensus of all steps involved in the end-to-end workflow. (3) Improvement: From these conversations, the BD analyst identifies several bottlenecks and inefficiencies; for example, entering the patient's demographic information into multiple databases that could instead be automatically synced, or waiting for the approval of another department that has a turnaround of one week but isn't strictly necessary. The BD analyst then draws a new, more streamlined workflow diagram and shares her findings with the billing team to implement. (4) Automation: Based on these observations, the billing team believes that several of the subtasks within this new workflow might be automatable. They enlist the help of the hospital's IT department to build an integration between their two database applications to automate this data entry. They also work on developing a robotic process automation (RPA) bot that can navigate multiple screens to automatically submit forms to insurers.

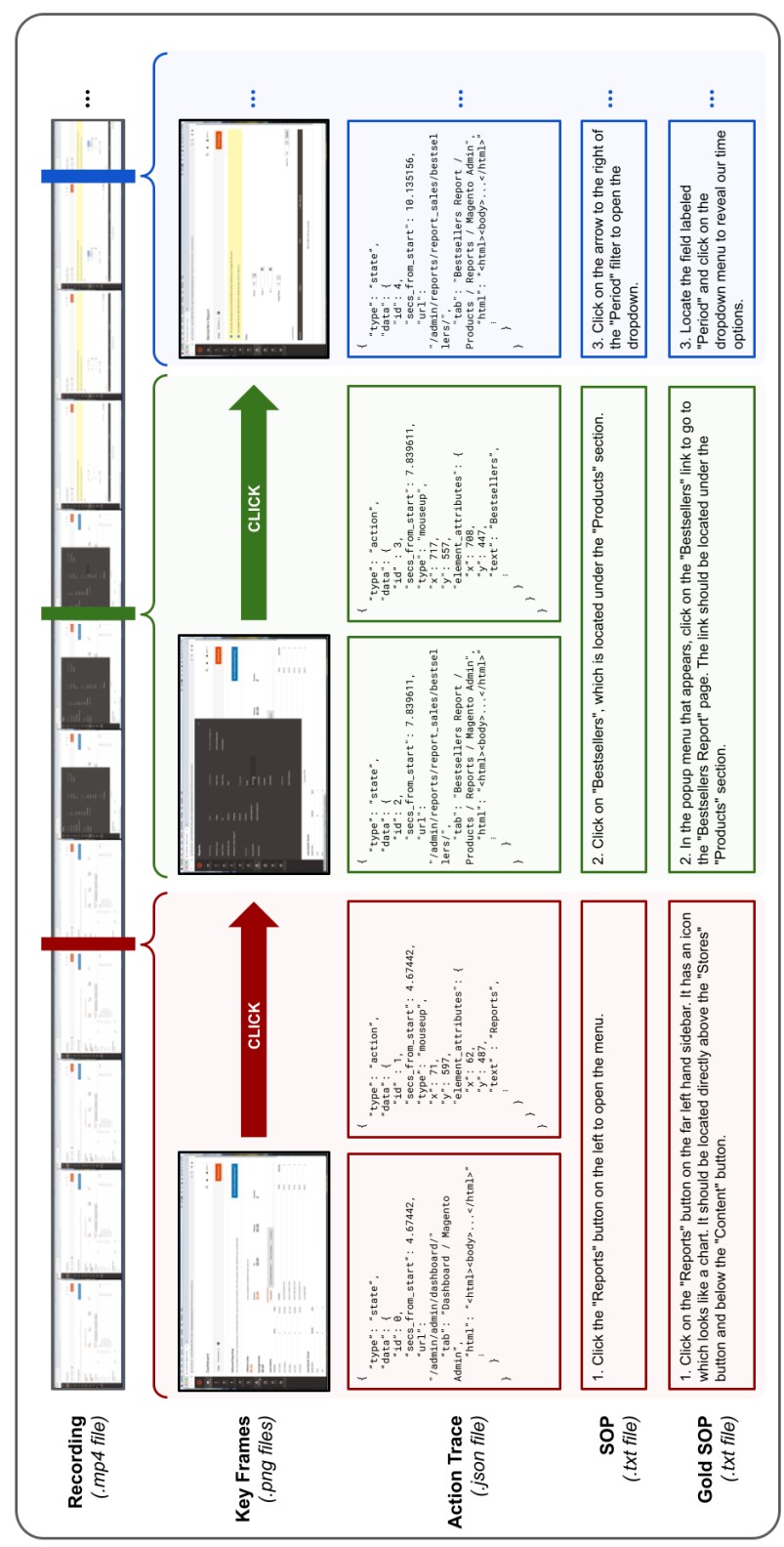

Figure 8: Data collected for each demonstration in **WONDERBREAD**. The example shown here contains the first 3 keyframes and 2 actions from demonstration **"0 @ 2023-12-25-15-10-58"** for solving **Task #0** (*"What is the top-1 best-selling product in 2022?"*)

# B  Benchmark Tasks

For clarity, we define the following notation: Our dataset contains a set of workflow demonstrations $\mathcal{D}$. Each demonstration $d \in \mathcal{D}$ is defined as $d = (I, \text{SOP}, (s_1, a_1, s_2, a_2, ..., a_{n-1}, s_n))$ where $I$ is the "Intent" or the natural language description of the workflow being done, SOP is a manually written step-by-step guide describing the steps taken in the demonstration, $s_i$ is the $i$th state of the webpage, and $a_i$ is the action taken at state $s_i$ (i.e. a 'click', 'keystroke', or 'scroll' event extracted from the trace). We represent each state $s_i$ as a $.png$ image which contains a single frame extracted from the screen recording of the demonstration. We select these frames by first logging the timestamp of every action $a_i$ taken during the demonstration, then iterating through every frame of the screen recording video and extracting the frames corresponding to those action timestamps. We refer to these as "key frames.". There are multiple demonstrations $d$ for each workflow, so $I$ is not unique. However, SOP and $(s_1, a_1, s_2, a_2, ..., a_{n-1}, s_n))$ are unique across different demonstrations.

Table 6: Tasks in **WONDERBREAD**. Here, "Demo" can include some combination of an intent ($I$), a SOP, screenshot key frames of states ($s_1, ..., s_n$), and/or a trace of actions ($a_1, ..., a_{n-1}$) for that demonstration.

| Task | Input | Output | Eval | Multi-modal | Multiple Demos |
|---|---|---|---|---|---|
| **Documentation** | | | | | |
| SOP Generation | 1 Demo | SOP | LLM | ✔ | – |
| Demo Segmentation | 2+ Demos | Clustering | ARI | ✔ | ✔ |
| **Knowledge Transfer** | | | | | |
| Question Answering | Question & 1+ Demos | Free text | LLM | ✔ | ✔ |
| Demo Validation | 1 Demo with SOP | Binary label | F1 | ✔ | – |
| **Improvement** | | | | | |
| Demo Ranking | 3+ Demos | Ranking | Kendall $\tau$ | ✔ | ✔ |
| SOP Improvement | 1 Demo & SOP | SOP | LLM | ✔ | – |

## B.1  Documentation

These subtasks assess a model's ability to generate documentation for existing workflows.

1. **SOP Generation Description:** Given specified components of a workflow demonstration, the model is tasked with generating a new SOP that documents the steps of that workflow. This evaluates a model's ability to generate written documentation.

    **Input:** Given a demonstration $d = (I, \text{SOP}, (s_1, a_1, s_2, a_2, ..., a_{n-1}, s_n))$, we provide the model with either $(I)$, $(I, (s_1, ..., s_n))$, or $(I, (s_1, a_1, ..., a_{n-1}, s_n))$. In our Results Table 3, these correspond to rows with one checkmark under the "Intent" column, two checkmarks under the "Intent" and "Keyframes" columns, and three checkmarks under the "Intent", "Keyframes", and "Trace" columns, respectively.
    **Output:** An new SOP denoted as $s'$ describing the steps of demonstration $d$.
    **Evaluation:** Pairwise per-line comparison between $s$ and $s'$ that determines the precision and recall as described in Appendix Section C.2

2. **Demonstration Segmentation** Given multiple demonstrations from separate workflows concatenated into a single sequence, identify when each demonstration starts and ends. This evaluates the model's ability to disambiguate between different workflows occurring in sequence.

    **Input:** A concatenated sequence of $k$ demonstrations $\{d^i\}_{i=1}^k$, represented as either $(s_1^1, ..., s_n^1 || ... || s_1^k, ..., s_n^k)$   or   $(s_1^1, a_1^1, ..., a_{n-1}^1, s_n^1 || ... || s_1^k, a_1^k, ..., a_{n-1}^k, s_n^k)$.
    **Output:** For each frame $s$ in the provided input, assign each of the frames to one of the $k$ demonstrations. This generates a clustering that maps frames to demonstrations. For example, given 20 frames from three demonstrations ($A,B,C$), an output assignment clustering might map frames 1-5 to demonstration $A$, frames 6-10 to demonstration $C$, and frames 11-20 to demonstration $B$.
    **Evaluation:** Given the $k$ clusters of frames, measure the adjusted rand score.

## B.2 Knowledge Transfer

These subtasks assess a model's ability to apply knowledge of workflows in practical scenarios.

1. **Question Answering -** Given a question about one or more workflow demonstrations, generate a natural language answer.

   **Input:** A brief question (instantiated from one of the templates in Appendix A.5), and one or two demonstrations, where each demonstration is represented as either $(SOP)$ or $(s_1, a_1, ..., a_{n-1}, s_n)$.

   **Output:** A natural language answer to the question.

   **Evaluation:** Using GPT-4-as-a-judge, compare a human-written reference answer to the generated answer and determine a score for specified criteria on a scale from 1 (bad) to 3 (good). Specified criterion include **completeness** (the response fully answers the question), **soundness** (the response is logically consistent), **clarity** (the response is unambiguous), and **compactness** (the response is concise).

2. **Demonstration Validation -** Given a demonstration and SOP, determine whether (a) the workflow was successfully completed; and (b) whether the demonstration exactly followed the steps of the SOP. For (b), it is not sufficient to merely complete the workflow, but the steps taken to complete it must align with its corresponding SOP.

   **Input:** For (a) we create "positive" examples by sampling full sequences of $(s_1, a_1, ..., a_{n-1}, s_n)$ from our dataset, and create "negatives" by truncating some sequences by a random number of frames to get $(s_1, a_1, ..., s_{k-1}, s_k)$ where $k < n$. Given this sequence, we prompt the model to provide a binary assessment of whether the workflow was completed or not. For (b), we create "positives" by sampling full sequences $(s_1, a_1, ..., a_{n-1}, s_n)$ from our dataset, then and either (a) randomly shuffle or (b) randomly delete frames from this sequence to generate "negative" examples. We prompt the model with this sequence and the SOP, and have it output a binary assessment of whether the sequence exactly followed the SOP.

   **Output:** For (a), a binary assessment of whether the given sequence was truncated. For (b), a binary assessment of whether the given sequence exactly followed the steps of its associated SOP.

   **Evaluation:** Binary classification metrics (ie. Accuracy, F1-Score).

## B.3 Improvement

These subtasks evaluate a model's capacity to improve a given workflow's efficiency.

1. **SOP Ranking -** Given a set of SOPs written by different human annotators for the same workflow, rank the SOPs in order of quality.

   **Input:** A set of $k$ SOPS $\{SOP^i\}_{i=1}^k$ written by different annotators for the same workflow.

   **Output:** A ranking of the quality of the SOPs from $1...k$, where 1 is best and $k$ is worst.

   **Evaluation:** Given a provided ground truth ranking, determine the Spearman correlation and Kendall's Tau between the predicted ranking and the ground truth.

2. **SOP Improvement -** Given a demonstration and low-quality SOP, and a rubric, generate an improved SOP that better captures what is shown in the demonstration.

   **Input:** One demonstration $d^1$, a low quality $SOP^1$ generated by a human and an SOP generation rubric $r$.

   **Output:** An improved $SOP^{1'}$ that better aligns with the provided rubric.

   **Evaluation:** LLM-based evaluation, where the model generates a rating of 1.0 - 5.0 conditioned upon a rubric.

# C    Evaluation

## C.1    Compute

We rely on the publicly available APIs for each of the multimodal FMs we benchmark in this report: GPT-4, Claude3 Sonnet, and Gemini Pro. Thus, we did not require any GPUs to run our benchmark. In terms of cost, the Gemini Pro 1 API was free to use, the Claude 3 API cost roughly $400 in credits, and the GPT-4 API cost roughly $1,000 in credits.

## C.2    LLM-Based Evaluation

### SOP Generation
The automated evaluation for the *SOP Generation* task utilized a pairwise per-step comparison operating over the generated new SOP and the reference high quality SOP. Through a series of iterative prompts, GPT-4 was tasked to identify if the intention of a step in the new SOP was encapsulated in any step of the reference SOP and vice versa. The record of which steps were not included in the alternative SOP were then utilized to calculate the per-step precision, recall, and F1-score.

The precision (P), recall (R), and F1-score (F1) are calculated as follows:

$$P = \frac{TP}{TP + FP}$$

$$R = \frac{TP}{TP + FN}$$

$$F1 = 2 \times \frac{P \times R}{P + R}$$

Where:

- $TP$ (True Positives) is the number of steps in the new SOP that correctly map steps in the reference SOP.
- $FP$ (False Positives) is the number of steps in the new SOP that do not map to any step in the reference SOP.
- $FN$ (False Negatives) is the number of steps in the reference SOP that do not map to any step in the new SOP.

For the *SOP Generation* task, we found that our LLM-based evaluator was able to achieve high correlation out of the box with human raters as shown in Appendix Table 9. We hypothesize that this is because the *SOP Generation* evaluation task is set up to only require the model to make a binary decision over an atomic fact, rather than assess the quality of an open-ended question as in the *Question Answering* task, as seen in other works on LLM-based evaluations [40].

### Question Answering
We rate each answer on a scale from 1 (bad) to 3 (good) on the following four criteria: **completeness** (the response fully answers the question), **soundness** (the response is logically consistent), **clarity** (the response is unambiguous), and **compactness** (the response is concise). Our original LLM-based evaluators had low correlation with human raters – an average Pearson correlation of 0.56 for scoring free reponses questions on a scale of 1 (low quality) to 3 (high) across the four axes of soundness, completeness, clarity, and compactness. We noticed that GPT-4 tended to be overly generous in its ratings. Adding a 3-shot example to our evaluation prompt (one for each possible score) and refining the prompt to "score harsher" helped increase the average correlation with human raters by 54% (to 0.86), as shown in Appendix Table 8.

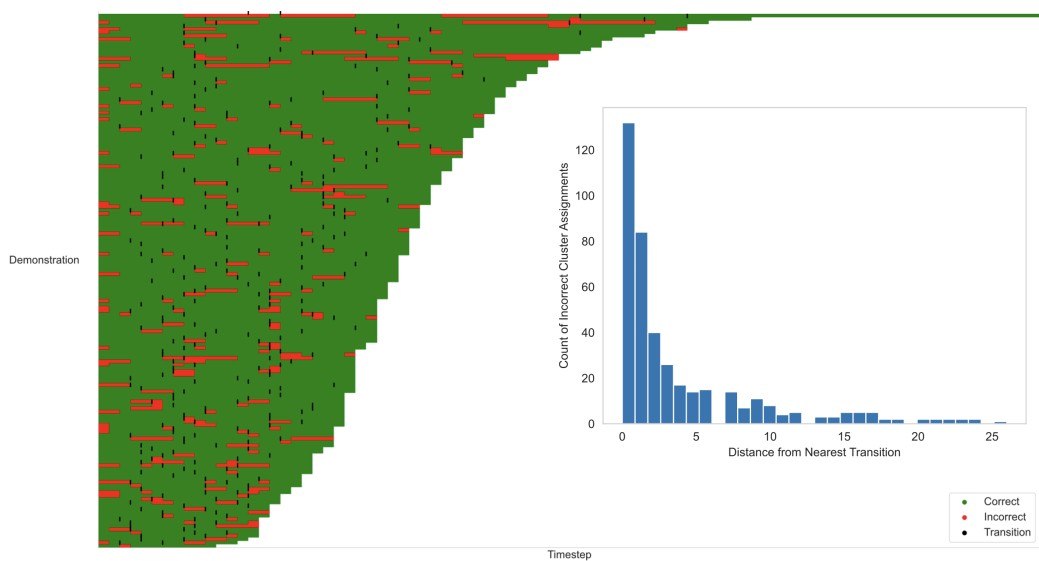

Figure 9: **Demo Segmentation:** Results from GPT-4 evaluated on $k = 3$ concatenated workflows when provided the workflow intent, SOP, and keyframes. (Outset) Each row represents a concatenated sequence of frames from 4 demonstrations. Green line segments are frames that were classified as belonging to the correct task. Red segments are incorrectly classified frames. Black markers indicate a transition between tasks in the ground truth sequence. (Inset) The distribution of distances between each incorrect frame prediction and its closest transition point. Its heavy right skew indicates that the transitions between workflows are where most errors occur, but that GPT-4 is typically able to recover within 3 frames into a workflow.

# D  Additional Results

Table 7: **Knowledge Transfer:** Average scores across all 4 evaluation axes for question answering.

| Model | Completeness | Soundness | Clarity | Compactness | Average Score |
|---|---|---|---|---|---|
| Claude3 Sonnet | 1.56 | 1.83 | 2.18 | 2.61 | 2.05 |
| Gemini Pro 1 | 1.81 | 2.15 | 2.83 | 2.95 | 2.44 |
| GPT-4 | 2.20 | 2.51 | 2.96 | 2.85 | 2.63 |
| Human | 3.00 | 3.00 | 2.64 | 2.88 | 2.88 |

Table 8: **Knowledge Transfer:** Correlation between GPT-4 and human-based evaluation based on 60 randomly sampled question-answer pairs.

| Criteria | Pearson Corr. | Pearson p-value | Spearman Corr. | Spearman p-value |
|---|---|---|---|---|
| Completeness | 0.84 | 5.38e-09 | 0.86 | 1.12e-09 |
| Soundness | 0.92 | 1.51e-12 | 0.88 | 2.34e-10 |
| Clarity | 0.80 | 1.01e-07 | 0.80 | 1.01e-07 |
| Compactness | 0.89 | 2.07e-13 | 0.89 | 7.41e-11 |

Table 9: **SOP Generation:** Correlation between GPT-4 and human-based evaluation of the precision/recall of generated SOPs based on 30 randomly sampled examples.

| Criteria | Pearson Corr. | Pearson p-value | Spearman Corr. | Spearman p-value |
|---|---|---|---|---|
| Precision | 0.84 | 4.63e-09 | 0.85 | 2.80e-09 |
| Recall | 0.88 | 1.63e-10 | 0.82 | 3.97e-08 |

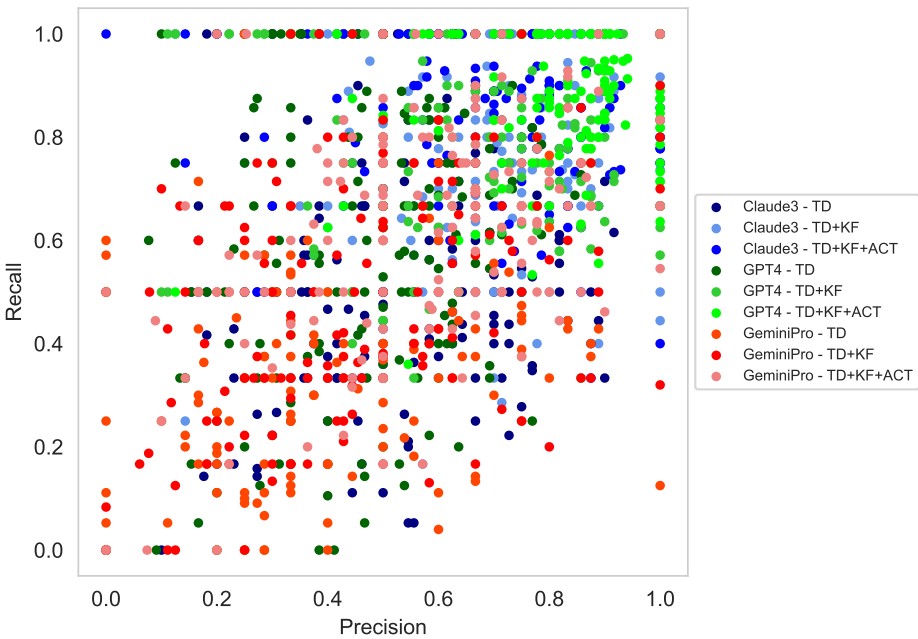

Figure 10: **SOP Generation:** Each point is an individual SOP. Higher and to the right is better. GPT-4 tends to excel at identifying all steps in a demonstration (i.e. higher recall) but hallucinates inaccurate or superfluous steps (i.e. lower precision).

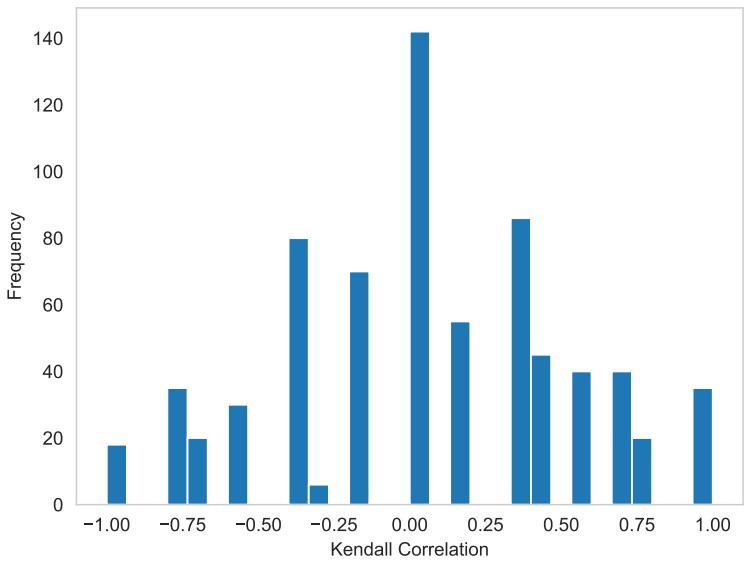

Figure 11: **SOP Ranking:** Ranking demos based solely on SOPs is essentially random

| Workflow Intent | GPT-4 response | Claude3 response | Ground truth |
|---|---|---|---|
| What is the top-1 best-selling product in 2022 | 1. Click on the text labeled 'REPORTS' on the left-hand side menu to open the reports section. 2. In the reports menu that appears, click on the text labeled 'Bestsellers' under the 'Sales' category. 3. On the 'Bestsellers Report' page, click on the dropdown labeled 'Day Month Year' to select the time period for the report. 4. Select 'Year' from the dropdown options to filter the report by year. 5. In the 'From' field, click on the text field labeled 'None' and type '01/01/2022' to set the start date for the report. 6. In the 'To' field, click on the text field labeled 'None' and type '12/31/2022' to set the end date for the report. 7. Click on the orange button labeled 'Show Report' to generate the bestsellers report for the year 2022. 8. The report will now display the list of bestselling products for the year 2022, along with their respective prices and order quantities. | 1. Navigate to the ""Generic e-commerce admin based on Adobe Magneto"" web application and log in if necessary. Once logged in, you will be directed to the Dashboard page. 2. On the left-hand side menu, locate and click on the 'REPORTS' section. This is a vertical menu bar, and 'REPORTS' is one of the options listed. 3. A dropdown menu will appear. Move the mouse cursor over the 'Products' section within the dropdown, then click on 'Bestsellers' which is a sub-option listed under 'Products'. 4. You will be taken to the 'Bestsellers Report' page. Here, locate the 'Period' dropdown menu, which is part of the 'Filter' section. Click on the dropdown that currently displays 'Day' by default. 5. In the dropdown menu that appears, click on 'Year' to change the period to an annual view. 6. Move to the 'From' date field, which has a calendar icon next to it. Click on the text field and type in '01/01/2022' to set the start date of the report to the beginning of the year 2022. 7. Next, move to the 'To' date field, which also has a calendar icon next to it. Click on the text field and type in '12/31/2022' to set the end date of the report to the end of the year 2022. 8. After setting the date range, locate and click on the 'Show Report' button, which is to the right of the date fields. This will generate the report based on the specified date range. 9. Review the generated report below the 'Filter' section. The report will display a list of products sold within the specified period, along with their respective prices and order quantities. The top-1 best-selling product of the year 2022 will be the first item listed in the 'Product' column with the highest 'Order Quantity'. | 1. Click on the "Reports" button on the far lefthand sidebar. It has an icon which looks like a chart. It should be located directly above the "Stores" button and below the "Content" button. 2. In the popup menu that appears, click on the "Bestsellers" link to go to the "Bestsellers Report" page. The link should be located under the "Products" section. 3. Locate the field labeled "Period" and click on the dropdown menu to reveal our time options. 4. Click on the "Year" option to set the reporting period to Year. 5. Click on the "From" textbox to focus it. It should be located directly underneath the "Period" field. 6. Type in the first day of our desired time period, which in this case is "01/01/2022" 7. Click on the "To" textbox to focus it. It should be located directly underneath the "From" field. 8. Type in the last day of our desired time period, which in this case is "12/31/2022" 9. Click on the orange "Show Report" button, which can be found on the top right of the page, in order to generate our best-selling product report. 10. The best-selling products will appear in a table at the bottom of the page. Scroll down to reveal this table. The results should be sorted in descending order by Order Quantity, so the top-1 product will simply be the first row in the results. |

(a) In the examples above, both models (middle columns) generate an almost entirely correct SOP (far right column) for the given workflow (far left column). Minor mistakes are marked in red. GPT4 incorrectly mentions to click on "Bestsellers" under the "Sales" category, but this tab is actually under the "Product" category. Claude3 mentions to "Navigate to the generic e-commerce admin based on Adobe Magneto", which is incorrect since the workflow starts with the user already on this webpage, and thus there is no need to do any navigation. This was a common failure mode noted for models – while including this step is not deleterious, it is not strictly correct. Otherwise, both models generate a correct SOP.

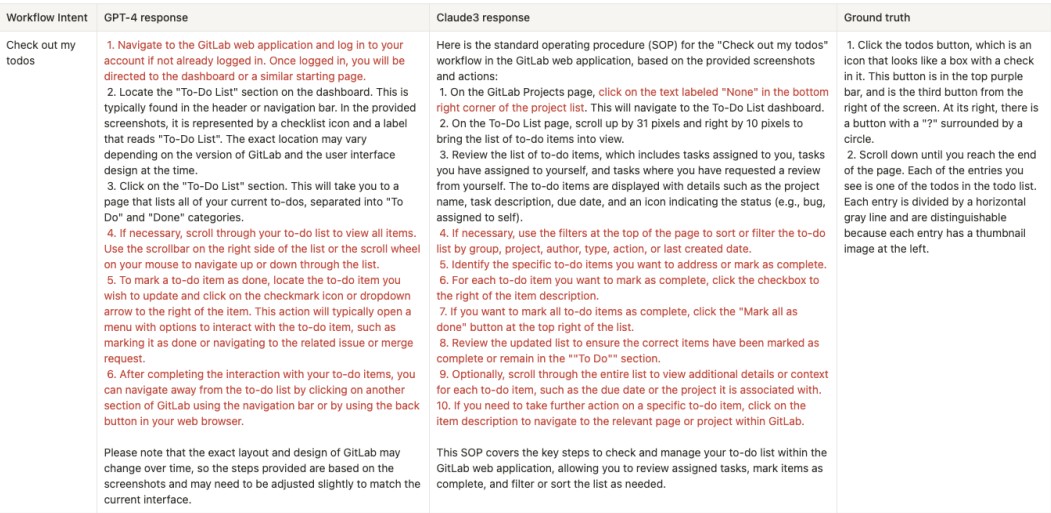

(b) In the examples above, both models (middle columns) make significant mistakes in generating SOPs relative to the ground truth (far right column). GPT4 provides extra information on how to mark the todo items as complete and recommends navigating away from the page, which is not a part of the workflow as demonstrated by the user. Claude3 misinterprets the task and assumes that the user wants to mark todo items as complete and gives instructions for that, which is incorrect. Claude3 also provides an incorrect instruction to navigate to the todo list.

Figure 12: **Qualitative Analysis of SOP Generation Task:** Above, we provide success (top) and failure cases (bottom) for GPT4 and Claude3 on the SOP generation task where the model is provided with the task description, key frames, and action trace.

| Question | Evidence | GPT-4 response | Claude3 response | Ground truth |
|---|---|---|---|---|
| What if instead of searching for "switch accessories" we wanted to search for "womens basketball". How would you change this workflow to accomplish that? | Trace+Screens hots | To search for "womens basketball" instead of "switch accessories," you would type "womens basketball" into the text field labeled 'combobox' and then press the 'Enter' key. | Simply replace "switch accessories" with "womens basketball" in the search query. The rest of the workflow remains the same. | Instead of typing "switch accessories" in the search bar simply type "womens basketball." |

(a) In the examples above, both models (middle columns) are able to successfully answer a question (far left column) about modifying a demonstration to achieve a slightly different goal. The models are provided key frame screenshots of the demonstration as well as an action trace. The ground truth answer is shown in the far right column.

| Question | Evidence | GPT-4 response | Claude3 response | Ground truth |
|---|---|---|---|---|
| Explain what the most common failure modes might be for a user performing this task. | SOP | The most common failure modes could include not finding the "My Account" or "My Orders" buttons due to a website layout change or misunderstanding the instructions on how to identify fulfilled orders and calculate the total amount spent. | Common failure modes include missing or overlooking UI elements like the "My Account" button or "My Orders" sidebar, and misunderstanding date ranges for calculating order totals over the past month. | One common failure mode could be not including all fulfilled orders from the past month if they are spread beyond just the first page of results. Another common failure mode is incorrectly including all orders from the first page when only orders from the past month are to be considered. |

(b) In the examples above, neither model (middle columns) answers the question (far left column) as expected in the ground truth answer (far right column). The models are expected to answer the most common error modes, however, both models give error modes that are relatively unlikely to happen for a human, as the UI elements are easy to find and less likely to be the cause for error than making a higher-level reasoning error as in the ground truth answer.

Figure 13: **Qualitative Analysis of Question Answering Task:** Above, we provide success (top) and failure cases (bottom) for GPT4 and Claude3 on the Question Answering task.

## D.1 Overall Dataset Stats

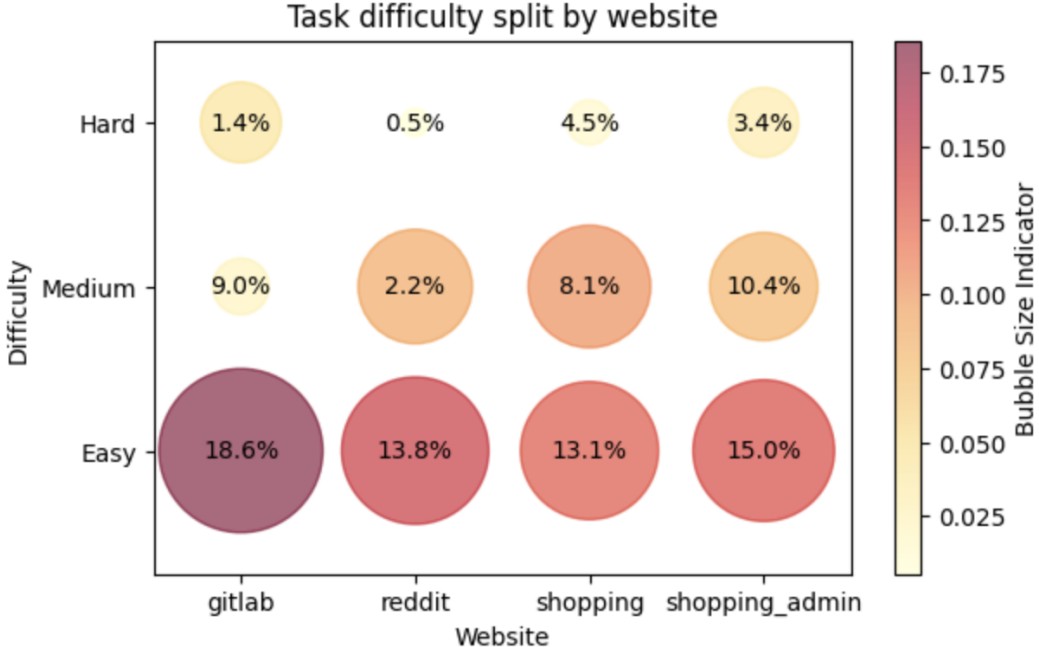

Figure 14: Distribution of task difficulty across websites.

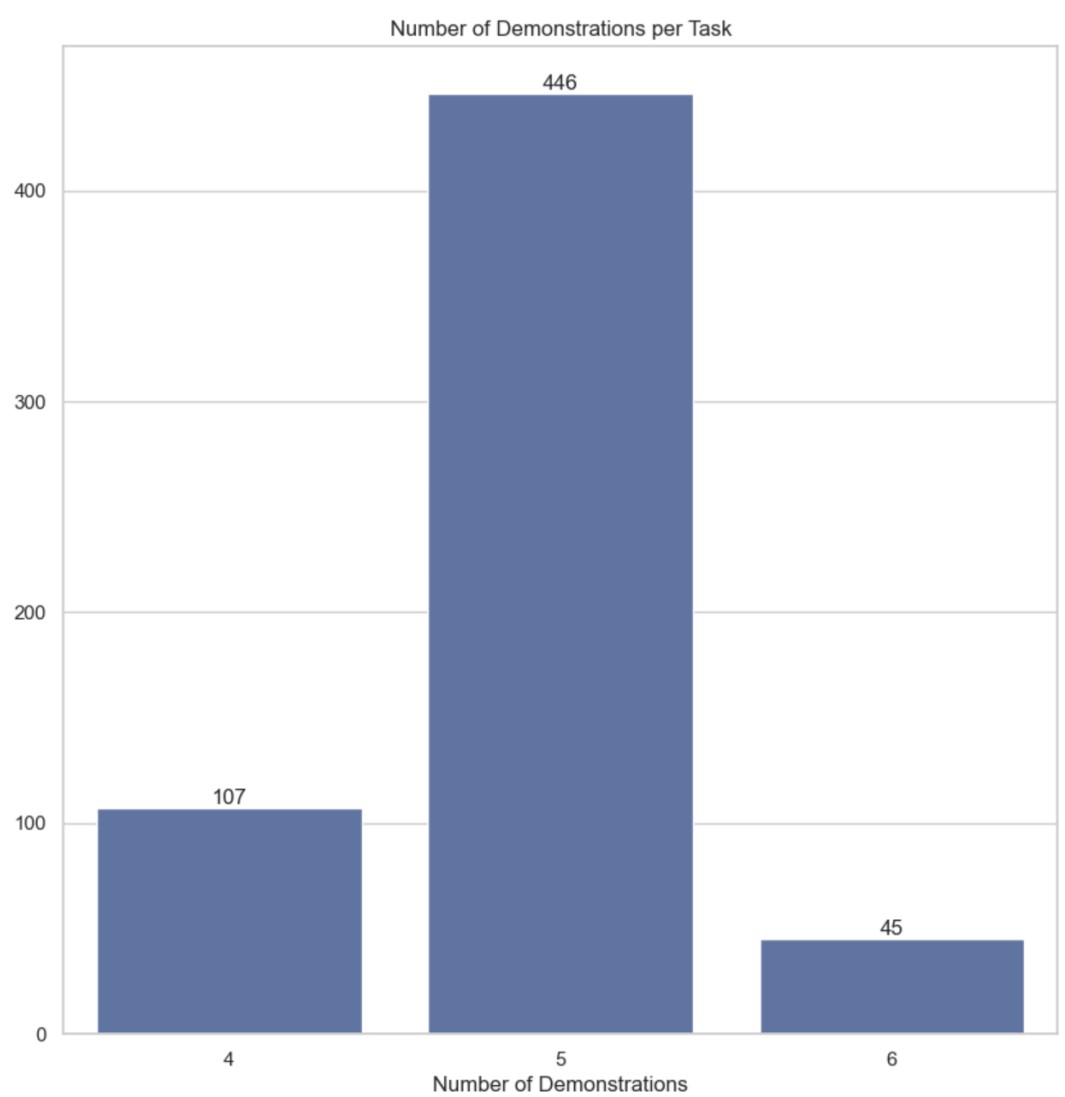

Figure 15: Number of demonstrations per task

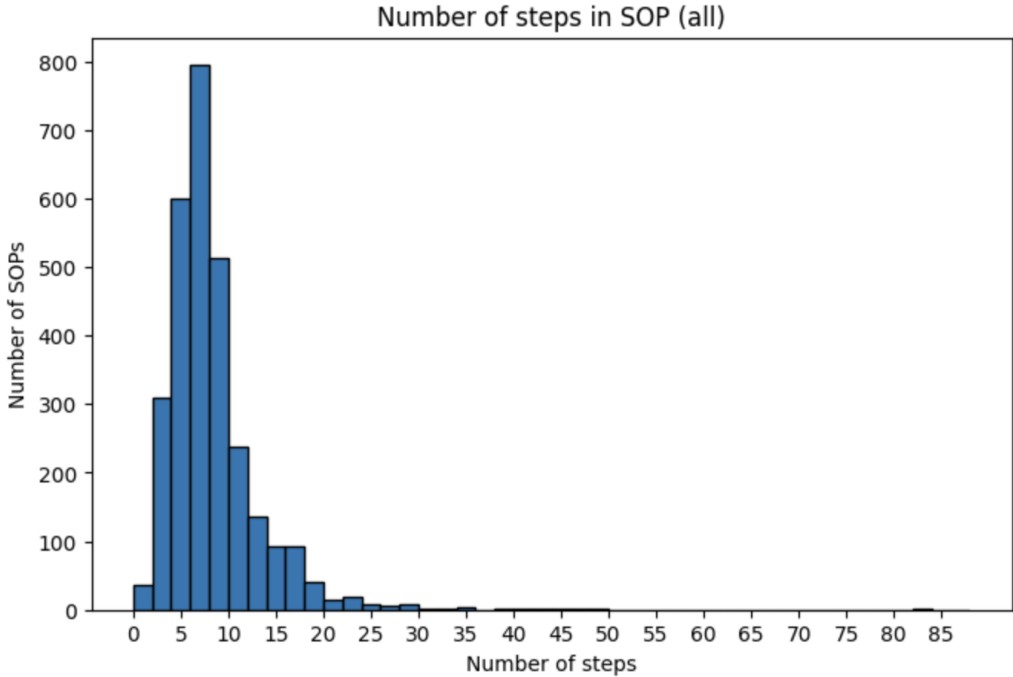

Figure 16: Number of steps in SOP per demonstration

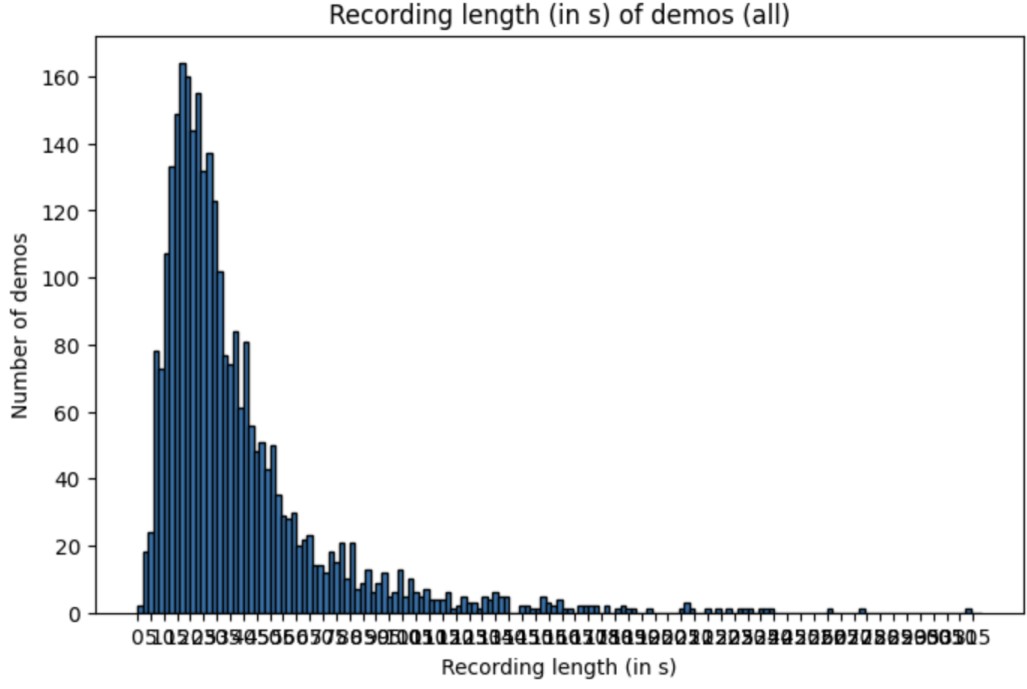

Figure 17: Length of video recording (in seconds) per demonstration

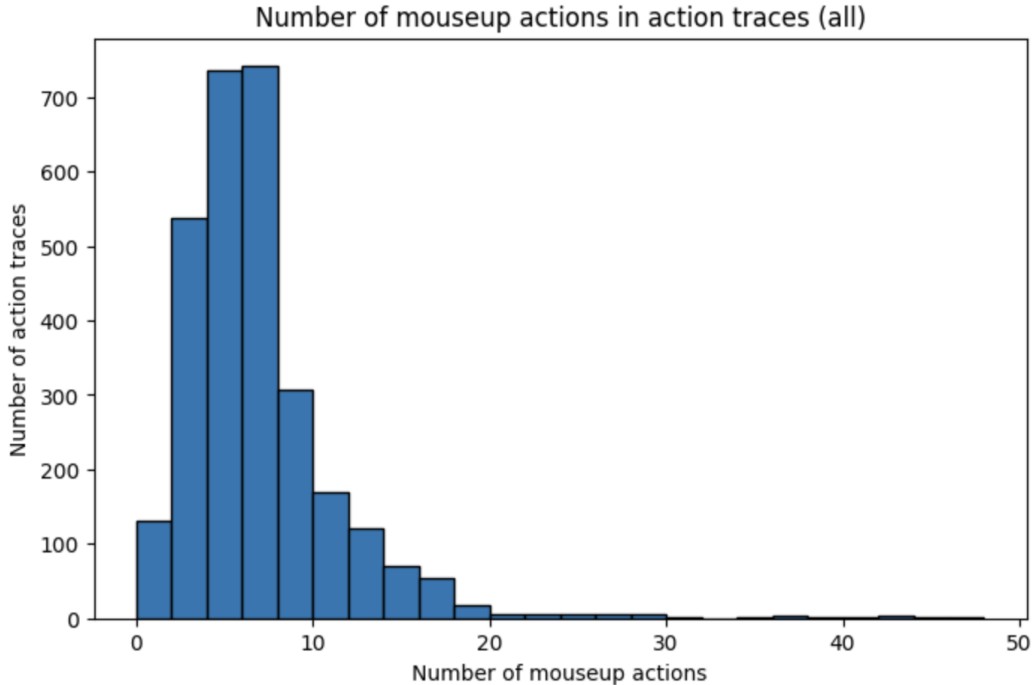

Figure 18: Number of clicks per demonstration

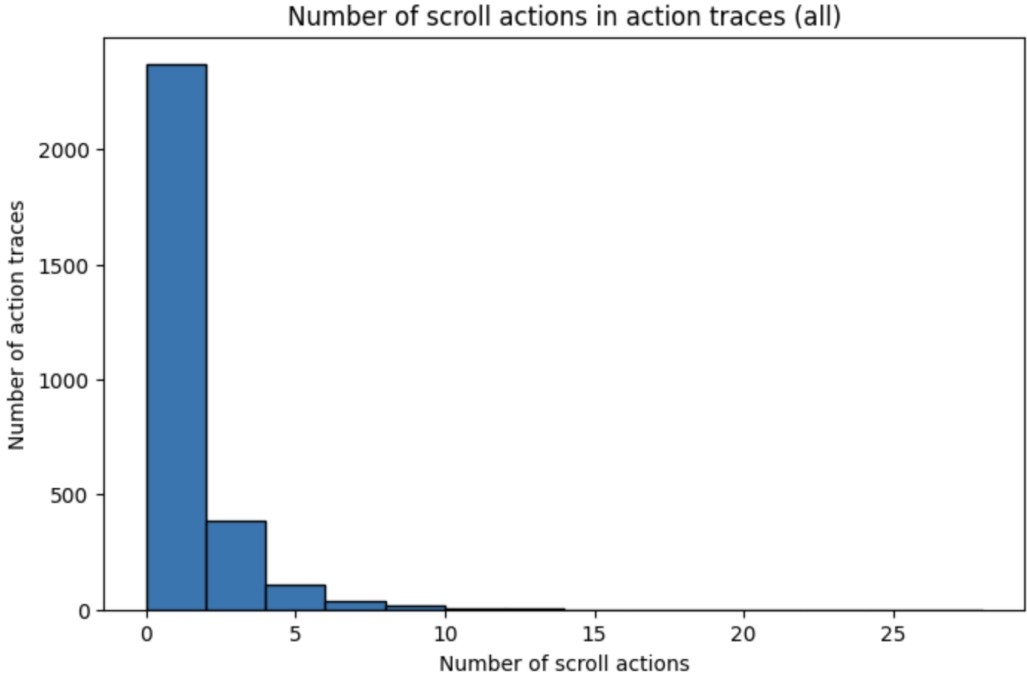

Figure 19: Number of scrolls per demonstration

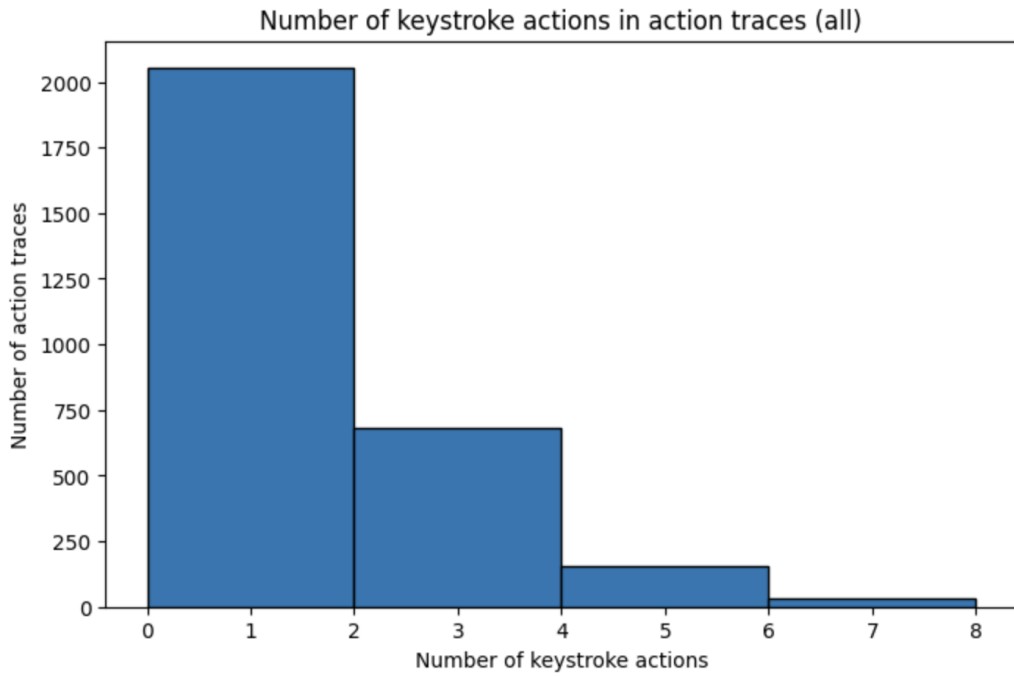

Figure 20: Number of keystrokes per demonstration

## D.2 Dataset Stats, Split By Difficulty

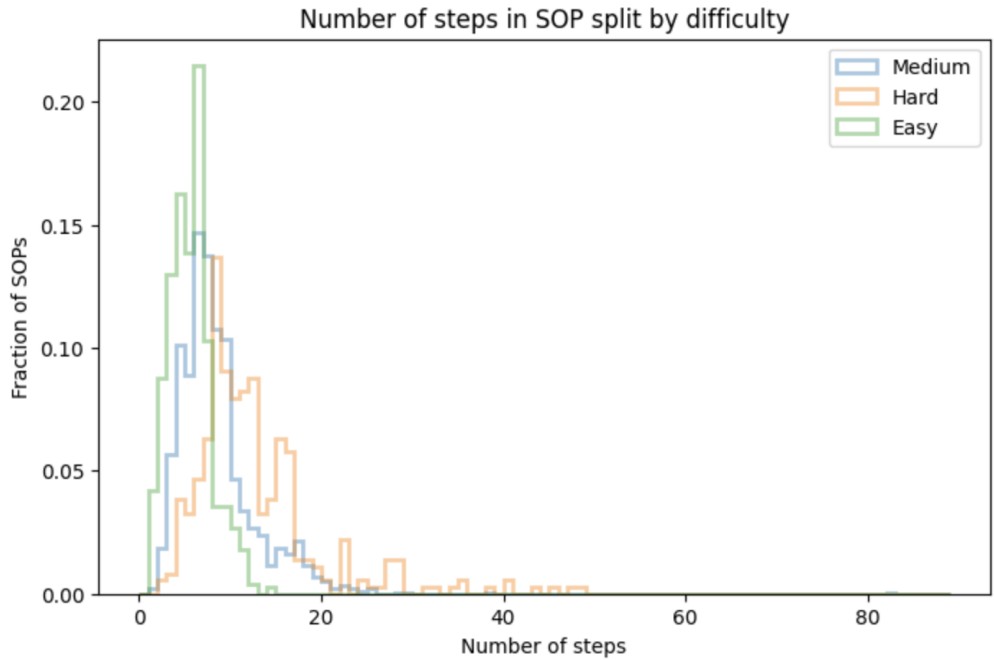

Figure 21: Number of steps per SOP per demonstration, split by task difficulty

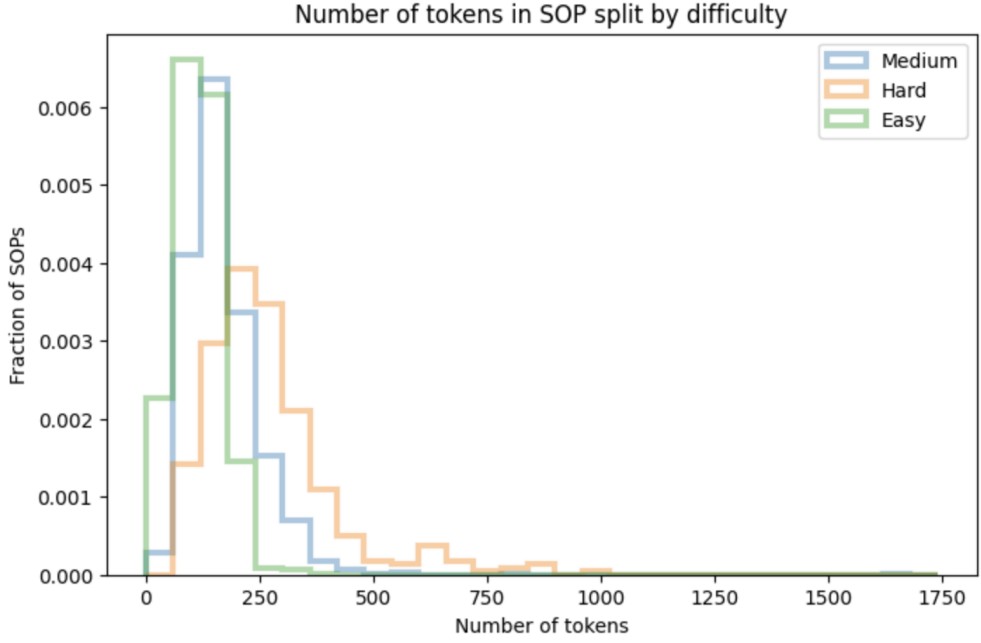

Figure 22: Number of tokens per SOP per demonstration, split by task difficulty

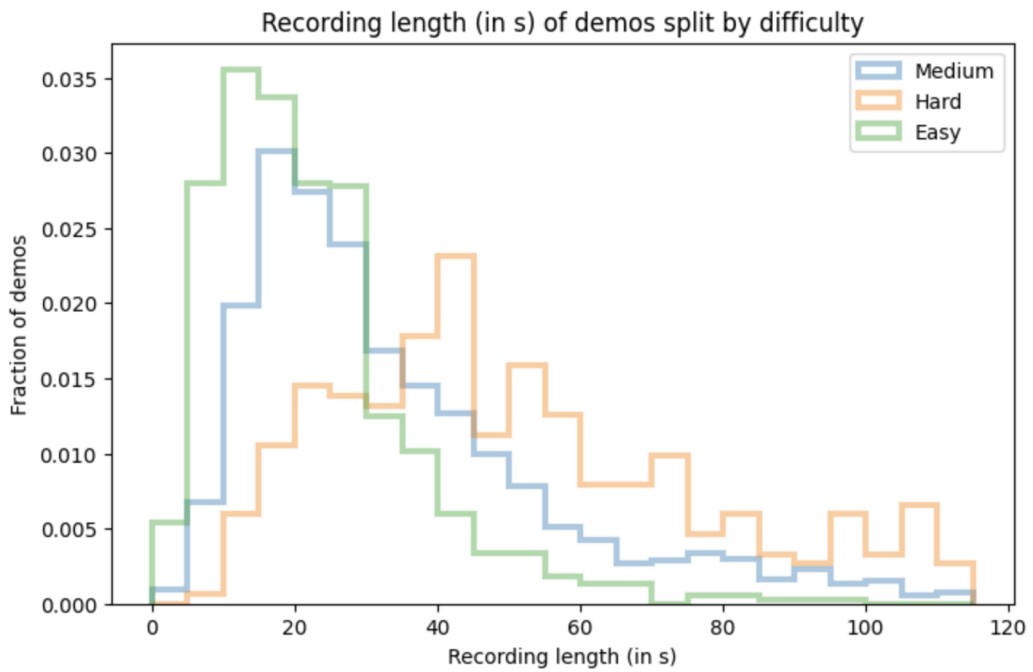

Figure 23: Length of video recording (in seconds) per demonstration, split by task difficulty

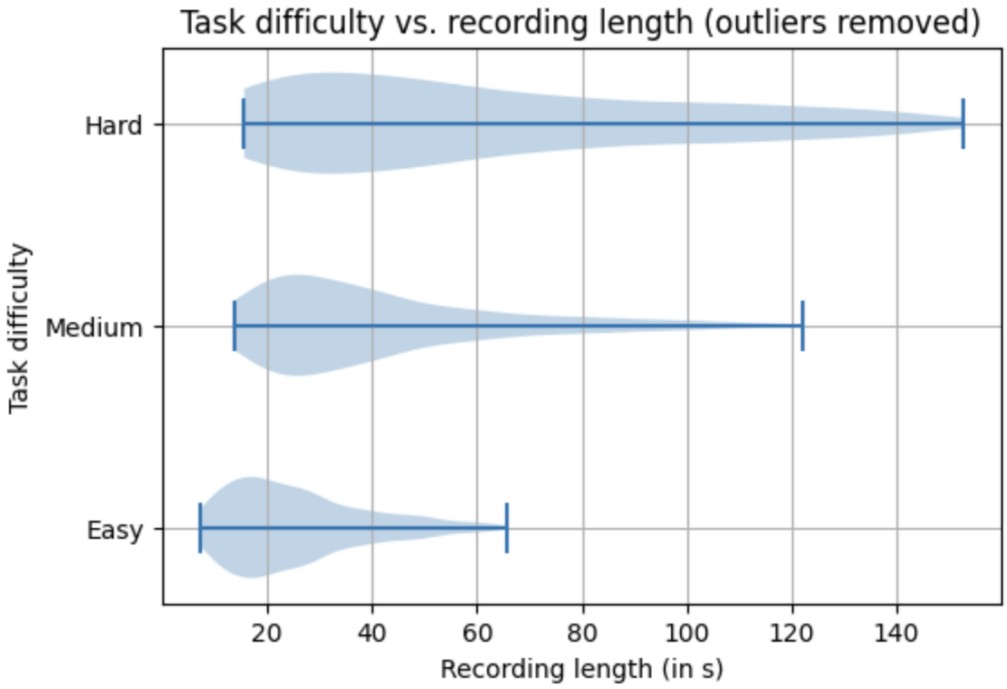

Figure 24: Length of video recording (in seconds) per demonstration, split by task difficulty

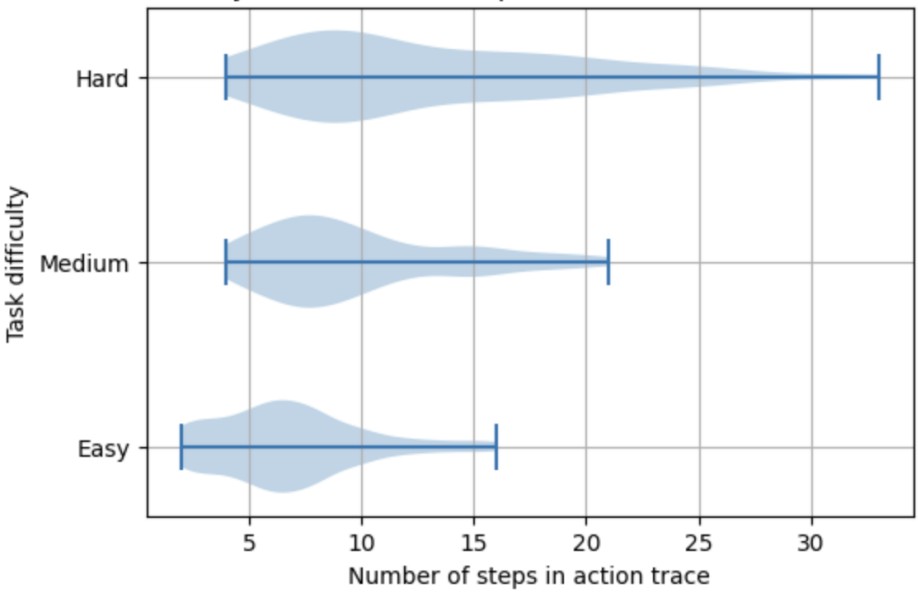
Figure 25: Number of actions per demonstration, split by task difficulty

| Difficulty | Min | Median | Max |
|------------|-----|--------|-----|
| Medium | 1 | 7 | 82 |
| Hard | 2 | 10 | 48 |
| Easy | 1 | 5 | 14 |

Table 10: Number of steps per SOP, split by task difficulty

| Difficulty | Min | Median | Max |
|------------|-----|--------|-----|
| Medium | 12 | 154 | 1631 |
| Hard | 62 | 240 | 976 |
| Easy | 18 | 114 | 382 |

Table 11: Number of tokens per SOP, split by task difficulty

### D.3 Dataset Stats, Split By Website

| Website | Min | Median | Max |
|---|---|---|---|
| shopping_admin | 30 | 163 | 704 |
| gitlab | 12 | 151 | 870 |
| shopping | 18 | 121 | 1631 |
| reddit | 43 | 148 | 382 |

Table 12: Number of tokens per SOP, split by website

| Website | Min | Median | Max |
|---|---|---|---|
| shopping_admin | 0 | 6 | 29 |
| gitlab | 0 | 6 | 44 |
| shopping | 1 | 4 | 47 |
| reddit | 2 | 6 | 23 |

Table 13: Number of mouseups per demonstration, split by website

| Website | Min | Median | Max |
|---|---|---|---|
| shopping_admin | 0 | 1 | 8 |
| gitlab | 0 | 1 | 7 |
| shopping | 0 | 0 | 7 |
| reddit | 0 | 1 | 8 |

Table 14: Number of keystrokes per demonstration, split by website

| Website | Min | Median | Max |
|---|---|---|---|
| shopping_admin | 0 | 0 | 6 |
| gitlab | 0 | 0 | 7 |
| shopping | 0 | 0 | 3 |
| reddit | 0 | 0 | 1 |

Table 15: Number of keypresses per demonstration, split by website

| Website | Min | Median | Max |
|---|---|---|---|
| shopping_admin | 0 | 1 | 13 |
| gitlab | 0 | 0 | 9 |
| shopping | 0 | 1 | 28 |
| reddit | 0 | 0 | 5 |

Table 16: Number of scrolls per demonstration, split by website

# E    Instructions for Annotators

The figures below contain the instructions and other training provided to the annotators.

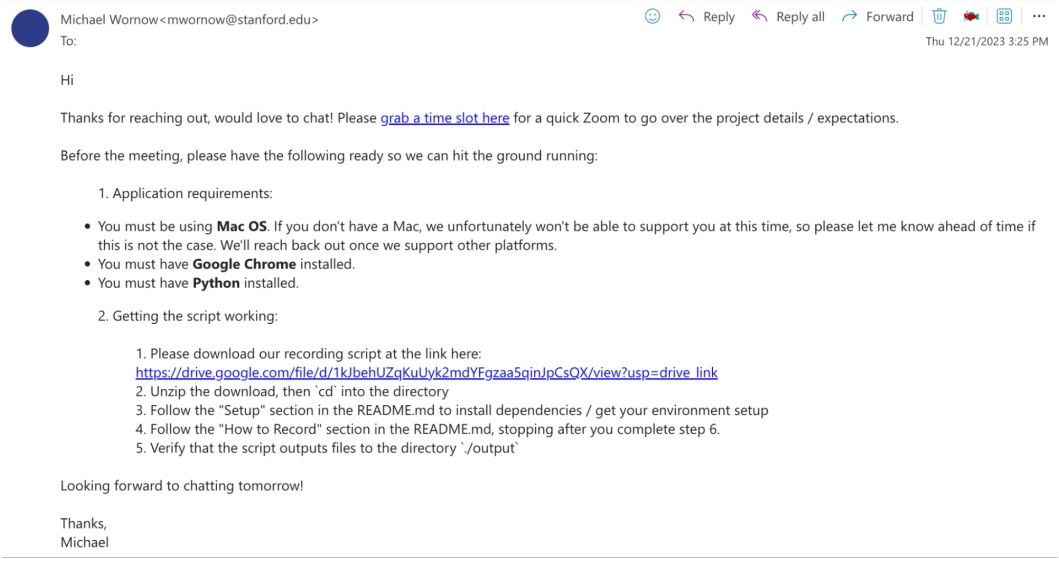

Figure 26: This is the initial onboarding email that asks annotators to set up an online meeting for training.

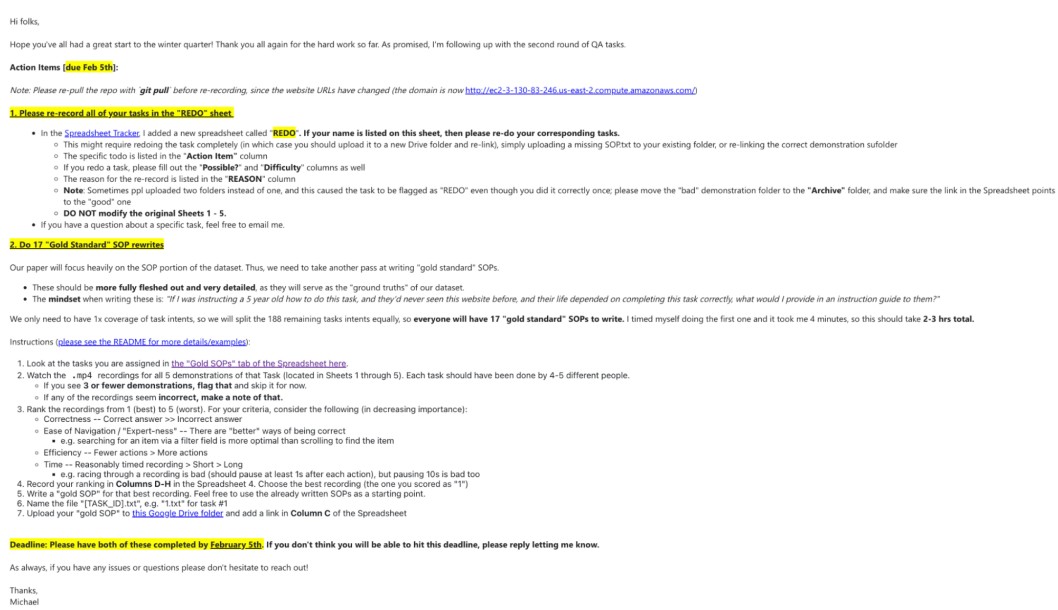

Figure 27: This email provides annotators instructions about how to re-record demonstrations after quality checks. Additionally, there are instructions to write Gold SOPs.

## Few Shot Demonstration Collection

**Goal:** Create human demonstrations for the tasks in WebArena.

**Principles:**

- We are simulating **experts**
    - Do the optimal (i.e. most direct) way to complete each task
    - No wasted clicks / typing.
    - No mistakes -- If you mis-step, then stop recording and re-record from scratch. You might need a couple rehearsals before you record for real.
- We want a **clean** dataset
    - When you record, **only have Google Chrome visible**.
        - Do not show any other Applications
        - No personal information
        - No messy desktop background
    - We recommend having Google Chrome in one monitor, and the rest of your stuff in another monitor.
- We want an **accurate** dataset, even though our **record.py** script has a slight lag.
    - Make sure that **record.py** has finished logging your action before you move onto the next action.
        - **Clicks** - usually no lag, so no need to worry
        - **Keystrokes** - usually no lag, so no need to worry
        - **Scrolls** - lags often b/c each 1px of scroll = 1 action, so make sure the script has "caught up" before you move onto the next action

**Disclaimers:**

- This only works on Mac currently.
- Please email mwornow@stanford.edu if you have any issues / questions / errors.

We will be publishing these recordings, so please make sure any personal information is hidden from view when recording. By participating in this project, you consent to having your recordings published.

## 📋 Tasks

All of the tasks are stored in `./tasks/` .

## 🔧 Setup

1. Enable the following Mac permissions:

    a. `System Preferences > Privacy & Security > Accessibility` , make sure VSCode and Terminal are enabled.

    b. `System Preferences > Privacy & Security > Screen Recording` , make sure VSCode and Terminal are enabled.

    c. `System Preferences > Privacy & Security > Input Monitoring` , make sure VSCode and Terminal are enabled.

2. Install **ffmpeg** with: `brew install ffmpeg`

3. Download this repo:

```
# Install repo
git clone https://github.com/Miking98/demonstration-collection.git
cd demonstration-collection/

# Create conda env + install dependencies
conda create -n demo_env python=3.10 -y
conda activate demo_env
pip3 install -r requirements.txt
pip3 install -e .
```

## ⚡ Quickstart

In a background terminal:

```
alias google-chrome="/Applications/Google\ Chrome.app/Contents/MacOS/Google\ Chrome"
google-chrome --remote-debugging-port=9222 --user-data-dir="/tmp/chrome_temp"
```

In another terminal:

```
conda activate demo_env
python record.py --is_webarena --name <TASK_ID>
```

Hit `Esc` to end the recording.

Then...

1. Fill out `SOP.txt`
2. Upload folder to the Google Drive
3. Fill out your task's row in the Google Spreadsheet tracker.

Figure 28: This screenshot provides instructions on how to get the environment and technology setup before recording demonstrations.

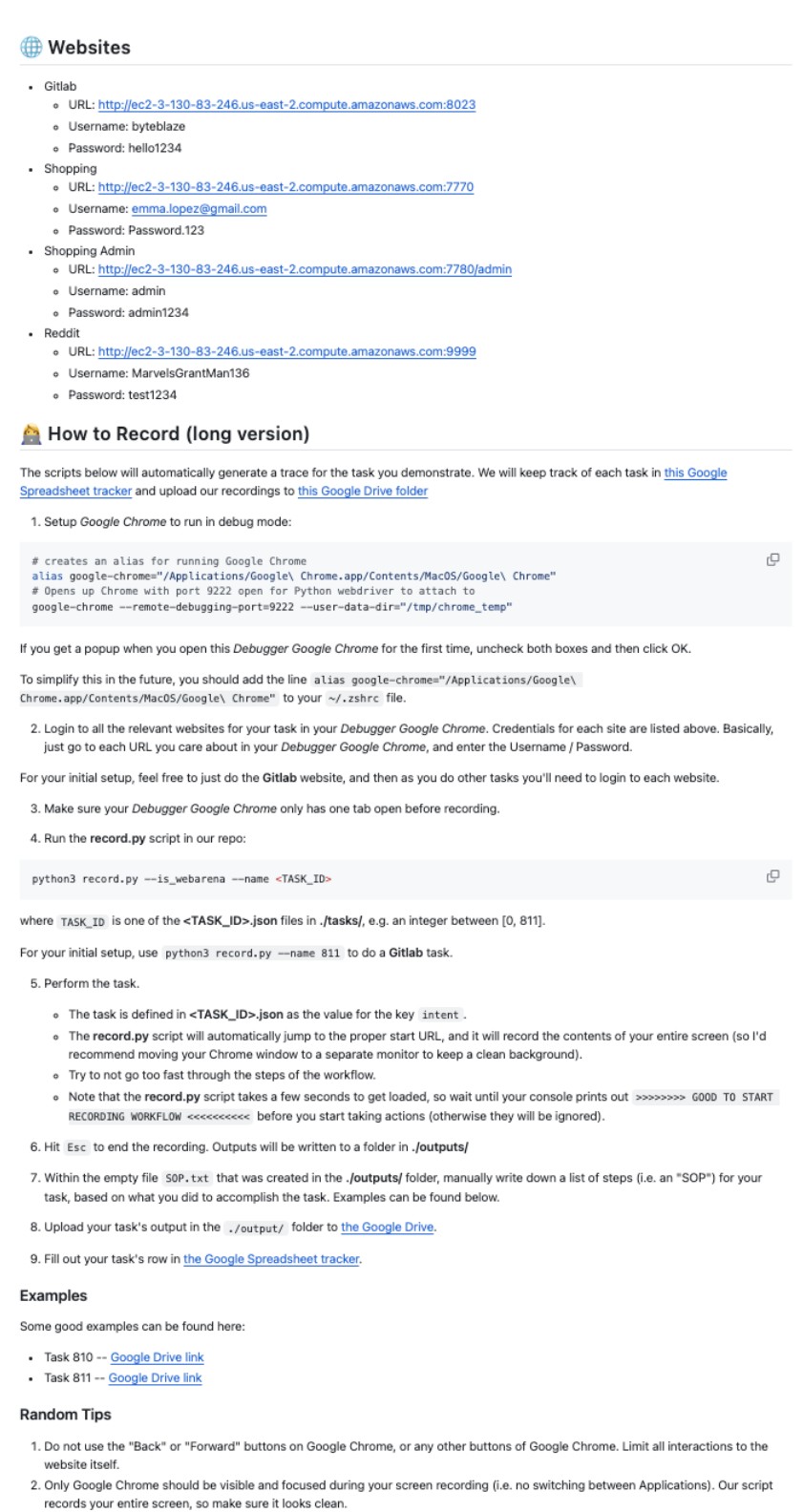

Figure 29: This screenshot provides instructions on the different websites and how to record.

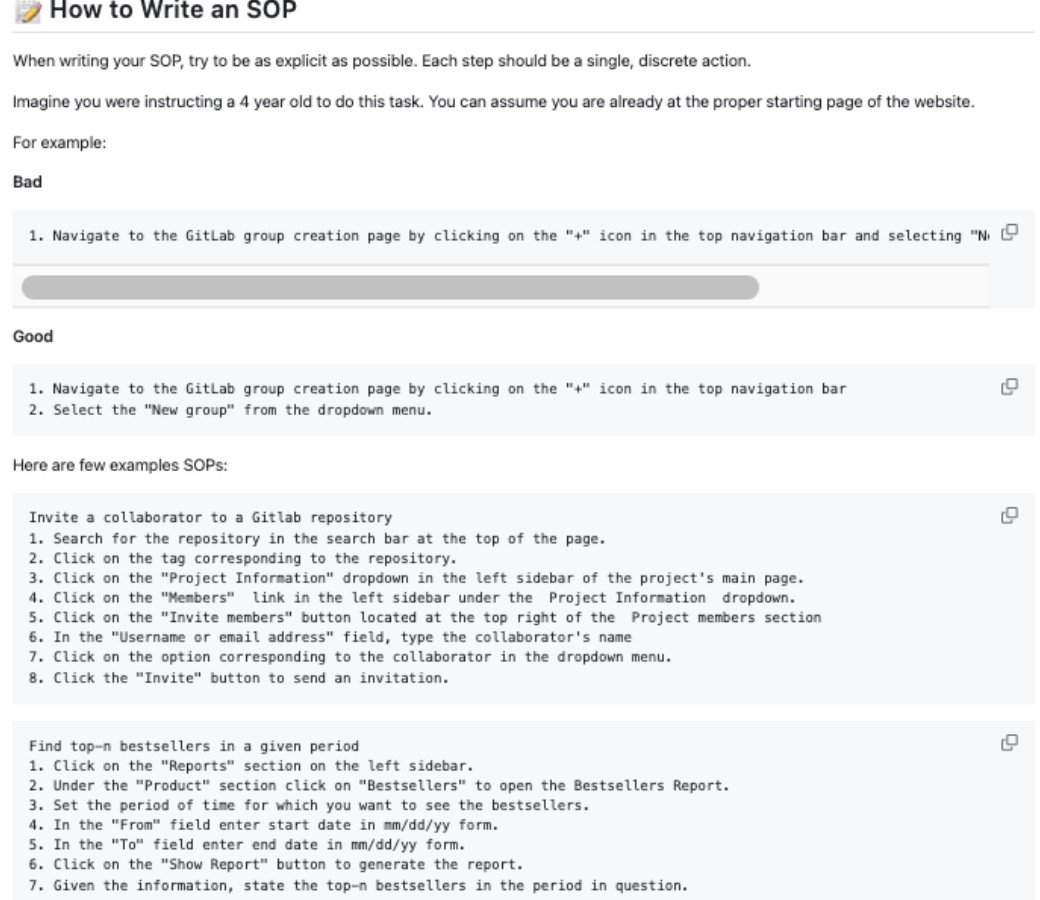

Figure 30: This screenshot explains how to write a SOP.

## 🏆 How to Write a "Gold" SOP

A "gold" SOP should be significantly more fleshed out and detailed than the SOPs that have been written so far.

These will serve as the "ground truths" of our dataset.

The mindset you should be in when writing these is:

> "If I was instructing a 5 year old how to do this general task, and they'd never seen this website before, and their life depended on completing this task correctly, what information would I give them?"

Just like the normal SOPs, the "gold SOPs" must meet the following checklist:

1. **Task Description:** Include the task description at the top of the SOP.txt file. There should be 2 new lines between task description and the steps below.
2. **Element Specification:** Each element mentioned should have a...
   - Descriptive name (i.e., "Accounting tab under the Finances Section"
   - Descriptive location (i.e., "Accounting tab in the left hand side-bar")
3. **Action Specification:** Each step should...
   - Only reference the following actions: **Press, Delete, Click, Type, Scroll** ; no hover!
   - Contain one discrete action (i.e., the step *"Search 'hi'"* is not a discrete action -- decompose it into three separate steps *"Click on the search bar"*, then *"Type 'hi'"*, then *"Press Enter"*)
   - Cover edge cases (i.e., if you don't see the button then scroll down)
   - Match the task (i.e. if the task is "Add Harry to the repo", then you should type "Harry" into the searchbar and not "John")
4. **Faithful to task recording:** The SOP should *exactly* match the steps in the `.mp4` recording.
5. **Generality:** Steps should be general, while also customizing to the specific task (i.e., "Click on the row corresponding to your order. In this case, that is the row with ID 000334")
6. **Formatting:**
   - Names of elements should be in quotes (i.e., click on the "Accounting" button)
   - Steps in SOP should be properly numbered (don't skip numbers or double count)
   - End each line with a period
   - Proper capitalization and spelling

For these "gold" SOPs, we want to put a special emphasis on:

1. **Coverage of edge cases** -- help the user complete the task by making note of ways in which the interface might change, and how to adapt
   - e.g. if a task involves looking through a table of results, and your Shipping Order just happens to be the first one, you should still make a note that the user might have to scroll / paginate through the results until they find the correct Shipping Order.
   - e.g. if you need to click the button at the bottom of the page, you should not assume that the user's browser window has the same size as yours, so you should let them know that they might need to scroll down if they can't see the button.
   - *Example:* Instead of *"Click on the toggle labeled 'Enable Product'"*, you "should write *"Look for the toggle labeled "Enable Product" which should be directly below the "Quantity" field. If the toggle is currently green, that means the product is currently enabled, which means you should click the toggle in order to disable the product. The toggle should change to a grey color to indicate the product is disabled. However, if the toggle is already greyed out, then do nothing since the product had already been disabled."*
2. **Detailed localization** of UI elements -- let the user know exactly where to find the element
   - e.g. "Click the 'Go to Result' button" is not sufficient. You must be extremely detailed in your specification of each element, i.e. its relative position on the screen, its proximity to other landmark elements, its color, what type of element it is, etc.
   - *Example:* Instead of *"Click the 'Edit' link"*, you should write *"Click on the blue "Edit" link at the far righthand side of the row corresponding to the "Configurable Product" we previously found."*
3. **Generalizability** -- write these instructions so that they could apply to any instantiation of the **Intent Template** corresponding to your task
   - e.g. Write your instructions generally, then provide your task-specific stuff as asides.
   - *Example:* Instead of *"Type "Out of Office" in the "What's your status?" input box."*, you should write *"Type the desired Gitlab status in the "What's your status?" input box. In this case, we should type "Out of Office""*
4. **Explanations** of each action -- briefly explain why we take each step (in the context of the next action, or the larger task)
   - e.g. What is the point of each individual action?
   - *Example:* Instead of *"Click the "From" text field"*, you should write *"Click the "From" text field **to focus it.**"*
   - *Example:* Instead of *"Click on the toggle labeled 'Enable Product'"*, you should write *"Click on the toggle labeled 'Enable Product' **to disable the product**"*

Here is an example of a "gold SOP":

```
What is the top-1 best-selling product in 2022?

1. Click on the "Reports" button on the far lefthand sidebar. It has an icon which looks like a chart. It should be
2. In the popup menu that appears, click on the "Bestsellers" link to go to the "Bestsellers Report" page. The link :
3. Locate the field labeled "Period" and click on the dropdown menu to reveal our time options.
4. Click on the "Year" option to set the reporting period to Year.
5. Click on the "From" textbox to focus it. It should be located directly underneath the "Period" field.
6. Type in the first day of our desired time period, which in this case is "01/01/2022". Make sure the textbox is em|
7. Click on the "To" textbox to focus it. It should be located directly underneath the "From" field.
8. Type in the last day of our desired time period, which in this case is "12/31/2022". Make sure the textbox is empl
9. Click on the orange "Show Report" button, which can be found on the top right of the page, in order to generate ou
10. The best-selling products will appear in a table at the bottom of the page. Scroll down if you cannot see the fu|
```

Note the detailed localization for each UI element (*"'Reports' button on the far lefthand sidebar. It has an icon which looks like a chart..."*) the edge case coverage (*"Make sure the textbox is empty before you type into it."* and *"Scroll down if you cannot see the full table"*) and generalizability (*"Type in the first day of our desired time period, which in this case is "01/01/2022"."*).

Figure 31: This screenshot is the first part of instructions on how to write a gold SOP.

**Instructions:**

Follow these steps when writing your "gold SOPs":

1. Look at the tasks you are assigned in the Spreadsheet here.
2. Watch the `.mp4` recordings for all 5 demonstrations of that Task (located in Sheets 1 through 5). Each task should have been done by 4-5 different people.
   - If you see **3 or fewer demonstrations, flag that** and skip it for now.
   - If any of the recordings seem **incorrect, make a note of that.**
3. Rank the recordings from 1 (best) to 5 (worst). For your criteria, consider the following (in decreasing importance):
   - Correctness -- Correct answer >> Incorrect answer
   - Ease of Navigation / "Expert-ness" -- There are "better" ways of being correct
     - e.g. searching for an item via a filter field is more optimal than scrolling to find the item
   - Efficiency -- Fewer actions > More actions
   - Time -- Reasonably timed recording > Short > Long
     - e.g. racing through a recording is bad (should pause at least 1s after each action), but pausing 10s is bad too 4. Record your ranking in **Columns D-H** in the Spreadsheet 4. Choose the best recording (the one you scored as "1") 5. Write a "gold SOP" for that best recording. Feel free to use the already written SOPs as a starting point. 6. Name the file "[TASK_ID].txt", e.g. "1.txt" for task #1 6. Upload your "gold SOP" to this Google Drive folder and add a link in **Column C** of the Spreadsheet

# Output Format

Recordings are stored in two output formats:

1. `trace.json` - a JSON log of all actions and states
   - Actions
     - Clicks: (x,y) on screen, HTML element (if on a webpage), timestamp
     - Keystrokes: Key, HTML element (if on webpage), timestamp
   - States
     - Name of active desktop application (e.g. "Google Chrome", "Terminal", etc.)
     - Bounding boxes of accessible HTML elements (if on webpage)
2. `trace.mp4` - a full screen recordings of you completing this task
3. `screenshots/` - a folder containing screen shots of each state (taken from still frames in `trace.mp4` corresponding to the timestamps of each state)
4. `SOP.txt` - A manually generated summary of the steps of this process, based on the task description and screenshots

# Todos

Divergence from `record.py` in eclair:

1. `state.to_json()` auto-serializes `json_state` as string in eclair, but that's not true here.
2. In eclair, `execute_scripts` has been renamed `execute_js_scripts`, and now executes `proxy-select.js` (so that dropdowns appear in playwright/selenium). Here, we don't do `proxy-select.js`.
3. **Important:** Need to run `trace_cleanup.py` in eclair to fix (x,y) coords
4. Add `is_focused`, `is_checked`, etc. to elements

# Bugs

- Fix `window_position` and `window_size` bugs; also adjust `x,y` coords to account for browser chrome/position within `element_attributes` of `action`.
- Might want to expand `json_state` to be more inclusive, or to always include last element clicked

Figure 32: This screenshot is the second part of instructions on how to write a gold SOP.

# F  Prompts

In this section, we delineate the various prompts used in **WONDERBREAD** for generating model outputs and conducting LLM-based evaluations.

## F.1  Documentation Task Prompts

Prompts utilized in the two Documentation related tasks introduced in Section 4.1.

### F.1.1  Demo Segmentation

In this section, we have provided the prompts utilized in the demonstration segmentation task. The full prompt is broken up into two parts, the second of which has two variations depending one what the user wants to model to return. These prompts are available in our Github repo at the following location: ⬚ https://github.com/HazyResearch/wonderbread/blob/main/wonderbread/benchmark/tasks/documentation/demo_segmentation/prompts.py.

```
prompt__intro: str = lambda n_tasks, task_descriptions: f"""# Task
You are a process mining automation tool. Your are given a recording of a worker doing multiple
    workflows (potentially overlapping).
Your job is to segment this recording into discrete workflows -- i.e. identify which actions correspond
    to which workflow.
Workflow segmentation is important for process mining because it allows us to analyze the performance
    of each workflow separately.

# Workflow

The {n_tasks} workflows being executed in the recording are as follows:
{task_descriptions}

# Workflow Demonstration

You are given the following recording of the worker completing these {n_tasks} workflows over the
    course of this recording.

The recording is presented in chronological order.
The workflows are executed in sequence, but may be present in any order. You can assume that the worker
    always finishes a workflow before starting the next one.
The recording may include both screenshots and the actions taken to transition between screenshots.

Each screenshot and action is labeled with a unique identifier ("UUID"). We will use these UUIDs to
    refer to specific screenshots and actions in the recording when segmenting the recording into the
    {n_tasks} workflows.

Here is the overall recording:"""
```

Prompt 1: Introduction part of the prompt for the demonstration segmentation task. This introduction precedes the collection screens

```
prompt__close_uuid: str = lambda n_tasks, task_descriptions, sops : f"""
# Instructions

Given what you observe in the previous recording, please classify each UUID as belonging to one of the
    {n_tasks} workflows.

As a reminder, the workflows are as follows. Each workflow is assigned a classification letter:
{task_descriptions}

The workflows may be present in the recording in any order. You can assume that the worker always
    finishes a workflow before starting the next one, so there are no overlapping workflows.

{sops if sops else ""}

Provide your answer as a JSON dictionary with the following format:
{{
    "UUID_1": <workflow classification>,
    "UUID_2": <workflow classification>,
    ...
}}

Please write your JSON below:
"""
```

Prompt 2: A variation of the second part of the segmentation task prompt that asks the model to predict the start and end screenshots/actions for each workflow demonstration. As shown in the prompt, the model is asked to structure it's output in the form of a JSON dictionary with specific keys.

```
prompt__close_start_end: str = lambda n_tasks, task_descriptions, sops : f"""
# Instructions

Given what you observe in the previous recording, please tell me the start and end UUIDs for each of
    the {n_tasks} workflows.

As a reminder, the workflows are as follows. Each workflow is assigned a classification letter:
{task_descriptions}

The workflows may be present in the recording in any order. You can assume that the worker always
    finishes a workflow before starting the next one, so there are no overlapping workflows.

{sops if sops else ""}

Provide your answer as a JSON dictionary with the following format:
{{
    "A": {{
        "start": <start UUID for workflow A>,
        "end": <end UUID for workflow B>
    }},
    "B": {{
        "start": <start UUID for workflow A>,
        "end": <end UUID for workflow B>
    }},
    ...
}}

You must respond with valid JSON. Please write your JSON below:
"""
```

Prompt 3: A variation of the second part of the segmentation task prompt that asks the model to classify each individual screenshot and action.

### F.1.2 SOP Generation

Here we have included the prompts utilized in the SOP Generation task. The full prompt is broken apart into multiple partial prompts with slight variations depending on the ablation setting (what information is being provided to the model). These prompts are available in our codebase at the following link: ⬡ https://github.com/HazyResearch/wonderbread/blob/main/wonderbread/benchmark/tasks/documentation/sop_generation/prompts.py

```
prompt__start: str = lambda task_descrip, ui_name : f"""# Task
Your job is to write a standard operating procedure (SOP) for a workflow.

# Workflow

The workflow is: "{task_descrip if task_descrip else 'Some unspecified digital task'}"

# User Interface

The workflow will be executed within a web application. The web application is called: "{ui_name}"
"""
```

Prompt 4: Introduction partial prompt for the SOP generation task. This text preceeds the following "Final Part" of the SOP Generation prompt as well as any representation of the demonstration included.

```
prompt__end: str = lambda : f"""Here is a sample format for what your SOP should look like:
```
1. Type the name of the repository in the search bar at the top left of the screen. The placeholder
    text in the search bar is "Find a repository...", and it is located directly to the right of the
    site logo.
2. A list of repositories will appear on the next page. Scroll down until you see a repository with the
    desired name. The name of the repository will be on the lefthand side of the row in bold font.
    Stop when you find the name of the repository.
3. Click on the relevant repository to go to the repository's main page.
```

Note, the above SOP is just an example. Use the same format, but the actions will be different for your
    workflow.

Be as detailed as possible. Each step should be a discrete action that reflects what you see in the
    corresponding step. Don't skip steps.

Please write your SOP below:"""
```

Prompt 5: Final part of the partial prompt for the SOP Generation task. This text is preceded by both the introduction (above) and any other included representations of the demonstration (depending on the ablation setting).

```
    prompt__td_intro: str = lambda task_descrip, ui_name: f"""{prompt__start(task_descrip, ui_name)}"""

prompt__td_close: str = lambda : f"""
# Instructions

Write an SOP for completing this workflow on this website. The SOP should simply contain an enumerated
    list of actions taken by the user to complete the given workflow.
In your SOP, list all of the actions taken (i.e., buttons clicked, fields entered, mouse scrolls etc.).
    Be descriptive about elements (i.e., 'the subheading located under the "General" section').

{prompt__end()}"""

prompt__td_kf_intro: str = lambda task_descrip, ui_name: f"""{prompt__start(task_descrip, ui_name)}

# Workflow Demonstration

You are given the following sequence of screenshots which were sourced from a demonstration of the
    workflow.
The screenshots are presented in chronological order.

Here are the screenshots of the workflow:"""

prompt__td_kf_close: str = lambda : f"""
# Instructions

Given what you observe in the screenshots, write an SOP for completing the workflow on this website.
    The SOP should simply contain an enumerated list of actions taken by the user to complete the
    given workflow.
In your SOP, list all of the actions taken (i.e., buttons clicked, fields entered, mouse scrolls etc.).
    Be descriptive about elements (i.e., 'the subheading located under the "General" section'). Use
    the location of the mouse to identify which exact elements were clicked.

{prompt__end()}"""

prompt__td_act_intro: str = lambda task_descrip, ui_name: f"""{prompt__start(task_descrip, ui_name)}

# Workflow Demonstration

You are given the following sequence of actions which were sourced from a demonstration of the workflow
    .
The actions are presented in chronological order.
Note that the action is written in a simplified DSL (domain-specific language) that we use to describe
    actions taken by users. You will need to translate this into a natural language description of the
     action and add more details about what was happening, why, and what elements were interacted with
    .

Here are the actions of the workflow:"""

prompt__td_act_close: str = lambda : f"""
# Instructions

Given what you observe in the sequence of DSL actions, write an SOP for completing the workflow on this
    website. The SOP should simply contain an enumerated list of actions taken by the user to
    complete the given workflow.
In your SOP, list all of the actions taken (i.e., buttons clicked, fields entered, mouse scrolls etc.).
    Be descriptive about elements (i.e., 'the subheading located under the "General" section'). Use
    the location of the mouse to identify which exact elements were clicked.

{prompt__end()}"""

prompt__td_kf_act_intro: str = lambda task_descrip, ui_name: f"""{prompt__start(task_descrip, ui_name)}

# Workflow Demonstration

You are given the following sequence of screenshots which were sourced from a demonstration of the
    workflow.
The screenshots are presented in chronological order.
```

```
Between each screenshot, you are also provided the action that was taken to transition between
    screenshots.
However, the action is written in a simplified DSL (domain-specific language) that we use to describe
    actions taken by users. You will need to translate this into a natural language description of the
     action and add more details about what was happening, why, and what elements were interacted with
     .

Here are the screenshots and actions of the workflow:"""

prompt__td_kf_act_close: str = lambda : f"""
# Instructions

Given what you observe in the previous sequence of screenshots and DSL actions, write an SOP for
    completing the workflow for this specific interface. The SOP should simply contain an enumerated
    list of actions taken by the user to complete the given workflow.
In your SOP, list all of the actions taken (i.e., buttons clicked, fields entered, mouse scrolls etc.).
     Be descriptive about elements (i.e., 'the subheading located under the "General" section'). Use
    the location of the mouse to identify which exact elements were clicked.

{prompt__end()}
"""
```

Prompt 6: Multiple variations of the full SOP Generation prompt. The various generations correspond to different ablation settings, ie. which representations of the demonstration are shown to the model.

We also created alternative forms of the SOP Generation task's prompts that task the model to build the SOP by examining each step of the workflow independently. These variations are included below:

```
prompt__start__pairwise: str = lambda task_descrip, ui_name : f"""# Task
Your job is to determine the single action that was taken between these screenshots were taken.

# User Interface

The web application where the screenshots are taken from is called: "{ui_name}"
"""

prompt__end__pairwise: str = lambda : f"""Here is a sample format for what your output should look like
    :
```
1. Click on the searchbar at the top left of the screen to focus it. The placeholder text in the search
    bar is "Find a repository...", and it is located directly to the right of the site logo.
2. Type the name of the repository into the searchbar.
```

Note, the above output is just an example. Use the same format, but the action might be different for
    your screenshots.
You might have only one item in your output, or you might have multiple items. It depends on the action
    that took place between the screenshots.
Be as detailed as possible. Each step should be a discrete action that reflects what you see in the
    screenshots. Don't skip steps.
Only include the action that took place between the screenshots, and do not make any assumptions about
    what happened before or after the screenshots were taken.

Please write your output below:"""

prompt__td_kf_intro__pairwise: str = lambda task_descrip, ui_name: f"""{prompt__start__pairwise(
    task_descrip, ui_name)}

# Workflow Demonstration

You are given the following two screenshots which were sourced from a demonstration of the workflow.
The screenshots are presented in chronological order.
The first one was taken directly before the action was taken, and the second one was taken directly
    after the action was executed.
Note that these screenshots could have been taken at any step of the workflow.

Here are the screenshots of this specific step from the larger workflow:"""

prompt__td_kf_close__pairwise: str = lambda : f"""
# Instructions

Given what you observe in the screenshots, write the step(s) corresponding to this action that would go
     into a larger SOP for completing the workflow on this website.
Make sure to list all of the actions taken to go from one screenshot to the other (i.e., buttons
    clicked, fields entered, mouse scrolls etc.). Be descriptive about elements (i.e., 'the subheading
     located under the "General" section'). Use the location of the mouse to identify which exact
    elements were clicked.

{prompt__end__pairwise()}
"""

prompt__td_kf_act_intro__pairwise: str = lambda task_descrip, ui_name: f"""{prompt__start__pairwise(
    task_descrip, ui_name)}

# Workflow Demonstration
```

```
You are given the following two screenshots which were sourced from a demonstration of the workflow.
The screenshots are presented in chronological order.
The first one was taken directly before the action was taken, and the second one was taken directly
    after the action was executed.
Note that these screenshots could have been taken at any step of the workflow.

Between each screenshot, you are also provided the action that was taken to transition between
    screenshots.
However, the action is written in a simplified DSL (domain-specific language) that we use to describe
    actions taken by users. You will need to translate this into a natural language description of the
     action and add more details about what was happening, why, and what elements were interacted with
    .

Here are the screenshots and action of this specific step from the larger workflow:"""

prompt__td_kf_act_close__pairwise: str = lambda : f"""
# Instructions

Given what you observe in the screenshots and DSL action, write the step(s) corresponding to this
    action that would go into a larger SOP for completing the workflow on this website.
Make sure to list all of the actions taken to go from one screenshot to the other (i.e., buttons
    clicked, fields entered, mouse scrolls etc.). Be descriptive about elements (i.e., 'the subheading
     located under the "General" section'). Use the location of the mouse to identify which exact
    elements were clicked.

{prompt__end__pairwise()}
"""

prompt__join_pairwise: str = lambda sop, separator : f"""
Your job is to create a standard operating procedure (SOP) for a workflow that outlines each step taken
     to complete the workflow.

Previously, you were given subsets of consecutive screenshots taken from a longer sequence of
    screenshots of a workers doing the workflow. You were asked to write the step(s) taken between
    each screenshot. Our goal is to compile these smaller sets of steps into a larger SOP for
    completing the entire workflow.

I've copied your responses for this previous pairwise screenshot analysis below. Each pair of
    screenshots is separated by {separator}.

```
{sop}
```

Your job now is to combine these steps into a single, coherent SOP for completing the entire workflow.
     The steps are ordered chronologically, so you do not need to worry about the ordering of the
    steps. Instead, you should remove any duplicate steps and ensure that the steps flow logically
    from one to the next.

Please write your unified SOP below:
"""
```

Prompt 7: Alternative variations of the SOP Generation Prompts that build the SOP by independently examining each step of the workflow.

## F.2 Knowledge Transfer Prompts

Prompts utilized in the two Knowledge Transfer related tasks introduced in Section 4.2.

### F.2.1 Demo Validation

This section contains prompts utilized in the Demonstration Validation Task, in which the model is asked to characterize if the workflow successfully completed or if the correct overall trajectory was followed. These prompts are available in the following file in our Github repo: ○ https://github.com/HazyResearch/wonderbread/blob/main/wonderbread/benchmark/tasks/knowledge_transfer/demo_validation/prompts.py

```
prompt__validate_task_completion__intro: str = lambda task_descrip, sop: f"""# Task
Your job is to decide whether the workflow was successfully completed, as depicted by the following
    sequence of screenshots.

# Workflow

The workflow is: "{task_descrip if task_descrip else 'Unknown'}"

# User Interface

The workflow was executed within the web application shown in the screenshots.

{section__sop(sop) if sop is not None else ''}

# Workflow Demonstration

You are given the following sequence of screenshots which were sourced from a demonstration of the
    workflow.
The screenshots are presented in chronological order.

Between each screenshot, you are also provided the action that was taken to transition between
    screenshots.

Here are the screenshots and actions of the workflow:"""

prompt__validate_task_completion__close: str = lambda : f"""
# Instructions

Given what you observe in the previous sequence of screenshots and actions, was the workflow
    successfully completed?
If the workflow is asking a question, consider it completed successfully if you could deduce the answer
     to the question by viewing the screenshots.
If the workflow was completed successfully, then set `was_completed` to `true`

Provide your answer as a JSON dictionary with the following format:
{{
    "thinking": <think step by step what the answer should be>,
    "was_completed": <true/false>
}}

Please write your JSON below:
"""
```

Prompt 8: The two parts of the prompt utilized to evaluate task 'completion', ie. whether the workflow was successfully completed. As shown in the prompt, the model is asked to structure it's output in the form of a JSON dictionary with specific keys.

```
prompt__validate_task_trajectory__intro: str = lambda task_descrip: f"""# Task
Your job is to decide whether the workflow that is demonstrated in the following sequence of
    screenshots ACCURATELY FOLLOWED the Step-by-Step Guide.

# Workflow

The workflow is: "{task_descrip if task_descrip else 'Unknown'}"

# User Interface

The workflow was executed within the web application shown in the screenshots.

# Workflow Demonstration

You are given the following sequence of screenshots which were sourced from a demonstration of the
    workflow.
The screenshots are presented in chronological order.

Between each screenshot, you are also provided the action that was taken to transition between
    screenshots.

Here are the screenshots and actions of the workflow:"""

prompt__validate_task_trajectory__close: str = lambda sop : f"""

{section__sop(sop) if sop is not None else ''}

NOTE: The screenshots may not map 1-to-1 to the steps in the Step-by-Step Guide. i.e. screenshot #3 may
     correspond to step #2 (or multiple steps) in the Step-by-Step Guide.
However, as long as the general flow of the workflow is the same, then the workflow is considered to
    have accurately followed the Step-by-Step Guide.
Also note that elements may be interchangeably referred to as buttons or links (the distinction is not
    important).

# Instructions

Given what you observed in the previous sequence of screenshots and actions, was the Step-by-Step Guide
     accurately followed? If any of the steps are missing, or if any of the steps were performed out
    of order, then the Step-by-Step Guide was not accurately followed and `was_accurate` should be `
    false`.

Provide your answer as a JSON dictionary with the following format:
{{
    "thinking": <think step by step what the answer should be>,
    "inaccurate_steps": <optional list of steps that were inaccurate>
    "was_accurate": <true/false>
}}

Please write your JSON below:
"""
```

Prompt 9: The two parts of the prompt utilized to evaluate task 'trajectory', ie. whether the step-by-step guide was accurately followed. As shown in the prompt, the model is asked to structure it's output in the form of a JSON dictionary with specific keys.

```
section__sop: str = lambda sop: f"""# Step-by-Step Guide

Here are the sequence of steps that were supposed to be followed to complete this workflow:
{sop}
"""
```

Prompt 10: Helper function utilized in building the Task Trajectory Prompts. Allows the user to quickly interject a formatted representation of the Step-by-Step Guide (SOP) for the demonstration.

### F.2.2 Question Answering

In this section, we list the prompts used for our GPT4-based scoring of the "completeness", "soundness", "clarity" and "compactness" of model-generated answers for the Question Answering task. Note that in these prompts, a score of 1 corresponds to "good" and 3 corresponds to "bad." To obtain Figure 6 in the main results, however, these scores were flipped for clarity (i.e. in the radar plot, a score of 1 is "bad." The prompts listed below are also available at  https://github.com/HazyResearch/wonderbread/blob/main/wonderbread/benchmark/tasks/knowledge_transfer/question_answering/prompts.py

```
prompt__completeness_score: str = lambda question, human_label, response: f"""# Task
Your job is to evaluate the completeness of the response to a given question.
You are also provided with the human label for the question, which is the ideal response.

The question provided is related to analyzing a workflow demonstration in a web application.
You won't be provided with information about the web application, but only the question, human label,
    and response.
You should evaluate the response based on the information provided in the response itself.

For evaluating the completeness of the response, you should consider the following:
- Whether the response fully answers the question
- Whether the response is complete in covering relevant information
- Whether the response encompasses all aspects and outcomes related to the question

Please rate the completeness of the response on a scale of 1-3, where:
1. The response is complete and fully answers the question
2. The response is mostly complete but is missing some key information
3. The response is incomplete and does not fully answer the question

Here is a sample question, human label and response for your reference:
Question: How would a user completing the task know that the workflow is completed?
Human Label: When the user reaches a page whose title is "Customers" and it has a table showing
    customer information like name, phone number, etc.
Response: A user would know that the workflow is completed by successfully navigating to the "Customers
    " page, where they can view the list of all customers, as shown in the final screenshot. This
    indicates that the user has reached the intended section of the web application to manage or
    review customer information.
Score: 2
Explanation: The response is mostly complete but is missing some key information. The response should
    also mention that the page should have a table showing customer information like name, phone
    number, etc, as per the human label.

Also, provided are two other examples for your reference:
Question: Explain what the most common failure modes might be for a user performing this task.
Human Label: A common failure mode could be replying to an existing comment instead of posting a new
    comment.
Response: The most common failure modes for a user performing this task might include entering an
    incorrect or inappropriate comment, failing to click the 'Post' button after typing the comment,
    or experiencing a technical issue such as a browser error or a problem with the website's server
    that prevents the comment from being posted successfully.
Score: 2

Question: Why does the user click the Commits button in step #5?
Human Label: To find the total number of commits Eric made, the "commits" button must be clicked.
Response: The user clicks the "Commits" button in step #5 to view the list of all commits made in the
    desired repository, specifically to find commits that occurred on a particular date.
Score: 3

Here are the details for the question you need to evaluate:

# Question
{question}

# Human Label
{human_label}

# Response to evaluate
{response}

Return only the number corresponding to the rating, nothing else.
"""
```

Prompt 11: Prompt for evaluating "completeness" scores in the question answering task. The model is tasked to only return a number corresponding to the ranking and nothing else.

```
prompt__soundness_score: str = lambda question, human_label, response: f"""# Task
Your job is to evaluate the soundness of the response to a given question.
You are also provided with the human label for the question, which is the ideal response.

The question provided is related to analyzing a workflow demonstration in a web application.
You won't be provided with information about the web application, but only the question, human label,
    and response.
You should evaluate the response based on the information provided in the response itself.

For evaluating the soundness of the response, you should consider the following:
- Whether the response accurately answers the question
- Whether the response avoids assumptions not backed by data or evidence
- Whether the response is logical and reasonable based on the context provided

Please rate the soundness of the response on a scale of 1-3, where:
1. The response is completely sound and logical without making extra assumptions
2. The response is mostly sound but may contain some minor logical flaws or assumptions
3. The response is unsound and contains major logical flaws or assumptions

Here is a sample question, human label and response for your reference:
Question: How would a user completing the task know that the workflow is completed?
Human Label: When the user reaches a page whose title is "Customers" and it has a table showing
    customer information like name, phone number, etc.
Response: When the user sees the list of customers after just clicking on the "Customers" tab.
Score: 2
Explanation: The response is partially sound but incorrectly says that the user should just click on
    the "Customers" tab, which is not accurate as the user would have to perform more actions to reach
     the final page.

Also, provided are two other examples for your reference:
Question: Explain what the most common failure modes might be for a user performing this task.
Human Label: A common failure mode could be replying to an existing comment instead of posting a new
    comment.
Response: The most common failure modes for a user performing this task might include entering an
    incorrect or inappropriate comment, failing to click the 'Post' button after typing the comment,
    or experiencing a technical issue such as a browser error or a problem with the website's server
    that prevents the comment from being posted successfully.
Score: 1

Question: Why does the user click the Commits button in step #5?
Human Label: To find the total number of commits Eric made, the "commits" button must be clicked.
Response: The user clicks the "Commits" button in step #5 to view the list of all commits made in the
    desired repository, specifically to find commits that occurred on a particular date.
Score: 1

Here are the details for the question you need to evaluate:

# Question
{question}

# Human Label
{human_label}

# Response to evaluate
{response}

Return only the number corresponding to the rating, nothing else.
"""
```

Prompt 12: Prompt for evaluating "soundness" scores in the question answering task. The model is tasked to only return a number corresponding to the ranking and nothing else.

```
prompt__clarity_score: str = lambda question, response: f"""# Task
Your job is to evaluate the clarity of the response to a given question.

The question provided is related to analyzing a workflow demonstration in a web application.
You won't be provided with information about the web application, but only the question, human label,
    and response.
You should evaluate the response based on the information provided in the response itself.

For evaluating the clarity of the response, you should consider the following:
- Whether the response is presented in an unambiguous and straightforward manner
- Whether the response needs any clarification or additional information to be easily understood
- Whether the response can have only one interpretation

Please rate the clarity of the response on a scale of 1-3, where:
1. The response is clear, unambiguous, and easily understood
2. The response is somewhat clear but may require some additional information or clarification
3. The response is unclear, ambiguous, or can have multiple interpretations

Here is a sample question and response for your reference:
Question: How would a user completing the task know that the workflow is completed?
Response: When the user sees the list of customers after just clicking on the "Customers" tab.
Score: 2
Explanation: The response is somewhat clear but could be more specific about the final outcome.

Here is another sample question and response for your reference:
Question: Explain what the most common failure modes might be for a user performing this task.
Response: Not scrolling down through all the posts.
Score: 3
Explanation: The response is unclear and lacks details on why not scrolling down through all the posts
    can lead to failure modes.

Also, provided is another example for your reference:
Question: Explain what the most common failure modes might be for a user performing this task.
Human Label: A common failure mode could be replying to an existing comment instead of posting a new
    comment.
Response: The most common failure modes for a user performing this task might include entering an
    incorrect or inappropriate comment, failing to click the 'Post' button after typing the comment,
    or experiencing a technical issue such as a browser error or a problem with the website's server
    that prevents the comment from being posted successfully.
Score: 1

Here are the details for the question you need to evaluate:

# Question
{question}

# Response to evaluate
{response}

Return only the number corresponding to the rating, nothing else.
"""
```

Prompt 13: Prompt for evaluating "clarity" scores in the question answering task. The model is tasked to only return a number corresponding to the ranking and nothing else.

```
prompt__compactness_score: str = lambda question, response: f"""# Task
Your job is to evaluate the compactness of the response to a given question.

The question provided is related to analyzing a workflow demonstration in a web application.
You won't be provided with information about the web application, but only the question, human label,
    and response.
You should evaluate the response based on the information provided in the response itself.

For evaluating the compactness of the response, you should consider the following:
- Whether the response is short and to the point
- Whether the response is concise and does not contain unnecessary information

Please rate the compactness of the response on a scale of 1-3, where:
1. The response is concise, to the point, and does not contain any unnecessary information
2. The response is somewhat compact but may contain some unnecessary information
3. The response is verbose and contains a lot of unnecessary information

Here is a sample question and response for your reference:
Question: Explain what the most common failure modes might be for a user performing this task.
Response: The most common failure modes for a user performing this task could include not being able to
    locate the "Forums" button due to changes in the website layout or updates, difficulty in finding
    the "news" section if the alphabetical sorting changes or if the user overlooks it, and
    potentially missing the "down arrow" to dislike submissions if the interface is not intuitive or
    if the symbols used for liking and disliking are not clear. Additionally, users might struggle to
    identify posts by "Hrekires" if there are many submissions or if the username display is not
    prominent.
Score: 2
Explanation: The response is somewhat compact but contains unnecessary information about the specific
    failure modes. It could be more concise and focus on the general failure modes.

Also, provided are two other examples for your reference:
Question: Explain what the most common failure modes might be for a user performing this task.
Human Label: A common failure mode could be replying to an existing comment instead of posting a new
    comment.
Response: The most common failure modes for a user performing this task might include entering an
    incorrect or inappropriate comment, failing to click the 'Post' button after typing the comment,
    or experiencing a technical issue such as a browser error or a problem with the website's server
    that prevents the comment from being posted successfully.
Score: 3

Question: Why does the user click the Commits button in step #5?
Human Label: To find the total number of commits Eric made, the "commits" button must be clicked.
Response: The user clicks the "Commits" button in step #5 to view the list of all commits made in the
    desired repository, specifically to find commits that occurred on a particular date.
Score: 2

Here are the details for the question you need to evaluate:

# Question
{question}

# Response to evaluate
{response}

Return only the number corresponding to the rating, nothing else.
"""
```

Prompt 14: Prompt for evaluating "compactness" scores in the question answering task. The model is tasked to only return a number corresponding to the ranking and nothing else.

## F.3 Improvement Task Prompts

Prompts utilized in the two Improvement tasks related tasks introduced in Section 4.3.

### F.3.1 SOP Improvement

In this section, we enumerate the three variations of SOP Improvements task. These prompts are available in our Github at the following link: ⭘ https://github.com/HazyResearch/wonderbread/blob/main/wonderbread/benchmark/tasks/improvement/sop_improvement/prompts.py

```
prompt__rewrite_sop__intro: str = (
    lambda task_descrip: f"""# Task
Your job is to improve upon the Standard Operating Procedure (SOP) for the workflow that is
    demonstrated in the following sequence of screenshots and actions.

# SOP Rubric
- Element Specification: Each element referenced in the SOP has a descriptive name and location (i.e.,
    "Accounting tab under the Finances Section")
- Action Type: The only actions referenced in the SOP should be one of the following: Press, Delete,
    Click, Type, Scroll.
- Edge Case Coverage: the SOP describes any edge cases that the user might encounter, and how to solve
    them  (i.e., "if you don't see button, scroll down")
- Discrete Action: The SOP only contains one discrete action per step (i.e., the action "click on the
    text bar and type "hello"" should be converted to two separate steps: (1) click on the text bar
    and (2) type "hello")
- Action Relevance: Each action should be true to the task  (i.e., if the task is to find the "grey t-
    shirt" clothing item, then an action which instructs the user to type text in the search bar
    should type the text "grey t-shirt")
- Generality: The steps of the SOP should reflect how to do this task in general and not overfit to the
     specific window size or screen of the demonstration (i.e., "Scroll until you find the row with
    your order" rather than "Scroll 130 pixels down")

# Workflow

The workflow is: "{task_descrip if task_descrip else 'unspecified'}"

# User Interface

The workflow was executed within the web application shown in the screenshots.

# Workflow Demonstration

You are given the following sequence of screenshots which were sourced from a demonstration of the
    workflow.
The screenshots are presented in chronological order.

Between each screenshot, you are also provided the action that was taken to transition between
    screenshots.

Here are the screenshots and actions of the workflow:"""
)

prompt__rewrite_sop__close: str = (
    lambda sop: f"""

# Standard Operating Procedure

Here are the sequence of steps that you should have followed to complete this workflow:

{sop}

NOTE: The screenshots may not map 1-to-1 to the steps in the Standard Operating Procedure. i.e.
    screenshot #3 may correspond to step #2 (or multiple steps) in the Standard Operating Procedure.
However, as long as the general flow of the workflow is the same, then the workflow is considered to
    have accurately followed the Standard Operating Procedure.
Also note that elements may be interchangeably referred to as buttons or links (the distinction is not
    important).

# Instructions

Given what you observed in the previous sequence of screenshots and actions, rewrite the Standard
    Operating Procedure to increase clarity and accuracy. If any of the steps are missing, or if any
    of the steps were performed out of order, then the Standard Operating Procedure should be updated
    to correct these mistakes.

Provide your answer as a numbered list with the following format:
1. The first action to be taken goes here
2. The second action to be taken goes here
3. The third action goes here ...

Please write the new updated Standard Operating Procedure below using the guidelines from the SOP
    Rubric above:
"""
)
```

Prompt 15: The introduction and closing parts of the prompt for basic form of the SOP Improvement task where only the SOP is given to the model. The model is asked to output it's response in the form of a SOP given an example structure.

```python
prompt__rewrite_sop__intro_kf: str = (
    lambda task_descrip: f"""# Task
Your job is to improve upon the Standard Operating Procedure (SOP) for the workflow that is
    demonstrated in the following sequence of screenshots and actions.

# SOP Rubric
- Element Specification: Each element referenced in the SOP has a descriptive name and location (i.e.,
    "Accounting tab under the Finances Section")
- Action Type: The only actions referenced in the SOP should be one of the following: Press, Delete,
    Click, Type, Scroll.
- Edge Case Coverage: the SOP describes any edge cases that the user might encounter, and how to solve
    them  (i.e., "if you don't see button, scroll down")
- Discrete Action: The SOP only contains one discrete action per step (i.e., the action "click on the
    text bar and type "hello"" should be converted to two separate steps: (1) click on the text bar
    and (2) type "hello")
- Action Relevance: Each action should be true to the task  (i.e., if the task is to find the "grey t-
    shirt" clothing item, then an action which instructs the user to type text in the search bar
    should type the text "grey t-shirt")
- Generality: The steps of the SOP should reflect how to do this task in general and not overfit to the
    specific window size or screen of the demonstration (i.e., "Scroll until you find the row with
    your order" rather than "Scroll 130 pixels down")

# Workflow

The workflow is: "{task_descrip if task_descrip else 'unspecified'}"

# User Interface

The workflow was executed within the web application shown in the screenshots.

# Workflow Demonstration

You are given the following sequence of screenshots which were sourced from a demonstration of the
    workflow.
The screenshots are presented in chronological order.

Here are the screenshots of the workflow:"""
)

prompt__rewrite_sop__close_kf: str = (
    lambda sop: f"""

# Standard Operating Procedure

Here are the sequence of steps that you should have followed to complete this workflow:

{sop}

NOTE: The screenshots may not map 1-to-1 to the steps in the Standard Operating Procedure. i.e.
    screenshot #3 may correspond to step #2 (or multiple steps) in the Standard Operating Procedure.
However, as long as the general flow of the workflow is the same, then the workflow is considered to
    have accurately followed the Standard Operating Procedure.
Also note that elements may be interchangeably referred to as buttons or links (the distinction is not
    important).

# Instructions

Given what you observed in the previous sequence of screenshots, rewrite the Standard Operating
    Procedure to increase clarity and accuracy. If any of the steps are missing, or if any of the
    steps were performed out of order, then the Standard Operating Procedure should be updated to
    correct these mistakes.

Provide your answer as a numbered list with the following format:
1. The first action to be taken goes here
2. The second action to be taken goes here
3. The third action goes here ...

Please write the new updated Standard Operating Procedure below using the guidelines from the SOP
    Rubric above:
"""
)
```

Prompt 16: The introduction and closing parts of the prompt for the version of the SOP Improvement task where the key frames are provided to the model in addition to the SOP. The model is asked to output it's response in the form of a SOP given an example structure.

```python
prompt__rewrite_sop__intro_act: str = (
    lambda task_descrip: f"""# Task
Your job is to improve upon the Standard Operating Procedure (SOP) for the workflow that is
    demonstrated in the following sequence of screenshots and actions.

# SOP Rubric
- Element Specification: Each element referenced in the SOP has a descriptive name and location (i.e.,
    "Accounting tab under the Finances Section")
- Action Type: The only actions referenced in the SOP should be one of the following: Press, Delete,
    Click, Type, Scroll.
- Edge Case Coverage: the SOP describes any edge cases that the user might encounter, and how to solve
    them  (i.e., "if you don't see button, scroll down")
```

```
- Discrete Action: The SOP only contains one discrete action per step (i.e., the action "click on the
    text bar and type "hello"" should be converted to two separate steps: (1) click on the text bar
    and (2) type "hello")
- Action Relevance: Each action should be true to the task  (i.e., if the task is to find the "grey t-
    shirt" clothing item, then an action which instructs the user to type text in the search bar
    should type the text "grey t-shirt")
- Generality: The steps of the SOP should reflect how to do this task in general and not overfit to the
     specific window size or screen of the demonstration (i.e., "Scroll until you find the row with
    your order" rather than "Scroll 130 pixels down")

# Workflow

The workflow is: "{task_descrip if task_descrip else 'unspecified'}"

# Workflow Demonstration

You are given the following sequence of actions which were sourced from a demonstration of the workflow
    .
The actions are presented in chronological order.

Here are the actions of the workflow:"""
)

prompt__rewrite_sop__close_act: str = (
    lambda sop: f"""

# Standard Operating Procedure

Here are the sequence of steps that you should have followed to complete this workflow:

{sop}

NOTE: The actions may not map 1-to-1 to the steps in the Standard Operating Procedure. i.e. action #3
    may correspond to step #2 (or multiple steps) in the Standard Operating Procedure.
However, as long as the general flow of the workflow is the same, then the workflow is considered to
    have accurately followed the Standard Operating Procedure.
Also note that elements may be interchangeably referred to as buttons or links (the distinction is not
    important).

# Instructions

Given what you observed in the previous sequence of actions, rewrite the Standard Operating Procedure
    to increase clarity and accuracy. If any of the steps are missing, or if any of the steps were
    performed out of order, then the Standard Operating Procedure should be updated to correct these
    mistakes.

Provide your answer as a numbered list with the following format:
1. The first action to be taken goes here
2. The second action to be taken goes here
3. The third action goes here ...

Please write the new updated Standard Operating Procedure below using the guidelines from the SOP
    Rubric above:
"""
)
```

Prompt 17: The introduction and closing parts of the prompt for the version of the SOP Improvement where the action trace is provided to the model in addition to the SOP. The model is asked to output it's response in the form of a SOP given an example structure.

### F.3.2   SOP Ranking

This section contains the prompt utilized for the SOP ranking task. This prompt is also included in our codebase in the following file: ⭘ https://github.com/HazyResearch/wonderbread/blob/main/wonderbread/benchmark/tasks/improvement/sop_ranking/prompts.py.

```
prompt__rank_sop: str = lambda task_descrip, section__sops: f"""# Task
You are a business process management consultant whose job is to evaluate the effeciency of different
    versions of a standard operating procedure (SOP) for the same workflow.

You are given several SOPs, and your job is rank them based on their quality.

# Workflow

The workflow you are evaluating is: "{task_descrip}"

# SOPs

Here are the SOPs you are evaluating. Each SOP is given a distinct ID (i.e. "#1", "#2", etc.), and the
    content of the SOP is enclosed in triple backticks (i.e. "```"). Note that the SOPs are not
    necessarily listed in order of quality, and their IDs are 1-indexed.

{section__sops}

# Question
```

```
Given the SOPs, rank them based on their relative quality. Consider the efficiency of their steps, as
     well as whether or not they achieve the desired workflow.

# Answer

Answer in the following valid JSON format:

{{
    "thinking": "<think step-by-step about the correct SOP ranking; DO NOT use quote marks in your
     response>",
    "pred_ranking": "A list that ranks SOPs by their ID. The first ID in the list corresponds to the
     best SOP, and the last ID corresponds to the worst SOP (i.e., if there are two SOPs and #2 is
     better than #1, then you would return [2, 1])"
}}

Answer:"""
```

Prompt 18: The prompt utilized for the SOP ranking task. As shown in the prompt, the model is asked to structure it's output in the form of a JSON dictionary with specific keys.

