# OpenReview forum: "WONDERBREAD: A Benchmark for Evaluating Multimodal Foundation Models on Business Process Management Tasks"
_NeurIPS.cc/2024/Datasets_and_Benchmarks_Track — NeurIPS 2024 Track Datasets and Benchmarks Poster_

### Official Review · Reviewer_nBnV · 2024-07-21
**Benchmark for understanding the impact of foundation models in business process management tasks**

**Rating:** 8
**Confidence:** 2
**Correctness:** Yes
**Clarity:** Yes

**Review:**

This paper provides a comprehensive benchmark for measuring the effectiveness of foundation models to document, improve and automate business processes. This is potentially very impactful since foundation models are revolutionizing the way that businesses work, yet there is still little work to measure how they impact business processes.

The paper is very well written and the description of the benchmark is easy to follow. Some of the strengths of the paper include:

- While other benchmarks focus only on workflow automation, this paper focuses on the entire lifecycle of a business process including documentation, knowledge transfer, improvement and automation. The paper offers a detailed comparison with previous benchmarks (Table 1)

- The dataset and experiments are published and made available in github. The dataset has a large number of demonstrations (2928) by human experts.

- The experimental evaluation is very thorough, with lots of supporting material in the Appendix

- Baseline results are provided using state-of-the-art foundation models

On the other hand, some of the weaknesses of the paper include:

- The benchmark is based on narrow use cases (e.g., navigation in websites for e-commerce and content management) that are not necessarily representative of the wide variety of business processes in enterprises.

- It does not provide much detail about how the foundation models are actually used in the baseline experiments for the different tasks. For example, I would have liked to see a description of the prompts that were used for the different tasks (including for the LLM-based raters), what the challenges of prompt engineering were, whether the in-context-learning uses few shots, etc.

- A limitation of the work is that it does not consider options such as parameter efficient tuning which can significantly improve quality

**Strengths:**

The paper is very well written and the description of the benchmark is easy to follow. Some of the strengths of the paper include:

- While other benchmarks focus only on workflow automation, this paper focuses on the entire lifecycle of a business process including documentation, knowledge transfer, improvement and automation. The paper offers a detailed comparison with previous benchmarks (Table 1)

- The dataset and experiments are published and made available in github. The dataset has a large number of demonstrations (2928) by human experts.

- The experimental evaluation is very thorough, with lots of supporting material in the Appendix

- Baseline results are provided using state-of-the-art foundation models

**Additional Feedback:**

-

**Documentation:**

Yes

**Limitations:**

Yes

**Opportunities For Improvement:**

- The benchmark is based on narrow use cases (e.g., navigation in websites for e-commerce and content management) that are not necessarily representative of the wide variety of business processes in enterprises.

- It does not provide much detail about how the foundation models are actually used in the baseline experiments for the different tasks. For example, I would have liked to see a description of the prompts that were used for the different tasks (including for the LLM-based raters), what the challenges of prompt engineering were, whether the in-context-learning uses few shots, etc.

- A limitation of the work is that it does not consider options such as parameter efficient tuning which can significantly improve quality

**Relation To Prior Work:**

Yes

**Summary And Contributions:**

This paper provides a comprehensive benchmark for measuring the effectiveness of foundation models to document, improve and automate business processes. This is potentially very impactful since foundation models are revolutionizing the way that businesses work, yet there is still little work to measure how they impact business processes.

---

> ### Author Rebuttal · Authors · 2024-08-17
>
> Thank you so much for your feedback, we really appreciate the consideration! We are happy to see that we are aligned on the high potential impact of evaluating FMs on BPM applications, and that **WONDERBREAD** can help to address gaps in current benchmarks. We also also glad to see that you found our experimental setup and evaluations thorough.
>
> We respond to each of the questions raised below:
>
> ### (1) The benchmark is based on narrow use cases (e.g., navigation in websites for e-commerce and content management) that are not necessarily representative of the wide variety of business processes in enterprises.
>
> Please see our global response above on **(1) Dataset Diversity and Task Selection.**
>
> We have also added the following paragraph to our **Discussion** section to suggest ways to build upon the blueprint laid out in **WONDERBREAD** and mitigate this issue in future versions of the dataset:
>
> > To our knowledge, **WONDERBREAD** is currently the largest dataset for BPM tasks. However, it is still limited in its ability to capture the broad variety of real-world enterprise workflows. Scaling the approach outlined in this paper represents an exciting future research direction. We propose several ways to increase the size and diversity of data: (1) Synthetically generate demonstrations using AI agents trained on existing workflow examples and reject invalid demonstrations, as detailed in [8] [42]. (2) Crowdsource human demonstrations through platforms like Amazon Mechanical Turk. (3) Collaborate with a large enterprise willing to deploy our recording script to collect real-world workflows. (4) Scrape how-to videos and screen recordings of workflows from sites like Youtube.
>
> ### (2) It does not provide much detail about how the foundation models are actually used in the baseline experiments for the different tasks. For example, I would have liked to see a description of the prompts that were used for the different tasks (including for the LLM-based raters), what the challenges of prompt engineering were, whether the in-context-learning uses few shots, etc.
>
> Please see our global response above on **(2) Details on Prompts Used for Experiments and Evaluations.**
>
> In terms of challenges faced, we did have initial difficulty in getting our LLM-based automated evaluations to align with human ratings for several tasks, as noted in **Appendix Section C.2 - LLM-Based Evaluation: Question Answering**. We went through several rounds of prompt engineering in which we sampled a small subset of LLM-based evaluations, compared the ratings with human annotations, and identified common areas of disagreement. We then modified the prompt to emphasize certain desired behaviors (e.g. being harsher at grading on specific axes) and repeated this process twice before achieving the higher agreements demonstrated in **Appendix Table 8** and **Appendix Table 9**. Please also see the newly added **Appendix Section F** for full prompts used.
>
> ### (3) A limitation of the work is that it does not consider options such as parameter efficient tuning which can significantly improve quality
>
> As all of the models used for our state-of-the-art baseline results were closed source, we did not have access to these models’ underlying weights and thus could not conduct any finetuning. However, this is definitely an interesting area for future research, and we are excited to see what methodological advances our benchmark and dataset inspired as well!

---

> > ### Author Rebuttal · Authors · 2024-08-25
> >
> > We thank the reviewer for their time and thoughtful comments, and appreciate the thorough review!
> >
> > With the discussion period ending later this week, we wanted to ask if there was anything missing in our response or updated manuscript that the reviewer still had questions about. Thank you again for the time, and appreciate the consideration!

---

### Official Review · Reviewer_4WGS · 2024-07-25
**Interesting work with room for improvements**

**Rating:** 6
**Confidence:** 4
**Clarity:** The paper is well written and easy to…

**Review:**

WONDERBREAD introduces a pioneering benchmark for evaluating multimodal foundation models on diverse BPM tasks, effectively addressing critical gaps in the field, though it falls short on data diversity and scalability analysis.

**Strengths:**

- The introduction is clearly written and effectively identifies the significant gap in BPM research and practice, which well motivates this work.
- This paper introduces the first benchmark for evaluating multimodal LLMs on diverse BPM tasks, using a human-annotated dataset that includes intent, recording, action trace, key frames, and standard operating procedures (SOPs).
- Six novel BPM tasks are defined, covering a range from workflow documentation to knowledge transfer and process improvement, with a thorough evaluation of these tasks against state-of-the-art LLMs.

**Additional Feedback:**

Considering the current limitations in data diversity and volume, what strategies are proposed or could be considered to scale the size of the dataset to enhance model training and performance?

**Correctness:**

The dataset is constructed in a sound way, and the experiment design is reasonable to me.

**Documentation:**

Sufficient details are provided about the proposed dataset.

**Ethics:**

I don't have any ethical concerns about this work.

**Limitations:**

The limitations are discussed in section 6, where the authors mention that the data lacks real-world scenarios and the results do not cover open-source models.

**Opportunities For Improvement:**

- Limited Data Diversity: The data is sourced from only four websites, which may not adequately represent the diversity of BPM scenarios and could limit the generalizability of the findings.
- Insufficient Scalability Analysis: The paper does not discuss how model performance might scale with an increased number of training examples.

**Relation To Prior Work:**

The prior work is clearly covered and the proposed dataset is well compared with the prior work.

**Summary And Contributions:**

The paper presents WONDERBREAD, the first benchmark designed specifically for evaluating multimodal LLMs in business process management (BPM) tasks beyond mere automation. Existing ML benchmarks in BPM predominantly focus on full end-to-end automation, while this benchmark addresses more varied BPM functions such as documentation and process improvement. The benchmark includes a dataset of 2928 human-annotated workflow demonstrations and introduces six novel BPM tasks derived from real-world applications. The evaluation results reveal that while state-of-the-art FMs excel at generating documentation, they underperform in tasks requiring the validation of workflow completion.

---

> ### Author Rebuttal · Authors · 2024-08-17
>
> We appreciate your thorough feedback and insights! Your suggestions have significantly improved our manuscript, and we're grateful for the time you dedicated to reviewing our work. We are glad that to se that you appreciated the thoroughness of our evaluations and the depth of our coverage of BPM tasks.
>
> Below, we address each point you raised in your review:
>
>
> ### (1) Limited Data Diversity: The data is sourced from only four websites, which may not adequately represent the diversity of BPM scenarios and could limit the generalizability of the findings.
>
> Please see our global response above on **(1) Dataset Diversity and Task Selection**.
>
> First, **WONDERBREAD** is currently the largest BPM dataset assembled. So despite its limitations, we believe it still represents a valuable contribution to the field as-is.
>
> Second, please note that despite collecting workflows from only 4 websites, we curate a diverse set of 6 BPM scenarios covering workflow documentation, knowledge transfer, and improvement. These BPM tasks are formulated in a general, workflow agnostic manner (i.e. "documenting a workflow" is a skill that can be applied to any workflow regardless of its source website or application). Thus, as more workflows are added to **WONDERBREAD** these tasks can be seamlessly reapplied to them.
>
> Third, we believe that the 4 websites we use provide fairly decent coverage of typical digital tasks. These websites are sourced from the WebArena [2] benchmark, a popular workflow automation benchmark that contains self-hosted clones of Adobe Magento (an enterprise content management system), Gitlab (an enterprise code repository), Postmill (a forum), and an ecommerce website. While these sites do not cover the entire space of enterprise workflows, they contain non-trivial workflows representative of general digital tasks (the SOTA model has a success rate of 35% [1])
>
> There are several advantages from using these 4 web applications as the source of our dataset. First, these websites are free and can be self-hosted, which enables fairer benchmarking and reproducibility (in contrast to enterprise workflows which require proprietary or pay-to-use software). Second, these websites are shared with another well-known benchmark called WebArena [2] which has been well-studied in the automation literature, and thus our dataset can also be utilized by researchers building automation agents.
>
> [1] Sodhi, Paloma, et al. ‘SteP: Stacked LLM Policies for Web Actions’.​​[2] Zhou, Shuyan, et al. "Webarena: A realistic web environment for building autonomous agents."
>
>
> ### (2) Insufficient Scalability Analysis: The paper does not discuss how model performance might scale with an increased number of training examples.
>
> The primary focus of this first version of **WONDERBREAD** was benchmarking and evaluation, and thus we do not do any model training in our experiments. This meant that measuring the impact of scaling our dataset on model training fell outside the scope of this initial work.
>
> However, we do agree that the ~3,000 high quality workflow demonstrations in **WONDERBREAD** would be extremely useful in training future BPM and workflow automation models. As the focus of this paper was on datasets + benchmarks, however, we left answering such methodological questions to future work. We are equally excited to see how the performance of models built on our dataset scale!
>
> ### (3) Considering the current limitations in data diversity and volume, what strategies are proposed or could be considered to scale the size of the dataset to enhance model training and performance?
>
> This is a great question -- There are several strategies that could be used to scale the size of the **WONDERBREAD** dataset for improved model training and performance. We have added the following paragraph to our Discussion section in response:
>
> > To our knowledge, **WONDERBREAD** is currently the largest dataset for BPM tasks. However, it is still limited in its ability to capture the broad variety of real-world enterprise workflows. Scaling the approach outlined in this paper represents an exciting future research direction. We propose several ways to increase the size and diversity of data: (1) Synthetically generate demonstrations using AI agents trained on existing workflow examples and reject invalid demonstrations, as detailed in [8] [42]. (2) Crowdsource human demonstrations through platforms like Amazon Mechanical Turk. (3) Collaborate with a large enterprise willing to deploy our recording script to collect real-world workflows. (4) Scrape how-to videos and screen recordings of workflows from sites like Youtube.
>
> [8] Bai, Hao, et al. "Digirl: Training in-the-wild device-control agents with autonomous reinforcement learning." arXiv preprint arXiv:2406.11896 (2024).
>
> [42] Pan, Jiayi, et al. "Autonomous evaluation and refinement of digital agents." arXiv preprint arXiv:2404.06474 (2024).

---

> > ### Author Rebuttal · Authors · 2024-08-25
> >
> > We thank the reviewer again for the time and feedback!
> >
> > As the discussion period ends this week, we wanted to ask if there was anything missing in our response or updated manuscript that the reviewer still had questions about, and if there were any additional clarifications that would be helpful in the final evaluation of our work. We hope the reviewer is able to take into account our responses above and updated manuscript into the final evaluation of the work. If there is no update, we certainly respect the reviewer’s decision, and thank you again for the time and thoughtful suggestions.

---

### Official Review · Reviewer_rYLa · 2024-07-31
**an evaluation benchmark on enterprise workflow**

**Rating:** 5
**Confidence:** 4
**Correctness:** Yes
**Clarity:** Yes

**Review:**

Strength
- Compared with existing benchmarks, WONDERBREAD includes BMP tasks such as documentation, knowledge transfer, or process improvement, and step-by-step written guides (SOPs), which can provide a more comprehensive and transparent assessment of models than end-to-end workflow completion rates.
- WONDERBREAD is the high quality of demonstrations achieved through several months of quality assurance.
- WONDERBREAD boasts a sophisticated, automated evaluation framework, replete with algorithmic metrics and assessments powered by Large Language Models (LLMs).

Questions.

1. Could you use an example, such as a physician submitting a medication order, to explain in detail what the four stages of BPM do, so as to accelerate the understanding of BPM?
2. Could you show some examples of qualitative analysis, such as success and failure cases?
3. In Workflow Selection, why tasks that include multiple websites from WebArena are ignored?  Could you please explain in detail how the excluded workflows are designed to be impossible, underspecified, or have incorrect evluations.
4. What is the basis for setting these 6 novel tasks? Is there any reference?
5. "s_i is the ith state of the webpage", what is its format, an image, a json file, or any other else? In Figure 3, how did the videos feed into the LLM? If s_i is represented as an image frame, it should be LVLM not LLM.
6. It is not clear about how to use GPT4 to evaluate in the SOP Generation Description, Question Answering, and SOP Improvement tasks. Could you please show the prompts for these evaluations? Do you consider and solve position bias in the evaluation process?
7. In the SOP Generation Description task, you said there are two kinds of input, (w,(s1,...,sn)) or (w,(s1,a1,...an-1,sn)), but only one performance table for SOP Generation (Table 3) is provided. What is the performance for the other situation?
8. The prompt of LLM for making the output answers for each task is missing. Could you please add the prompts and explain how to get and parse the answer in each task?
9. Why are open source LLMs not involved in the evaluation? Whether only three LLMs are limited?
10. In the "Demo Segmentation", "Question Answering", and "Demo Ranking" tasks, two situations (multiple demos and single demo) are both involved in these tasks. Is it necessary to distinguish between the two to calculate performance?
11. What is the meaning of "V-Measure" in Figure 4? Why does the additional information to the model in the form of an SOP and intent decrease performance for Gemini Pro 1? Is additional information necessary?
12. In Figure 6, why does "soundness" have lower scores? According to the "completeness", "soundness", "clarity", and "compactness" in "Question Answering" task, does the prompt specify the level of achievement required to achieve the corresponding scores in different dimensions? For example, what condition can reach to the "3" in "clarity"?
13. How to obtain the conclusion that multimodal FMs excel at reasoning over short demonstration lengths but struggle with longer workflows, there is no experiment about demonstration length.
14. The 120 questions in "Question Answering" are from [9][20]? How to match these questions to workflows adapately?

**Strengths:**

see Review

**Additional Feedback:**

see Review

**Documentation:**

Yes

**Limitations:**

Yes

**Opportunities For Improvement:**

see Review

**Relation To Prior Work:**

Yes

**Summary And Contributions:**

This paper introduces WONDERBREAD, pioneering the first comprehensive benchmark designed to assess multimodal models across a spectrum of typical process mining tasks. This benchmark encompasses a rich dataset of 2,928 human demonstrations that span videos, images, and textual data, complemented by detailed step-by-step guides known as Standard Operating Procedures (SOPs) and exhaustive action logs. Their emphasis is on the practical application of these models to three pivotal Business Process Management (BPM) tasks—documentation, knowledge transfer, and process enhancement—which have been notably understated in current Machine Learning (ML) benchmarks for workflow automation. WONDERBREAD features an automated evaluation harness with programmatic metrics and LLM-based assessments, providing baseline results for state-of-the-art multimodal models.

---

> ### Author Rebuttal · Authors · 2024-08-17
>
> Thank you so much for the incredibly detailed comments and feedback. Going through all of your suggestions has greatly strengthened our manuscript, and we sincerely thank you for your time and attention to detail when reviewing our paper. We have added a significant number of new figures and additional content thanks to your suggestions. We are glad to see that you resonated with the effort put into ensuring the high quality of our dataset, as well as your appreciation for our automated LLM-based evaluation framework. We respond to each of the issues raised in your review below:
>
> ### (1) Could you use an example, such as a physician submitting a medication order, to explain in detail what the four stages of BPM do, so as to accelerate the understanding of BPM?
>
> Thank you for the suggestion. We have added the following to **Appendix Section A.6 -- Example Hypothetical BPM Project** to provide a concrete example of what each stage of BPM might entail for a hypothetical project:
>
> > For clarity, we provide the following as an example of what a hypothetical BPM project might entail. Let’s say a hospital wants to accelerate the workflow by which admitted patients have their insurance verified. Today, the process is done completely manually by a team of billing specialists. The workflow involves copying the patient’s demographic information into several databases and visiting an insurer’s web portal to verify that the patient’s insurance coverage is accurate and up-to-date. A hypothetical BPM project for accelerating this workflow might progress as follows: (1) Documentation: First, a business development (BD) analyst interviews all of the billing specialists on the team, conducts shadowing sessions over Zoom, and watches screen recordings collected by the team in order to create written documentation of the insurance verification workflow. (2) Knowledge Transfer: After creating a draft of the workflow, the BD analyst hosts a series of in-person brainstorming sessions with the team; the BD analyst identifies several gaps in her current understanding of the workflow, and they collaboratively arrive at a shared consensus of all steps involved in the end-to-end workflow. (3) Improvement: From these conversations, the BD analyst identifies several bottlenecks and inefficiencies; for example, entering the patient’s demographic information into multiple databases that could instead be automatically synced, or waiting for the approval of another department that has a turnaround of one week but isn’t strictly necessary. The BD analyst then draws a new, more streamlined workflow diagram and shares her findings with the billing team to implement. (4) Automation: Based on these observations, the billing team believes that several of the subtasks within this new workflow might be automatable. They enlist the help of the hospital’s IT department to build an integration between their two database applications to automate this data entry. They also work on developing a robotic process automation (RPA) bot that can navigate multiple screens to automatically submit forms to insurers.
>
> ### (2) Could you show some examples of qualitative analysis, such as success and failure cases?
>
> This is a great suggestion, thank you for the comment! In response, we have added **Appendix Figure 11** and **Appendix Figure 12**, which provide examples of success and failure cases for two of our tasks, as well as line-by-line qualitative analysis of GPT-4 and Claude model outputs.
>
> ### (3) In Workflow Selection, why tasks that include multiple websites from WebArena are ignored? Could you please explain in detail how the excluded workflows are designed to be impossible, underspecified, or have incorrect evaluations.
>
> Thank you for the comment, we should have clarified this better. In response, we have added the following paragraph to **Appendix A.2.1 Workflow Selection**:
>
> > First, several workflows in WebArena are designed to be impossible. These are the workflows that have a correct answer marked as "N/A". For example, one workflow has the intent "What are the main criticisms of this product?'' and marks the correct answer as "N/A" since there are no criticisms. We remove all of these workflows. An example of a \textit{de facto} impossible workflow is "Assign the issue regarding flash alerts to myself and primer.'' Though there is a non-N/A answer for this workflow, upon manual inspection we found that the Gitlab interface does not actually allow issues to be assigned to more than one user, and thus we removed it from our dataset. Underspecified workflows are those whose answer we found arbitrary upon manual inspection. For example, an intent such as "Show me the email address of the customer who is the most unhappy with Circe fleece'' is underspecified as the phrase "most unhappy'' is unquantifiable when there are multiple one-star reviews. We remove all of these underspecified workflows. Finally, we exclude workflows that have valid intents but whose expected answers were deemed incorrect upon manual inspection. Example workflows from the Webarena dataset with these mistakes include "the number of commits of the contributor who has the most commits to branch main'' in Gitlab being stated incorrectly or that "the amount spent on home decoration shopping during 1/29/2023'' being calculated incorrectly are also excluded. Finally, we ignore the 23 workflows in  WebArena that include multiple websites. We do this for simplicity as our recording script could only handle one website at a time. This left us with a final total of 598 workflows.
>
> (continued below)

---

> > ### Author Rebuttal · Authors · 2024-08-17
> >
> > (continued...)
> >
> > ### (4) What is the basis for setting these 6 novel tasks? Is there any reference?
> >
> > We sourced these 6 tasks from two main sources: (1) a general review of the process mining and workflow automation literature, in particular [6, 19, 23, 34, 38, 46]; and (2) prior case studies that applied foundational models to narrow BPM use cases, in particular [9, 18, 20, 24, 39, 49, 53]. Ultimately, the 6 tasks we chose were based on their ability to effectively represent a broad set of practical use cases, as well as their potential to benefit from the application of multimodal foundational models.
> >
> > [6] ​​Adriano Augusto, Raffaele Conforti, Marlon Dumas, Marcello La Rosa, Fabrizio Maria Maggi, Andrea Marrella, Massimo Mecella, and Allar Soo. Automated discovery of process models from event logs: Review and benchmark. IEEE transactions on knowledge and data engineering, 31(4):686–705, 2018.
> >
> > [9] Alessandro Berti and Mahnaz Sadat Qafari. Leveraging large language models (llms) for process mining (technical report). arXiv preprint arXiv:2307.12701, 2023.
> >
> > [18] Marlon Dumas, Fabiana Fournier, Lior Limonad, Andrea Marrella, Marco Montali, Jana-Rebecca Rehse, Rafael Accorsi, Diego Calvanese, Giuseppe De Giacomo, Dirk Fahland, et al. Ai-augmented business process management systems: a research manifesto. ACM Transactions on Management Information Systems, 14(1):1–19, 2023.
> >
> > [19] Marlon Dumas, Marcello La Rosa, Jan Mendling, Hajo A Reijers, et al. Fundamentals of business process management, volume 2. Springer, 2018.
> >
> > [20] Dirk Fahland, Fabian Fournier, Lior Limonad, Inna Skarbovsky, and Ava JE Swevels. How well can large language models explain business processes? arXiv preprint arXiv:2401.12846, 2024.
> >
> > [23] Fabian Friedrich, Jan Mendling, and Frank Puhlmann. Process model generation from natural language text. In Advanced Information Systems Engineering: 23rd International Conference, CAiSE 2011, London, UK, June 20-24, 2011. Proceedings 23, pages 482–496. Springer, 2011.
> >
> > [24] Michael Grohs, Luka Abb, Nourhan Elsayed, and Jana-Rebecca Rehse. Large language models can accomplish business process management tasks. In International Conference on Business Process Management, pages 453–465. Springer, 2023.
> >
> > [34] Volodymyr Leno, Artem Polyvyanyy, Marlon Dumas, Marcello La Rosa, and Fabrizio Maria Maggi. Robotic process mining: vision and challenges. Business & Information Systems Engineering, 63:301–314, 2021.
> >
> > [38] Jorge Munoz-Gama, Niels Martin, Carlos Fernandez-Llatas, Owen A Johnson, Marcos Sepúlveda, Emmanuel Helm, Victor Galvez-Yanjari, Eric Rojas, Antonio Martinez-Millana, Davide Aloini, et al. Process mining for healthcare: Characteristics and challenges. Journal of Biomedical Informatics, 127:103994, 2022.
> >
> > [39] Vinod Muthusamy, Yara Rizk, Kiran Kate, Praveen Venkateswaran, Vatche Isahagian, Ashu Gulati, and Parijat Dube. Towards large language model-based personal agents in the enterprise: Current trends and open problems. In Findings of the Association for Computational Linguistics: EMNLP 2023, pages 6909–6921, 2023.
> >
> > [46] Lars Reinkemeyer. Process mining in action. Process Mining in Action Principles, Use Cases and Outlook, 2020.
> >
> > [49] Yara Rizk, Praveen Venkateswaran, Vatche Isahagian, Austin Narcomey, and Vinod Muthusamy. A case for business process-specific foundation models. In International Conference on Business Process Management, pages 44–56. Springer, 2023.
> >
> > [53] Maxim Vidgof, Stefan Bachhofner, and Jan Mendling. Large language models for business process management: Opportunities and challenges. arXiv preprint arXiv:2304.04309, 2023.
> >
> > ### (5) "s_i is the ith state of the webpage", what is its format, an image, a json file, or any other else? In Figure 3, how did the videos feed into the LLM? If s_i is represented as an image frame, it should be LVLM not LLM.
> >
> > Apologies for the confusion! We have added the following to **Appendix Section B - Benchmark Tasks** to clarify this:
> >
> > > We represent each state $s_i$ as a $.png$ image which contains a single frame extracted from the screen recording of the demonstration. We select these frames by first logging the timestamp of every action $a_i$ taken during the demonstration, then iterating through every frame of the screen recording video and extracting the frames corresponding to those action timestamps. We refer to these as "key frames."
> >
> > And yes great catch re:LLM nomenclature. We have fixed **Figure 3** (and the other figures) to say “Multimodal FM” instead of “LLM”.
> >
> > ### (6) It is not clear about how to use GPT4 to evaluate in the SOP Generation Description, Question Answering, and SOP Improvement tasks. Could you please show the prompts for these evaluations? Do you consider and solve position bias in the evaluation process?
> >
> > In response to the queries, we have detailed all relevant prompts and methodologies in **Appendix Section F** of our report. The evaluation prompts utilized in the SOP Generation, Question Answering, and SOP Improvement tasks are included along with a brief description and a link to their implementation in our Github repository. We did not consider position bias in the current evaluation process, but will add it to the issue tracker for our next release version.
> >
> > (continued below)

---

> > ### Author Rebuttal · Authors · 2024-08-17
> >
> > (continued...)
> >
> >
> > ### (7) In the SOP Generation Description task, you said there are two kinds of input, (w,(s1,...,sn)) or (w,(s1,a1,...an-1,sn)), but only one performance table for SOP Generation (Table 3) is provided. What is the performance for the other situation?
> >
> > Apologies for the confusion -- to clarify, we have renamed $w$ to $I$ (for “intent”) and thus there are three types of inputs that we evaluate: (1) intent only -- $(I)$, (2) intent plus screenshots -- $(I, (s_1, ..., s_n))$, (3) intent plus screenshots plus actions -- $(I, (s_1, a_1, ..., a_{n-1}, s_n))$.
> >
> > We have added the following sentence to **Appendix Section B.1 -- SOP Generation Description** to clarify the mapping between these input formats and our results:
> >
> > > Given a demonstration $d = (I, SOP,  (s_1, a_1, s_2, a_2, ..., a_{n-1}, s_n))$, we provide the model with either $(I)$, $(I, (s_1, ..., s_n))$, or $(I, (s_1, a_1, ..., a_{n-1}, s_n))$. In our Results Table 3, these correspond to rows with one checkmark under the "Intent" column, two checkmarks under the "Intent" and "Keyframes" columns, and three checkmarks under the "Intent", "Keyframes", and "Trace" columns, respectively.
> >
> > ### (8) The prompt of LLM for making the output answers for each task is missing. Could you please add the prompts and explain how to get and parse the answer in each task?
> >
> > In response to this feedback, we have included all prompts in the **Appendix Section F** of our manuscript.  The prompt for each task explicitly describes the form of the response that is expected from the model, which is generally either a JSON type dictionary with a specific structure or a stand alone output (such as a number or letter). We have included brief descriptions with each prompt that indicate which output is expected in this new section.
> >
> > ### (9) Why are open source LLMs not involved in the evaluation? Whether only three LLMs are limited?
> >
> > For our initial baseline results, we focused on the three most capable state-of-the-art models to provide the strongest possible baselines -- namely, Claude, GPT-4, and Gemini. These all happened to be closed source models, but they are publicly accessible via APIs (which we leverage for our paper).
> >
> > We believe that this approach was merited for establishing a strong baseline to which future models (both open and closed source) can be compared. Additionally, as a multimodal benchmark, our tasks heavily depend on a model being able to process interleaved text and images. To our knowledge, there aren’t capable open source VLMs that accept interleaved text and images as input. Please also see **Table 3** in [1] (another benchmark that did not use open source models for interleaved text/images).
> >
> > However, we do agree that developing new open source models with these capabilities is an important area of future research, and we look forward to adding them to our benchmark as they are developed.
> >
> > And if there is a specific model you would like to see evaluated, please let us know and we will be happy to include it in our baseline results!
> >
> > [1] Lin, Kevin Qinghong, et al. "VideoGUI: A Benchmark for GUI Automation from Instructional Videos." arXiv preprint arXiv:2406.10227 (2024).
> >
> > (continued below)

---

> > ### Author Rebuttal · Authors · 2024-08-17
> >
> > (continued...)
> >
> > ### (10) In the "Demo Segmentation", "Question Answering", and "Demo Ranking" tasks, two situations (multiple demos and single demo) are both involved in these tasks. Is it necessary to distinguish between the two to calculate performance?
> >
> > Apologies for the confusion -- for the **“Demo Segmentation”** and **“Demo Ranking”** (aka **“SOP Ranking”**) tasks, our experiments only consider multiple demos. For **“Demo Segmentation”**, we always concatenate $k = 3$ demonstrations together for the results we report -- running **“Demo Segmentation”** with a single demonstration is a degenerate case in that every demo comes from the same workflow and thus there is nothing to segment. For **“Demo/SOP Ranking”**, we only consider workflows with at least 3 demonstrations -- again, the single demo case is degenerate because we’d just be ranking a single item.
> >
> > For **“Question Answering”**, there are questions that include multiple demos and questions that only include single demos. The split between question templates is as follows:
> >
> > Single Demo:
> >
> > 1. Explain what the most common failure modes might be for a user performing this workflow.
> > 2. How would a user completing the task know that the workflow is completed?
> > 3. What is the purpose of doing this workflow?
> > 4. What if instead of X we wanted to do Y. How would you change this workflow to accomplish that?
> > 5. Why does the user click the button X in step #Z?
> > 6. Why does the user click the button X in screenshot #Y?
> > 7. Why does the user type the string X in step #Z?
> > 8. Why does the user type the string X in screenshot #Y?
> > 9. How would a user completing the task know that the workflow is completed?
> >
> > Multiple Demos:
> >
> > 10. Here are two workflows. Please identify the key differences between them.
> > 11. Here are two demonstrations, one of which is more efficient than the other. Please describe ways to improve the less optimal workflow.
> >
> > We have added the following paragraph to **Appendix Section A.4 - Question Answering Dataset Questions** to clarify this:
> >
> > > Question templates #1-9 only involve reasoning over a single demonstration, but #10-11 require reasoning over multiple demonstrations.
> >
> > Per your suggestion, we redid our analysis splitting by questions involving single v. multiple demonstrations. We observe slight drops in “completeness” and “soundness” scores (which penalize answers for being factually incorrect), but see minimal differences in the “clarity” and “compactness” metrics (which penalize answers for being incoherent or poorly written) as shown in the results below:
> >
> > Multi:
> >
> > | Model      | Completeness | Soundness | Clarity | Compactness | Average |
> > |------------|--------------|-----------|---------|-------------|---------|
> > | Human      | 3.00         | 3.00      | 2.43    | 2.93        | 2.84    |
> > | GPT-4      | 2.13         | 2.40      | 2.97    | 2.97        | 2.62    |
> > | GeminiPro  | 1.63         | 1.77      | 2.83    | 2.97        | 2.30    |
> > | Claude3    | 1.40         | 1.63      | 2.20    | 2.93        | 2.04    |
> >
> > Single:
> >
> > | Model      | Completeness | Soundness | Clarity | Compactness | Average |
> > |------------|--------------|-----------|---------|-------------|---------|
> > | Human      | 3.00         | 3.00      | 2.71    | 2.86        | 2.89    |
> > | GPT-4      | 2.22         | 2.54      | 2.96    | 2.81        | 2.63    |
> > | GeminiPro  | 1.87         | 2.28      | 2.83    | 2.94        | 2.48    |
> > | Claude3    | 1.61         | 1.90      | 2.18    | 2.50        | 2.05    |
> >
> > ### (11) What is the meaning of "V-Measure" in Figure 4? Why does the additional information to the model in the form of an SOP and intent decrease performance for Gemini Pro 1? Is additional information necessary?
> >
> > V-Measure is the harmonic mean of homogeneity (how similar the contents of a cluster are to each other) and completeness (how well samples of the same class are grouped into the same cluster). It ranges from 0 (low quality clustering) to 1 (high quality clustering), and is formally defined as follows: $V = 2 * \frac{h * c}{h + c}$ with $h = 1 - \frac{H(C | K)}{H(C)}$ and $c = 1 - \frac{H(K | C)}{H(K)}$ where $H(x)$ is the entropy of $x$ and $C$ are the classes while $K$ are the clusters.
> >
> > This additional information is not necessary for segmenting the demonstrations. We hypothesize that the additional information causes a decrease in performance for GeminiPro by distracting it from the count of states that it needs to assign to tasks. This is supported by the following analysis of the number of blanks in GeminiPro’s response (i.e. the number of states that it does not assign to any workflow), which indicates that this number increases the more information is provided in the prompt for GeminiPro:
> >
> > | Input Format | Count of Blanks for Demo Segmentation task |
> > |------------|--------------|
> > | TD+KF+SOP  |    298 |
> > KF+SOP  |   100 |
> > KF   |  30 |
> >
> > We consider any state that does not have an explicit assignment to be incorrect. Thus, the model is penalized for abstaining from state assignments. In fact, we find four demonstrations in which GeminiPro gave a completely blank response: 539 @ 2024-01-07-17-04-44, 350 @ 2024-01-04-18-16-40, 468 @ 2024-01-06-15-26-01, 333 @ 2024-01-01-13-19-13
> >
> > We have added a note to this effect in the results of **Section 5.2 - Knowledge Transfer:**
> >
> > > On inspection, we find that the frequency at which Gemini Pro outputs blank state mappings (i.e. to not assign a keyframe to any candidate workflow, which under our evaluation framework gets penalized as an incorrect mapping) increases with longer prompts, thus indicating a worse ability to follow the full context of the prompt.
> >
> > (continued below)

---

> > ### Author Rebuttal · Authors · 2024-08-17
> >
> > (continued...)
> >
> > ### (12) In Figure 6, why does "soundness" have lower scores? According to the "completeness", "soundness", "clarity", and "compactness" in "Question Answering" task, does the prompt specify the level of achievement required to achieve the corresponding scores in different dimensions? For example, what condition can reach to the "3" in "clarity"?
> >
> > Thank you for the clarification question. In response, we have added **Appendix Section F.1 - Prompts for Question Answering Task Evaluation**, which contains the full prompts used for each metric. This section now shows the criteria for each score, as well as in-context examples to show the level of achievement required for a response to achieve a high or low score.
> >
> > We have also added the following details to the results portion of **Section 5.2 - Question Answering** to address your question about the different metrics:
> >
> > > All models perform well in ``compactness'' and ``clarity'' but score lower on ``soundness'' and ``completeness.'' The former two are measures of the syntactic quality of writing, while the latter two are measures of the accuracy of the content.As ``soundness'' measures whether an answer avoids containing inaccurate details, these lower scores can be explained by the tendency of LLMs to hallucinate and infer information based on patterns learned from training data (which includes content from various similar websites like GitHub, GitLab, and Amazon) that are not present in the specific demonstration being evaluated [28]. The lower scores on ``completeness'' may be due to their inability to fully attend to the information contained in a prompt containing multiple states and actions across a demonstration [37, 39], leading to occasional omissions of relevant details despite scoring highly on the syntactic quality of their writing.
> >
> > [28] Huang, Lei, et al. "A survey on hallucination in large language models: Principles, taxonomy, challenges, and open questions." arXiv preprint arXiv:2311.05232 (2023).
> >
> > [37] Liu, Nelson F., et al. "Lost in the Middle: How Language Models Use Long Contexts. CoRR abs/2307.03172 (2023)." arXiv preprint arXiv:2307.03172 10 (2023).
> >
> > [39] Maitland, Amy, Ross Fowkes, and Stuart Maitland. "Can ChatGPT pass the MRCP (UK) written examinations? Analysis of performance and errors using a clinical decision-reasoning framework." BMJ open 14.3 (2024): e080558.
> >
> > ### (13) How to obtain the conclusion that multimodal FMs excel at reasoning over short demonstration lengths but struggle with longer workflows, there is no experiment about demonstration length.
> >
> > Thank you for the comment and the close read. This conclusion was an artifact of an earlier experiment that, upon review, did not end up showing this, but this sentence was not removed from the manuscript. We have removed it to address your comment and replaced it with a more appropriate characterization of our results in **Section 5 - Results:**
> >
> > > Our initial results show that current multimodal FMs, including GPT-4, Gemini, and Claude, excel at generating documentation which captures the higher-level characteristics of workflows but struggle with finer-grained analyses such as question answering and workflow quality assessment. Our zero-shot evaluations focus on the out-of-the-box capabilities of these models across 162 workflows with rankings. Some models were excluded from specific tasks due to API budget and quota limitations.
> >
> >
> > ### (14) The 120 questions in "Question Answering" are from [9][20]? How to match these questions to workflows adapately?
> >
> > Apologies for the confusion. The 120 questions in our “Question Answering” dataset were inspired by [9][20], but we manually created all of these questions ourselves. Thus, this is an entirely novel dataset. Please see **Appendix Section A.4 - Question Answering Dataset Questions** for the list of question templates that we created (reproduced below):
> >
> > 1. Explain what the most common failure modes might be for a user performing this workflow.
> > 2. How would a user completing the task know that the workflow is completed?
> > 3. What is the purpose of doing this workflow?
> > 4. What if instead of X we wanted to do Y. How would you change this workflow to accomplish that?
> > 5. Why does the user click the button X in step #Z?
> > 6. Why does the user click the button X in screenshot #Y?
> > 7. Why does the user type the string X in step #Z?
> > 8. Why does the user type the string X in screenshot #Y?
> > 9. How would a user completing the task know that the workflow is completed?
> > 10. Here are two workflows. Please identify the key differences between them.
> > 11. Here are two demonstrations, one of which is more efficient than the other. Please describe ways to improve the less optimal workflow.
> >
> > We have further added the following text to **Appendix Section A.4 - Question Answering Dataset Questions** in response to your comment to address the lack of clarity that you noted:
> >
> > > After creating these question templates based on our review of question types asked in prior work on process mining [9][20], we then transformed them into concrete questions instantiated with specific demonstrations from our dataset. In other words, turning “Why does the user click the button X in step #Z?” into “Why does the user click on ""Not Approved"" in step #4?” for demonstration 79 @ 2023-12-27-22-50-34. This was accomplished in three steps. First, for each question template we first came up with a list of characteristics that a linked demonstration would need. For example, a workflow with no button clicks would not be a viable candidate for the question template “Why does the user click the button X in step #Z?.” Next, we randomly sampled demonstrations without replacement from our set of Gold demonstrations until we came up with 10 instantiations of each question (20 for question template #11). Finally, we conducted two rounds of manual review to write “ground truth” answers and ensure each question was instantiated correctly.

---

> > > ### Author Rebuttal · Authors · 2024-08-25
> > >
> > > We wanted to thank the reviewer again for their time and comments!
> > >
> > > As the discussion period ends this week, we wanted to ask if there was anything missing in our response or updated manuscript that the reviewer still had questions about, and if there were any additional clarifications that would be helpful in the final evaluation of our work. Please let us know either way -- we really appreciate the very thoughtful questions and review of our work, and believe it has been made significantly stronger from your feedback so far!

---

### Author Rebuttal · Authors · 2024-08-17

# Global Response to Reviews

We want to start by thanking all of our reviewers for their time and attention to detail!

Overall, we are happy to see the positive reception of our work **WONDERBREAD**, a _**“pioneering benchmark”**_ (4WGS25) for business process management (BPM) tasks.

We are glad that the reviewers appreciated **WONDERBREAD** for being _**“well motivate[d]”**_ (4WGS25) and experimentally _**“sound”**_ (4WGS25) with an _**“emphasis...on...practical application”**_ (rYLa30) and the potential to be _**“very impactful”**_ (nBnV21) in addressing _**“critical gaps in the field”**_ (4WGS25) of ML for workflow automation.

In contrast to prior work focused solely on end-to-end workflow automation, reviewers noted that **WONDERBREAD** provides _**“a more comprehensive and transparent assessment of models”**_ (rYLa30) by covering the _**“entire lifecycle”**_ (nBnV21) of workflow improvement as the _**“first benchmark”**_ (4WGS25) to call attention to the broader universe of BPM tasks.

Reviewers recognized the _**“high quality”**_ (rYLa30) of our underlying multimodal dataset, noting that it covers a _**“comprehensive”**_ (nBnV21) set of BPM tasks that have been _**“notably understated in current benchmarks”**_ (rYLa30). In addition to the underlying dataset and benchmark tasks, the evaluation harness in **WONDERBREAD** was noted for being _**“sophisticated”**_ (rYLa30) and _**“thorough”**_ (4WGS25, nBnV21). Finally, we are glad that the reviewers found our paper to be _**“clearly written”**_ (4WGS25) and _**“easy to follow”**_ (nBnV21).

In addition to the strengths noted above, reviewers provided constructive feedback which we have implemented and which has made our paper significantly stronger. We thank them for taking the time to read our manuscript and offer these suggestions.

In response, we have:

1. **Uploaded an updated PDF** of our manuscript with all suggested changes implemented. Changes are highlighted in **blue text.**
2. **We provide a global response** below to common feedback across reviewers, namely questions around **(1) the diversity of the dataset and BPM tasks** and **(2) details on the prompts / input formats of our experiments.**
3. We have **added detailed line-by-line responses to each reviewer** under their respective sections.

Our responses are as follows:

(continued below)

---

> ### Author Rebuttal · Authors · 2024-08-17
>
> (continued...)
>
> ## (1) Dataset Diversity and Task Selection
>
> We agree that adding more diverse data is an important direction for future research, and will aim to increase the depth and breadth of data in future versions of **WONDERBREAD**. As is, **WONDERBREAD** in its current state is still the largest dataset assembled for BPM tasks, containing ~3k human demonstrations across 598 different workflows. Each demonstration includes rich multimodal data (video, text, HTML, action logs, etc.). Given its status as the “first” benchmark for BPM tasks (4WGS25), we believe **WONDERBREAD** is still a useful step forward for the field despite not covering every possible enterprise workflow.
>
> Going forward, **WONDERBREAD** offers a blueprint for compiling an even more diverse dataset. In response to these questions, we have added the following paragraph to our **Discussion** section outlining ways to increase the diversity of data in future versions of **WONDERBREAD**:
>
> > To our knowledge, **WONDERBREAD** is currently the largest dataset for BPM tasks. However, it is still limited in its ability to capture the broad variety of real-world enterprise workflows. Scaling the approach outlined in this paper represents an exciting future research direction. We propose several ways to increase the size and diversity of data: (1) Synthetically generate demonstrations using AI agents trained on existing workflow examples and reject invalid demonstrations, as detailed in [8] [42]. (2) Crowdsource human demonstrations through platforms like Amazon Mechanical Turk. (3) Collaborate with a large enterprise willing to deploy our recording script to collect real-world workflows. (4) Scrape how-to videos and screen recordings of workflows from sites like Youtube.
>
> Additionally, an important distinction to note is that although the workflows being demonstrated in **WONDERBREAD** do not capture every possible enterprise workflow, the 6 BPM tasks which we define on top of these workflows are very diverse and do cover a wide range of enterprise use cases. As detailed in **Section 4 - Benchmarks**, these tasks are defined as follows:
>
> 1. Automatically generating documentation for a workflow (“SOP Generation”)
> 2. Segmenting a video demonstration into discrete workflows (“Demo Segmentation”)
> 3. Answering arbitrary questions about a workflow (“Question Answering”)
> 4. Validating that a workflow was successfully completed (“Demo Validation”)
> 5. Ranking workflow executions sourced from across different annotators (“SOP Ranking”)
> 6. Improving standard operating procedures (“SOP Improvement”).
>
> This is a fairly general set of capabilities, and by far the largest set of BPM tasks assembled into a single benchmark. As noted by reviewer (nBnV21), our benchmark covers the “entire lifecycle” of BPM projects and thus offers significantly greater dataset and task diversity than prior work focused solely on end-to-end automation (as shown in **Table 1**).
>
> Another question the reviewers had is why our workflows are sourced from 4 websites. These websites are sourced from the WebArena [2] benchmark, and are self-hosted clones of Adobe Magento (an enterprise content management system), Gitlab (an enterprise code repository), Postmill (a forum), and an ecommerce website. While these sites do not cover the entire space of enterprise workflows, they contain non-trivial workflows representative of general digital tasks (the SOTA model has a success rate of 35% [1])
>
> There are several advantages from using these 4 web applications as the source of our dataset. First, these websites are free and can be self-hosted, which enables fairer benchmarking and reproducibility (in contrast to enterprise workflows which require proprietary or pay-to-use software). Second, these websites are shared with another well-known benchmark called WebArena [2] which has been well-studied in the automation literature, and thus our dataset can also be utilized by researchers building automation agents.
>
> [1] Sodhi, Paloma, et al. ‘SteP: Stacked LLM Policies for Web Actions’.
> ​​[2] Zhou, Shuyan, et al. "Webarena: A realistic web environment for building autonomous agents."
>
> ## (2) Details on Prompts Used for Experiments and Evaluations
>
> There were several questions asking for more details on our experimental setup, e.g. the specific prompts used for generating model outputs, the prompts used for our LLM-based evaluations, etc. We thank our reviewers for pointing out these areas of confusion, and we have added significant detail to our manuscript to make this clearer to the reader in response -- our paper is much stronger thanks to these suggestions. In particular, we have added all prompts used to **Appendix Section F** in response.
>
> This **Appendix Section F** now includes all prompts utilized in our benchmark, organized by task, for both generating model outputs and evaluating them. For each prompt utilized, we also include a brief description and a link to where the prompt exists in our codebase within our public Github repository for quick reference.

---

### Decision · Program_Chairs · 2024-09-26

**Decision:**

Accept (Poster)

**Comment:**

This paper provides a benchmark for evaluating multi-modal LLM performance on business process management tasks, which are of practical interest. Most reviewers agree that this is important for LLM application in businesses work, yet there is still little work in this area. Major concerns raised by the reviewers are the scale/diversity and experiment details. The authors gave careful and rich response to explain. I believe technical details are mostly clear now and the scale/diversity of the database is satisfactory (esp. when this is the first comprehensive benchmark in this area). Hence, the current version of of good shape. But I will leave the consideration of whether the BPM task is of general interest to the research community to the program chair.